# ProG: A Graph Prompt Learning Benchmark

**Chenyi Zi**[1*], **Haihong Zhao**[1*], **Xiangguo Sun**[2,✉], **Yiqing Lin**[3], **Hong Cheng**[2], **Jia Li**[1]

[1]Hong Kong University of Science and Technology (Guangzhou)
[2]Department of Systems Engineering and Engineering Management,
and Shun Hing Institute of Advanced Engineering, The Chinese University of Hong Kong
[3]Tsinghua University
✉ corresponding author: `xiangguosun@cuhk.edu.hk`

## Abstract

Artificial general intelligence on graphs has shown significant advancements across various applications, yet the traditional 'Pre-train & Fine-tune' paradigm faces inefficiencies and negative transfer issues, particularly in complex and few-shot settings. Graph prompt learning emerges as a promising alternative, leveraging lightweight prompts to manipulate data and fill the task gap by reformulating downstream tasks to the pretext. However, several critical challenges still remain: how to unify diverse graph prompt models, how to evaluate the quality of graph prompts, and to improve their usability for practical comparisons and selection. In response to these challenges, we introduce the first comprehensive benchmark for graph prompt learning. Our benchmark integrates **SIX** pre-training methods and **FIVE** state-of-the-art graph prompt techniques, evaluated across **FIFTEEN** diverse datasets to assess performance, flexibility, and efficiency. We also present 'ProG', an easy-to-use open-source library that streamlines the execution of various graph prompt models, facilitating objective evaluations. Additionally, we propose a unified framework that categorizes existing graph prompt methods into two main approaches: prompts as graphs and prompts as tokens. This framework enhances the applicability and comparison of graph prompt techniques. The code is available at: https://github.com/sheldonresearch/ProG.

## 1 Introduction

Recently, artificial general intelligence (AGI) on graphs has emerged as a new trend in various applications like drug design [59, 38], protein prediction [10], social analysis [49, 64, 52], etc. To achieve this vision, one key question is how to learn useful knowledge from non-linear data (like graphs) and how to apply it to various downstream tasks or domains. Classical approaches mostly leverage the 'Pre-train & Fine-tune' paradigm, which first designs some pretext with easily access data as the pre-training task for the graph neural network model, and then adjusts partial or entire model parameters to fit new downstream tasks. Although much progress has been achieved, they are still not effective and efficient enough. For example, adjusting the pre-trained model will become very time-consuming with the increase in model complexity [25, 24, 48, 23]. In addition, a natural gap between these pretexts and downstream tasks makes the task transferring very hard, and sometimes may even cause negative transfer [57, 30, 29]. These problems are particularly serious in few-shot settings.

To further alleviate the problems above, Graph Prompt Learning [50, 28] has attracted more attention recently. As shown in Figure 1, graph prompts seek to manipulate downstream data by inserting an additional small prompt graph and then reformulating the downstream task to the pre-training task without changing the pre-trained Graph Neural Network (GNN) model. Since a graph prompt is usually lightweight, tuning this prompt is more efficient than the large backbone model. Compared

---

*The first two authors contributed equally to this work. Listing order is random.

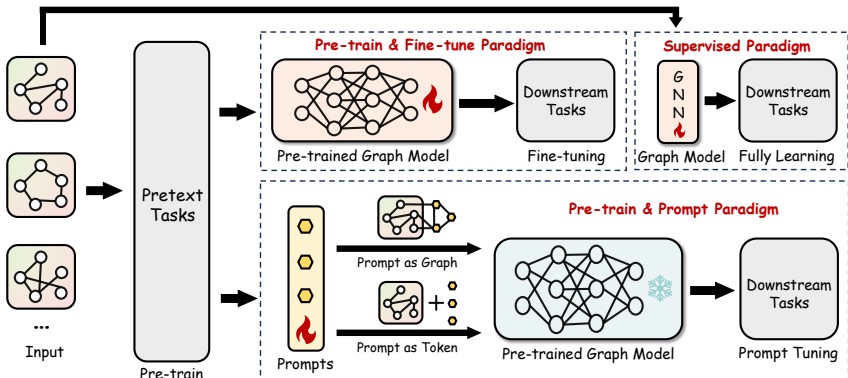

Figure 1: An overview of our benchmark.

with graph 'pre-training & fine-tuning', which can be seen as a model-level retraining skill, graph 'pre-training & prompting' is more customized for data-level operations, which means its potential applications might go beyond task transferring in the graph intelligence area like data quality evaluation, multi-domain alignment, learnable data augmentation, etc. Recently, many graph prompt models have been proposed and presented significant performance in graph learning areas [47, 46, 33, 8, 66, 5], indicating their huge potential towards more general graph intelligence. Despite the enthusiasm around graph prompts in research communities, three significant challenges impede further exploration:

❶ **How can we unify diverse graph prompt models given their varying methodologies?** The fragmented landscape of graph prompts hinders systematic research advancement. A unified framework is essential for integrating these diverse methods, creating a cohesive taxonomy to better understand current research and support future studies.

❷ **How do we evaluate the quality of graph prompts?** Assessing the efficiency, power, and flexibility of graph prompts is crucial for understanding their impact and limitations. Currently, there is no standardized benchmark for fair and comprehensive comparison, as existing works have inconsistent experimental setups, varying tasks, metrics, pre-training strategies, and data-splitting techniques, which obstruct a comprehensive understanding of the current state of research.

❸ **How can we make graph prompt approaches more user-friendly for practical comparison and selection?** Despite numerous proposed methods, the lack of an easy-to-use toolkit for creating graph prompts limits their potential applications. The implementation details of existing works vary significantly in programming frameworks, tricks, and running requirements. Developing a standardized library for various graph prompt learning approaches is urgently needed to facilitate broader exploration and application.

In this work, we wish to push forward graph prompt research to be more standardized, provide suggestions for practical choices, and uncover both their advantages and limitations with a comprehensive evaluation. To the best of our knowledge, this is the first benchmark for graph prompt learning. **To solve the first challenge**, our benchmark treats existing graph prompts as three basic components: prompt tokens, token structure, and insert patterns. According to their detailed implementation, we categorize the current graph prompts into two branches: graph prompt as an additional graph, and graph prompt as tokens. This taxonomy makes our benchmark approach compatible with nearly all existing graph prompting approaches. **To solve the second challenge**, our benchmark encompasses 6 different classical pre-training methods covering node-level, edge-level, and graph-level tasks, and 5 most representative and state-of-the-art graph prompt methods. Additionally, we include 15 diverse datasets in our benchmark with different scales, covering both homophilic and heterophilic graphs from various domains. We systematically investigate the effectiveness, flexibility, and efficiency of graph prompts under few-shot settings. Compared with traditional supervised methods and 'pre-training & fine-tuning' methods, graph prompting approaches present significant improvements. We even observe some negative transfer cases caused by 'pre-training & fine-tuning' methods but effectively reversed by graph prompting methods. **To solve the third challenge**, we develop a unified library for creating various graph prompts. In this way, users can evaluate their models or datasets with less effort. The contributions are summarized as follows:

❶ We propose the first comprehensive benchmark for graph prompt learning. The benchmark integrates 6 most used pre-training methods and 5 state-of-the-art graph prompting methods with

15 diverse graph datasets. We systematically evaluate the effectiveness, flexibility, and efficiency of current graph prompt models to assist future research efforts. (Detailed in Section 5)

❷ We offer 'ProG'[2], an easy-to-use open-source library to conduct various graph prompt models (Section 3). We can get rid of the distraction from different implementation skills and reveal the potential benefits/shortages of these graph prompts in a more objective and fair setting.

❸ We propose a unified view for current graph prompt methods (Section 2). We decompose graph prompts into three components: prompt tokens, token structure, and insert patterns. With this framework, we can easily group most of the existing work into two categories (prompt as graph, and prompt as tokens). This view serves as the infrastructure of our developed library and makes our benchmark applicable to most of the existing graph prompt works.

## 2    Preliminaries

**Understand Graph Prompt Nature in A Unified View.** Graph prompt learning aims to learn suitable transformation operations for graphs or graph representations to reformulate the downstream task to the pre-training task. Let $t(\cdot)$ be any graph-level transformation (e.g., "modifying node features", "augmenting original graphs", etc.), and $\psi^*(\cdot)$ be a frozen pre-trained graph model. For any graph $\mathcal{G}$ with adjacency matrix $\mathbf{A} \in \{0,1\}^{N \times N}$ and node feature matrix $\mathbf{X} \in \mathbb{R}^{N \times d}$ where $N$ denotes the node number and $d$ is feature number. It has been proven [47, 8] that we can always learn an appropriate prompt module $\mathcal{P}$, making them can be formulated as the following formula:

$$\psi^*(\mathcal{P}(\mathbf{X}, \mathbf{A}, \mathbf{P}, \mathbf{A}_{inner}, \mathbf{A}_{cross})) = \psi^*(t(\mathbf{X}, \mathbf{A})) + O_{\mathcal{P}\psi} \quad (1)$$

Here $\mathbf{P} \in \mathbb{R}^{K \times d}$ is the learnable representations of $K$ prompt tokens, $\mathbf{A}_{inner} \in \{0,1\}^{K \times K}$ indicate token structures and $\mathbf{A}_{cross} \in \{0,1\}^{K \times N}$ is the inserting patterns, which indicates how each prompt token connects to each node in the original graph. Prompt module $\mathcal{P}$ takes the input graph and then outputs an integrated graph with a graph prompt. This equation indicates that we can learn an appropriate prompt module $\mathcal{P}$ applied to the original graph to imitate any graph manipulation. $O_{\mathcal{P}\psi}$ denotes the *error bound* between the manipulated graph and the prompting graph w.r.t. their representations from the pre-trained graph model, which can be seen as a measurement of graph prompt flexibility.

**Task Definition (Few-shot Graph Prompt Learning).** Given a graph $\mathcal{G}$, a frozen pre-trained graph model $\psi^*$, and labels $\mathcal{Y}$, the loss function is to optimize the task loss $\mathcal{L}_{Task}$ as follows:

$$\mathcal{L}_{\mathcal{P}} = \mathcal{L}_{Task}(\psi^*(\mathcal{P}(\mathbf{X}, \mathbf{A}, \mathbf{P}, \mathbf{A}_{inner}, \mathbf{A}_{cross})), \mathcal{Y})$$

During the few-shot training phase, the labeled samples are very scarce. Specifically, given a set of classes $C$, each class $C_i$ has only $k$ labeled examples. Here, $k$ is a small number (e.g., $k \leq 10$), and the total number of labeled samples is given by $k \times |C| = |\mathcal{Y}|$, where $|C|$ is the number of classes. This setup is commonly referred to as $k$-shot classification [33]. In this paper, our evaluation includes both node and graph classification tasks.

**Evaluation Questions.** We are particular interested in the **effectiveness**, **flexibility**, and **efficiency** of graph prompt learning methods. Specifically, **in effectiveness**, we wish to figure out how effective are different graph prompt methods on various tasks (Section 5.1); and how well graph prompting methods overcome the negative transferring caused by pre-training strategies (Section 5.2). **In flexibility**, we wish to uncover how powerful different graph prompt methods are to simulate data operations (Section 5.3). **In efficiency**, we wish to know how efficient are these graph prompt methods in terms of time and space cost (Section 5.4).

## 3    Benchmark Methods

In general, graph prompt learning aims to transfer pre-trained knowledge with the expectation of achieving better positive transfer effects in downstream tasks compared to classical 'Pre-train & Fine-tune' methods. To conduct a comprehensive comparison, we first consider using a supervised method as the baseline for judging positive transfer (For example, if a 'pre-training and fine-tuning' approach can not beat supervised results, one negative transfer case is observed). We introduce various pre-training methods to generate pre-trained knowledge, upon which we perform fine-tuning

---

[2]Our code is available at https://github.com/sheldonresearch/ProG

as the comparison for graph prompts. Finally, we apply current popular graph prompt learning methods with various pre-training methods to investigate their knowledge transferability. The details of these baselines are as follows (Further introduction can be referred to Appendix A):

- **Supervised Method:** In this paper, we utilize Graph Convolutional Network (GCN) [21] as the baseline, which is one of the classical and effective graph models based on graph convolutional operations [50, 67, 32, 69] and also the backbone for both 'Pre-train & Fine-tune' and graph prompt learning methods. We also explore more kinds of graph models like GraphSAGE [13], GAT [54], and Graph Transformer [43], and put the results in Appendix E.

- **'Pre-train & Fine-tune' Methods:** We select 6 mostly used pre-training methods covering node-level, edge-level, and graph-level strategies. For **node-level**, we consider **DGI** [55] and **GraphMAE** [14], where DGI maximizes the mutual information between node and graph representations for informative embeddings and GraphMAE learns deep node representations by reconstructing masked features. For **edge-level**, we introduce **EdgePreGPPT** [46] and **EdgePreGprompt** [33], where EdgePreGPPT calculates the dot product as the link probability of node pairs and EdgePreGprompt samples triplets from label-free graphs to increase the similarity between the contextual subgraphs of linked pairs while decreasing the similarity of unlinked pairs. For **graph-level**, we involve **GCL** [65], **SimGRACE** [60], where GCL maximizes agreement between different graph augmentations to leverage structural information and SimGRACE tries to perturb the graph model parameter spaces and narrow down the gap between different perturbations for the same graph.

- **Graph Prompt Learning Methods:** With our proposed unified view, current popular graph prompt learning methods can be easily classified into two types, 'Prompt as graph' and 'Prompt as token' types. Specifically, the **'Prompt as graph'** type means a graph prompt has multiple prompt tokens with inner structure and insert patterns, and we introduce **All-in-one** [47], which aims to learn a set of insert patterns for original graphs via learnable graph prompt modules. For the **'Prompt as token'** type, we consider involving **GPPT** [46], **Gprompt** [33], **GPF** and **GPF-plus** [8]. Concretely, GPPT defines graph prompts as additional tokens that contain task tokens and structure tokens that align downstream node tasks and link prediction pretext. Gprompt focuses on inserting the prompt vector into the graph pooling by element-wise multiplication. GPF and GPF-plus mainly add soft prompts to all node features of the input graph.

**ProG: A Unified Library for Graph Prompts**. We offer our developed graph prompt library ProG as shown in Figure 2. In detail, the library first offers a **Model Backbone** module, including many widely-used graph models, which can be used for supervised learning independently. Then, it designs a **Pre-training** module involving many pre-training methods from various level types, which can pretrain the graph model initialized from the model backbone to preserve pre-trained knowledge and further fine-tune the pre-trained graph model. Finally, it seamlessly incorporates different graph prompt learning methods together into a unified **Prompting** module, which can freeze and execute prompt tuning for the pre-trained graph model from the pre-training module. For a fair comparison, it defines an **Evaluation** module, including comprehensive

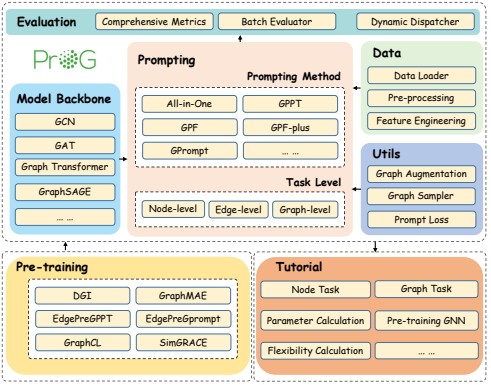

Figure 2: An overview of ProG

metrics, batch evaluator, and dynamic dispatcher, to ensure the same settings across different selected methods. Many essential operations are fused into the **Utils** module for repeated use, and all the data-related operations are included in the **Data** module for user-friendly operations to use the library. Additionally, the **Tutorial** module introduces the key functions that ProG offers.

## 4 Benchmark Settings

### 4.1 Datasets

To figure out how adaptive existing graph prompts on different graphs, we conduct our experiments across 15 datasets on node-level or graph-level tasks, providing a broad and rigorous testing ground

Table 1: Statistics of all datasets. N indicates node classification, and G indicates graph classification.

| Dataset | Graphs | Avg.nodes | Avg.edges | Features | Node classes | Task (N / G) | Category |
|---|---|---|---|---|---|---|---|
| Cora | 1 | 2,708 | 5,429 | 1,433 | 7 | N | Homophilic |
| Pubmed | 1 | 19,717 | 88,648 | 500 | 3 | N | Homophilic |
| CiteSeer | 1 | 3,327 | 9,104 | 3,703 | 6 | N | Homophilic |
| Actor | 1 | 7600 | 30019 | 932 | 5 | N | Heterophilic |
| Wisconsin | 1 | 251 | 515 | 1703 | 5 | N | Heterophilic |
| Texas | 1 | 183 | 325 | 1703 | 5 | N | Heterophilic |
| ogbn-arxiv | 1 | 169,343 | 1,166,243 | 128 | 40 | N | Homophilic & Large scale |

| Dataset | Graphs | Avg.nodes | Avg.edges | Features | Graph classes | Task (N / G) | Domain |
|---|---|---|---|---|---|---|---|
| MUTAG | 188 | 17.9 | 19.8 | 7 | 2 | G | Small Molecule |
| IMDB-BINARY | 1000 | 19.8 | 96.53 | 0 | 2 | G | Social Network |
| COLLAB | 5000 | 74.5 | 2457.8 | 0 | 3 | G | Social Network |
| PROTEINS | 1,113 | 39.1 | 72.8 | 3 | 2 | G | Proteins |
| ENZYMES | 600 | 32.6 | 62.1 | 3 | 6 | G | Proteins |
| DD | 1,178 | 284.1 | 715.7 | 89 | 2 | G | Proteins |
| COX2 | 467 | 41.2 | 43.5 | 3 | 2 | G | Small Molecule |
| BZR | 405 | 35.8 | 38.4 | 3 | 2 | G | Small Molecule |

for algorithmic comparisons. As shown in Table 1, we include 7 node classification datasets, covering homophilic datasets (Cora, Citeseer, PubMed, ogbn-arxiv) [42, 35, 15], heterophilic datasets (Wisconsin, Texas, Actor) [37, 53], and large-scale datasets (ogbn-arxiv). Additionally, we consider 8 graph classification datasets from different domains, including social networks (IMDB-B, COLLAB) [63], biological datasets (ENZYMES, PROTEINS, DD) [7, 2, 58], and small molecule datasets (MUTAG, COX2, BZR) [22, 41]. More details on these datasets can be found in Appendix C.

## 4.2 Implementation Details

**Data Split.** For the node task, we use 90% data as the test set. For the graph task, we use 80% data as the test set. To ensure the robustness of our findings, we repeat the sampling process five times to construct k-shot tasks for both node and graph tasks. We then report the average and standard deviation over these five results.

**Metrics.** Following existing multi-class classification benchmarks, we select Accuracy, Macro F1 Score, and AUROC (Area Under the Receiver Operating Characteristic Curve) as our main performance metrics [47, 68]. The Accuracy results are used to provide a straightforward indication of the model's overall performance in Section 5. F1 Score and AUROC results are provided in Appendix D.

**Prompt Adaptation to Specific Tasks.** We observe that certain prompt methods, such as GPPT, are only applicable to node classification, while GPF and GPF-plus are only suitable for graph classification. To this end, we design graph task tokens to replace the original node task tokens to make GPPT also applicable in graph-level tasks. Inspired by All-in-one and Gprompt, we utilize induced graphs in our benchmark to adapt GPF and GPF-plus for the node task. The concrete adaptation designs are shown in Appendix A.

**Hyperparameter Optimization.** To manage the influence of hyperparameter selection and maintain fairness, the evaluation process is standardized both with and without hyperparameter adjustments. Initially, default hyperparameters as specified in the original publications are used. To ensure fairness during hyperparameter tuning, **random search** is then employed to optimize the settings. In each trial across different datasets, a set of hyperparameters is randomly selected from a predefined search space for each model. For additional details on metrics, default hyperparameters, search spaces, and other implementation aspects, please see Appendix D.

## 5 Experimental Results

### 5.1 Overall Performance of Graph Prompt Learning

In Table 2 and Table 3, we present the optimal experimental results of the supervised methods, classical 'Pre-train & Fine-tune' methods, and various graph prompt methods on 1-shot node/graph

classification tasks across 15 datasets, where the evaluation metric is accuracy (Complete results can be referred to Appendix E). Here the 'optimal' results mean that the best result is shown from six classical 'Pre-train & Fine-tune' methods or from six pretraining-based variants of each graph prompt method. This presentation style can help reflect the upper bounds of classical 'Pre-train & Fine-tune' methods and graph prompt methods, better highlighting the advantages of graph prompts.

From these results, we can observe consistent advantages of various graph prompt learning methods. Almost all graph prompt methods consistently outperform the Supervised method on both node-level and graph-level tasks, demonstrating positive transfer and superior performance. Specifically, graph prompt methods generally surpass classical 'Pre-train & Fine-tune' methods across most datasets, showcasing enhanced knowledge transferability. In node-level tasks, GPF-plus, which focuses on modifying node features via learnable prompt features, achieves the best results on 4 out of 7 datasets with significant effects, attributed to its inherent ability to easily adapt to different pre-training methods [8]. In contrast, All-in-one, which concentrates on modifying graphs via learnable subgraphs, achieves the best results on 7 out of 8 datasets on graph-level tasks.

Table 2: Performance on 1-shot node classification. The best results for each dataset are highlighted in bold with a dark red background. The second-best are underlined with a light red background.

| Methods\Datasets | Cora | Citeseer | Pubmed | Wisconsin | Texas | Actor | Ogbn-arxiv |
|---|---|---|---|---|---|---|---|
| Supervised | $26.56_{\pm5.55}$ | $21.78_{\pm7.32}$ | $39.37_{\pm16.34}$ | $41.60_{\pm3.10}$ | $37.97_{\pm5.80}$ | $20.57_{\pm4.47}$ | $10.99_{\pm3.19}$ |
| Pre-train & Fine-tune | $52.61_{\pm1.73}$ | $35.05_{\pm4.37}$ | $46.74_{\pm14.89}$ | $40.69_{\pm4.13}$ | $46.88_{\pm4.69}$ | $20.74_{\pm4.12}$ | $16.21_{\pm3.82}$ |
| GPPT | $43.15_{\pm9.44}$ | $37.26_{\pm6.17}$ | $48.31_{\pm17.72}$ | $30.40_{\pm6.81}$ | $31.81_{\pm15.33}$ | $22.58_{\pm1.97}$ | $14.65_{\pm3.07}$ |
| All-in-one | $52.39_{\pm10.17}$ | $40.41_{\pm2.80}$ | $45.17_{\pm6.45}$ | $78.24_{\pm16.68}$ | $65.49_{\pm7.06}$ | $24.61_{\pm2.80}$ | $13.16_{\pm5.98}$ |
| Gprompt | $56.66_{\pm11.22}$ | $53.21_{\pm10.94}$ | $39.74_{\pm15.35}$ | $83.80_{\pm2.44}$ | $33.25_{\pm40.11}$ | $25.26_{\pm1.10}$ | $75.72_{\pm4.95}$ |
| GPF | $38.57_{\pm5.41}$ | $31.16_{\pm8.05}$ | $49.99_{\pm8.86}$ | $88.67_{\pm5.78}$ | $87.40_{\pm3.40}$ | $28.70_{\pm3.35}$ | $78.37_{\pm5.47}$ |
| GPF-plus | $55.77_{\pm10.30}$ | $59.67_{\pm11.87}$ | $46.64_{\pm18.97}$ | $91.03_{\pm4.11}$ | $95.83_{\pm4.19}$ | $29.32_{\pm8.56}$ | $71.98_{\pm12.23}$ |

Table 3: Performance on 1-shot graph classification. The best results for each dataset are highlighted in bold with a dark red background. The second-best are underlined with a light red background.

| Methods\Datasets | IMDB-B | COLLAB | PROTEINS | MUTAG | ENZYMES | COX2 | BZR | DD |
|---|---|---|---|---|---|---|---|---|
| Supervised | $57.30_{\pm0.98}$ | $47.23_{\pm0.61}$ | $56.36_{\pm7.97}$ | $65.20_{\pm6.70}$ | $20.58_{\pm2.00}$ | $27.08_{\pm1.95}$ | $25.80_{\pm6.53}$ | $55.33_{\pm6.22}$ |
| Pre-train & Fine-tune | $57.75_{\pm1.22}$ | $48.10_{\pm0.23}$ | $63.44_{\pm3.64}$ | $65.47_{\pm5.89}$ | $22.21_{\pm2.79}$ | $76.19_{\pm5.41}$ | $34.69_{\pm8.50}$ | $57.15_{\pm4.32}$ |
| GPPT | $50.15_{\pm0.75}$ | $47.18_{\pm5.93}$ | $60.92_{\pm2.47}$ | $60.40_{\pm15.43}$ | $21.29_{\pm3.79}$ | $78.23_{\pm1.38}$ | $59.32_{\pm11.22}$ | $57.69_{\pm6.89}$ |
| All-in-one | $60.07_{\pm4.81}$ | $51.66_{\pm0.26}$ | $66.49_{\pm6.26}$ | $79.87_{\pm5.34}$ | $23.96_{\pm1.45}$ | $76.14_{\pm5.51}$ | $79.20_{\pm1.65}$ | $59.72_{\pm1.52}$ |
| Gprompt | $54.75_{\pm12.43}$ | $48.25_{\pm13.64}$ | $59.17_{\pm11.26}$ | $73.60_{\pm4.76}$ | $22.29_{\pm3.50}$ | $54.64_{\pm9.94}$ | $55.43_{\pm13.69}$ | $57.81_{\pm2.68}$ |
| GPF | $59.65_{\pm5.06}$ | $47.42_{\pm11.22}$ | $63.91_{\pm3.26}$ | $68.40_{\pm5.09}$ | $22.00_{\pm1.25}$ | $65.79_{\pm17.72}$ | $71.67_{\pm14.71}$ | $59.36_{\pm1.18}$ |
| GPF-plus | $57.93_{\pm1.62}$ | $47.24_{\pm0.29}$ | $62.92_{\pm2.78}$ | $65.20_{\pm6.04}$ | $22.92_{\pm1.64}$ | $33.78_{\pm1.52}$ | $71.17_{\pm14.92}$ | $57.62_{\pm2.42}$ |

Similar observations can also be found in 3-shot and 5-shot results in the optimal and complete versions, which can be checked in Appendix E. Additionally, for comprehensive, we also offer the curve of the shot number from 1 to 10 and the accuracy in Appendix E.

## 5.2 Impact of Pre-training Methods

From Table 2, we further find that on the Wisconsin dataset, the classical Pre-train & Fine-tune method consistently exhibited negative transfer, whereas most graph prompt methods effectively transferred upstream pre-training knowledge, achieving significant results and positive transfer. Therefore, it is crucial to explore the effectiveness of various pre-training methods, especially for graph prompt learning. Table 4 and Table 5 show the experimental results of six variants of GPF-plus/All-in-one and the corresponding 'Pre-train & Fine-tune' methods. Concretely, we have the following key observations:

**Consistency in Pretext and Downstream Tasks.** In node classification tasks, the pre-trained knowledge learned by the node-level pre-training method (e.g., GraphMAE) shows superior transferability to downstream tasks using GPF-plus, achieving better results compared to traditional fine-tuning methods. Conversely, in graph classification tasks, the graph-level pre-training method (e.g., Sim-GRACE) demonstrates better transferability with All-in-one, resulting in more significant outcomes.

Table 4: 1-shot node classification performance comparison across different datasets of GPF-plus. ↑/↓ indicates the positive/negative transfer compared to the supervised baseline. ↓(%) means the negative transfer rate, computed by dividing the number of ↓ by the number of datasets.

| Methods\Datasets | Cora | Citeseer | Pubmed | Wisconsin | Texas | Actor | Ogbn-arxiv | ↓(%) |
|---|---|---|---|---|---|---|---|---|
| Supervised | $26.56_{\pm5.55}$ (-) | $21.78_{\pm7.32}$ (-) | $39.37_{\pm16.34}$ (-) | $41.60_{\pm3.10}$ (-) | $37.97_{\pm5.80}$ (-) | $20.57_{\pm4.47}$ (-) | $10.99_{\pm3.19}$ (-) | 0.0% |
| DGI | $33.15_{\pm7.84}$ (↑) | $21.64_{\pm3.92}$ (↓) | $42.01_{\pm12.54}$ (↑) | $37.49_{\pm7.56}$ (↓) | $45.31_{\pm5.01}$ (↑) | $19.76_{\pm3.53}$ (↓) | $7.21_{\pm2.91}$ (↓) | 57.0% |
| + GPF_plus | $17.29_{\pm6.18}$ (↓) | $\mathbf{26.60}_{\pm13.24}$ (↑) | $34.02_{\pm11.94}$ (↓) | $\mathbf{76.34}_{\pm6.26}$ (↑) | $79.54_{\pm1.89}$ (↑) | $22.42_{\pm9.66}$ (↑) | $16.83_{\pm10.02}$ (↑) | 29% |
| GraphMAE | $32.93_{\pm3.17}$ (↑) | $21.26_{\pm3.57}$ (↓) | $42.99_{\pm14.25}$ (↑) | $36.80_{\pm7.17}$ (↓) | $37.81_{\pm8.62}$ (↓) | $19.86_{\pm2.70}$ (↓) | $12.35_{\pm3.60}$ (↑) | 43.0% |
| + GPF_plus | $\mathbf{54.26}_{\pm7.48}$ (↑) | $\mathbf{59.67}_{\pm11.87}$ (↑) | $46.64_{\pm18.57}$ (↑) | $\mathbf{89.17}_{\pm9.79}$ (↑) | $\mathbf{94.09}_{\pm4.40}$ (↑) | $\mathbf{26.58}_{\pm7.84}$ (↑) | $\mathbf{49.81}_{\pm2.62}$ (↑) | 0.0% |
| EdgePreGPPT | $38.12_{\pm5.29}$ (↑) | $18.09_{\pm5.39}$ (↓) | $46.74_{\pm14.09}$ (↑) | $35.31_{\pm9.31}$ (↓) | $47.66_{\pm2.37}$ (↑) | $19.17_{\pm2.53}$ (↓) | $16.21_{\pm3.82}$ (↑) | 43.0% |
| + GPF_plus | $28.49_{\pm18.73}$ (↓) | $28.04_{\pm14.31}$ (↑) | $46.51_{\pm15.84}$ (↑) | $\mathbf{91.03}_{\pm4.11}$ (↑) | $92.14_{\pm5.46}$ (↑) | $29.32_{\pm8.56}$ (↑) | $\mathbf{71.98}_{\pm12.23}$ (↑) | 14.0% |
| EdgePreGprompt | $35.57_{\pm5.83}$ (↑) | $22.28_{\pm3.80}$ (↑) | $41.50_{\pm7.54}$ (↑) | $40.69_{\pm4.13}$ (↓) | $40.62_{\pm7.95}$ (↑) | $20.74_{\pm4.16}$ (↑) | $14.83_{\pm2.38}$ (↑) | 14.0% |
| + GPF_plus | $\mathbf{55.77}_{\pm10.30}$ (↑) | $49.43_{\pm8.21}$ (↑) | $42.79_{\pm18.18}$ (↑) | $85.44_{\pm2.23}$ (↑) | $87.70_{\pm3.77}$ (↑) | $22.68_{\pm3.64}$ (↑) | $57.44_{\pm6.95}$ (↑) | 0.0% |
| GCL | $52.61_{\pm1.73}$ (↑) | $27.02_{\pm4.31}$ (↑) | $42.49_{\pm11.29}$ (↑) | $33.94_{\pm7.74}$ (↓) | $40.31_{\pm13.68}$ (↑) | $20.19_{\pm1.98}$ (↓) | $4.65_{\pm1.19}$ (↓) | 43.0% |
| + GPF_plus | $34.18_{\pm17.71}$ (↓) | $28.86_{\pm22.88}$ (↑) | $37.02_{\pm11.29}$ (↓) | $68.57_{\pm8.77}$ (↑) | $\mathbf{95.83}_{\pm4.19}$ (↑) | $22.82_{\pm4.99}$ (↑) | $32.11_{\pm4.86}$ (↑) | 29.0% |
| SimGRACE | $40.40_{\pm4.66}$ (↑) | $35.05_{\pm4.37}$ (↑) | $37.59_{\pm8.17}$ (↓) | $37.37_{\pm3.68}$ (↓) | $46.88_{\pm4.64}$ (↑) | $19.78_{\pm1.89}$ (↓) | $8.13_{\pm3.26}$ (↓) | 57.0% |
| + GPF_plus | $21.33_{\pm14.86}$ (↓) | $24.61_{\pm21.21}$ (↓) | $35.90_{\pm9.06}$ (↓) | $73.64_{\pm5.25}$ (↑) | $90.74_{\pm3.12}$ (↑) | $20.51_{\pm4.24}$ (↑) | $46.71_{\pm3.17}$ (↑) | 43.0% |

Table 5: 1-shot graph classification performance comparison across different datasets of All-in-one.

| Methods\Datasets | IMDB-B | COLLAB | PROTEINS | MUTAG | ENZYMES | COX2 | BZR | DD | ↓(%) |
|---|---|---|---|---|---|---|---|---|---|
| Supervised | $57.30_{\pm0.98}$ (-) | $47.23_{\pm0.61}$ (-) | $56.36_{\pm7.97}$ (-) | $65.20_{\pm6.70}$ (-) | $20.58_{\pm2.00}$ (-) | $27.08_{\pm11.94}$ (-) | $25.80_{\pm6.53}$ (-) | $55.33_{\pm6.22}$ (-) | 0.0% |
| DGI | $57.32_{\pm0.90}$ (↑) | $42.22_{\pm0.73}$ (↓) | $64.65_{\pm2.10}$ (↑) | $64.13_{\pm7.90}$ (↓) | $17.83_{\pm1.88}$ (↓) | $29.44_{\pm9.68}$ (↑) | $26.48_{\pm7.61}$ (↑) | $57.15_{\pm4.32}$ (↑) | 38.0% |
| + All-in-one | $\mathbf{60.07}_{\pm4.81}$ (↑) | $39.56_{\pm5.00}$ (↓) | $62.58_{\pm7.07}$ (↑) | $75.73_{\pm7.75}$ (↑) | $23.96_{\pm1.45}$ (↑) | $50.72_{\pm9.93}$ (↑) | $79.20_{\pm1.65}$ (↑) | $55.97_{\pm6.52}$ (↑) | 13.0% |
| GraphMAE | $57.70_{\pm1.13}$ (↑) | $48.10_{\pm0.23}$ (↑) | $63.57_{\pm3.57}$ (↑) | $65.20_{\pm5.00}$ (-) | $22.21_{\pm2.79}$ (↑) | $28.47_{\pm14.72}$ (↑) | $25.80_{\pm6.53}$ (-) | $57.54_{\pm4.41}$ (↑) | 0.0% |
| + All-in-one | $52.62_{\pm3.04}$ (↓) | $40.82_{\pm14.63}$ (↓) | $66.49_{\pm6.26}$ (↑) | $72.27_{\pm8.87}$ (↑) | $23.21_{\pm1.72}$ (↑) | $56.68_{\pm7.38}$ (↑) | $75.37_{\pm5.99}$ (↑) | $\mathbf{58.77}_{\pm1.05}$ (↑) | 25.0% |
| EdgePreGPPT | $57.20_{\pm0.85}$ (↓) | $47.14_{\pm0.55}$ (↓) | $58.27_{\pm10.66}$ (↑) | $64.27_{\pm4.73}$ (↓) | $19.79_{\pm2.17}$ (↓) | $27.83_{\pm13.44}$ (↑) | $72.10_{\pm14.30}$ (↑) | $52.82_{\pm9.38}$ (↓) | 63.0% |
| + All-in-one | $59.12_{\pm0.77}$ (↑) | $42.74_{\pm4.65}$ (↓) | $65.71_{\pm5.49}$ (↑) | $\mathbf{79.87}_{\pm5.34}$ (↑) | $20.92_{\pm2.04}$ (↑) | $60.27_{\pm16.97}$ (↑) | $64.20_{\pm19.74}$ (↑) | $56.24_{\pm2.46}$ (↑) | 13.0% |
| EdgePreGprompt | $57.35_{\pm0.92}$ (↑) | $47.20_{\pm0.53}$ (↓) | $61.84_{\pm2.59}$ (↑) | $62.67_{\pm2.67}$ (↓) | $19.75_{\pm2.33}$ (↓) | $27.13_{\pm12.05}$ (↑) | $29.44_{\pm11.20}$ (↑) | $56.16_{\pm5.10}$ (↑) | 38.0% |
| + All-in-one | $53.78_{\pm2.82}$ (↓) | $42.87_{\pm6.19}$ (↓) | $61.82_{\pm7.53}$ (↑) | $75.47_{\pm5.00}$ (↑) | $21.88_{\pm0.56}$ (↑) | $49.06_{\pm5.53}$ (↑) | $66.60_{\pm17.36}$ (↑) | $57.60_{\pm4.37}$ (↑) | 25.0% |
| GCL | $57.75_{\pm1.02}$ (↑) | $39.62_{\pm0.63}$ (↓) | $63.44_{\pm3.64}$ (↑) | $65.07_{\pm8.38}$ (↓) | $23.96_{\pm1.99}$ (↑) | $53.14_{\pm21.32}$ (↑) | $29.07_{\pm7.00}$ (↑) | $60.62_{\pm1.56}$ (↑) | 25.0% |
| + All-in-one | $58.75_{\pm0.80}$ (↑) | $\mathbf{51.66}_{\pm2.60}$ (↑) | $64.36_{\pm7.30}$ (↑) | $70.53_{\pm8.58}$ (↑) | $19.46_{\pm2.85}$ (↓) | $52.55_{\pm13.51}$ (↑) | $50.93_{\pm16.83}$ (↑) | $59.72_{\pm1.52}$ (↑) | 13.0% |
| SimGRACE | $57.33_{\pm0.96}$ (↑) | $46.89_{\pm0.42}$ (↓) | $60.07_{\pm3.21}$ (↑) | $65.47_{\pm5.89}$ (↑) | $19.71_{\pm1.76}$ (↓) | $76.19_{\pm5.41}$ (↑) | $28.48_{\pm6.49}$ (↑) | $53.23_{\pm9.71}$ (↓) | 38.0% |
| + All-in-one | $\mathbf{58.83}_{\pm0.85}$ (↑) | $47.60_{\pm3.90}$ (↑) | $61.17_{\pm1.73}$ (↑) | $65.47_{\pm6.91}$ (↑) | $22.50_{\pm1.56}$ (↑) | $76.14_{\pm5.51}$ (↑) | $69.88_{\pm18.06}$ (↑) | $58.26_{\pm1.18}$ (↑) | 0.0% |

These observations indicate that although graph prompts may perform better, they are also selective to specific pre-training tasks. In general, node-level pre-training is more suitable for node-level tasks, while graph-level pre-training is more effective for graph-level tasks.

**Mitigation of Negative Transfer.** Classical pre-trained models often experience **significant negative transfer issues** during fine-tuning. In contrast, **prompt tuning can effectively handle these problems**. For example, in node classification, GPF-plus can reduce the negative transfer rate from 43% to 0%. Similarly, in graph classification, All-in-one successfully resolves negative transfer issues, reducing the rate from 38% to 0%. This demonstrates the efficacy of graph prompt methods, such as GPF-plus and All-in-one, in mitigating negative transfer, ensuring more reliable and accurate knowledge transfer in both node-level and graph-level tasks.

Figure 3a and Figure 3b respectively show the heat maps of Table 4 and Table 5, respectively, for help better understanding the impact of pre-training methods (Other heat maps of other graph prompt learning methods can be referred to Appendix E).

## 5.3 Flexibility Analysis

We leverage *error bound*, defined by Equation 1, to measure the flexibility of graph prompt methods [47]. The *error bound* quantifies the difference in graph representations between the original

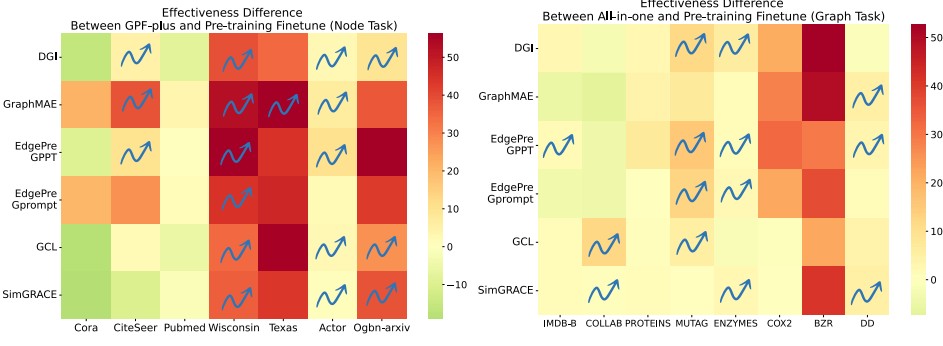

| (a) GPF-plus (1-shot node classification Task). | (b) All-in-one (1-shot graph classification Task). |

Figure 3: Head map of GPF-plus and All-in-one on node-level and graph-level tasks across various pre-training methods and datasets. The green interval shows that the graph prompt learning method performs worse than 'Pre-train & Fine-tune'. The yellow-to-red interval shows that the graph prompt learning method performs better. **An upward arrow** indicates that the 'Pre-train & Fine-tune' method experiences **negative transfer**, while the graph **prompt** learning method can **alleviate it significantly**.

graph and the manually manipulated graph restored by graph prompt methods. These manipulations include dropping nodes, dropping edges, and masking features. Table 6 presents the specific results. The experimental results indicate that **all graph prompt methods significantly reduced the error compared to the original error across all types of manipulations**. Notably, All-in-one and Gprompt achieved the lowest error bounds, demonstrating their flexibility in adapting to graph transformations. Specifically, All-in-one achieved a RED% of 95.06,

Table 6: Error bound between different prompt methods, RED(%): average reduction of each method to the original error.

| Prompt Model | Drop Nodes | Drop Edges | Mask Feature | RED (%) |
|---|---|---|---|---|
| ori_error | 0.4862 | 0.0713 | 0.6186 | |
| All-in-one | 0.0200 | 0.0173 | 0.0207 | 95.06 ↓ |
| Gprompt | 0.0218 | 0.0141 | 0.0243 | 94.88 ↓ |
| GPF-plus | 0.0748 | 0.0167 | 0.1690 | 77.85 ↓ |
| GPF | 0.0789 | 0.0146 | 0.1858 | 76.25 ↓ |

indicating a 95.06% reduction in original error, underscoring its comprehensive approach to capturing graph transformations. These findings highlight the importance of flexibility in prompt design for effective knowledge transfer. Flexible prompts that can adapt to various graph transformations enable more powerful graph data operation capability, thereby improving the performance of downstream tasks. Moreover, this flexibility is inspiring for more customized graph data augmentation, indicating more potential for other applications.

## 5.4 Efficiency Analysis

We evaluate the efficiency and training duration of various graph prompt methods across tasks, focusing on node classification in Cora and Wisconsin and graph classification in BZR and DD. **GPF-plus** achieves **high accuracy** with **relatively low training times** despite its **larger parameter size** on Cora and Wisconsin. In contrast, **All-in-One**, with **a smaller parameter size**, has **slightly larger training times** than GPF-plus due to similarity calculations. Among all the graph prompt methods, GPPT shows lower performance and the longest training duration because of its k-means clustering update in every epoch. Besides, we **compare the tunable parameters of graph prompt methods** during the training phase with **classical 'Pre-train & Fine-tune' methods** with various backbones for further analyzing the advantages of graph prompt learning. Consider an input graph has $N$ nodes, $M$ edges, $d$ features, along with a graph model which has $L$ layers with maximum layer dimension $D$. **Classical backbones (e.g., GCN) have** $O(dD + D^2(L-1))$ learnable parameter complexity. Other more complex backbones (e.g., graph transformer) may have larger parameters. In the **graph prompt learning framework**, tuning the prompt parameters with a frozen pre-trained backbone makes training converge faster. For **All-in-One**, with $K$ tokens and $m$ edges, **learnable parameter complexity is** $O(Kd)$. Other graph prompt methods also have similar tunable parameter advantages, with specific sizes depending on their designs (e.g., Gprompt only includes a learnable vector inserted into the graph pooling by element-wise multiplication).

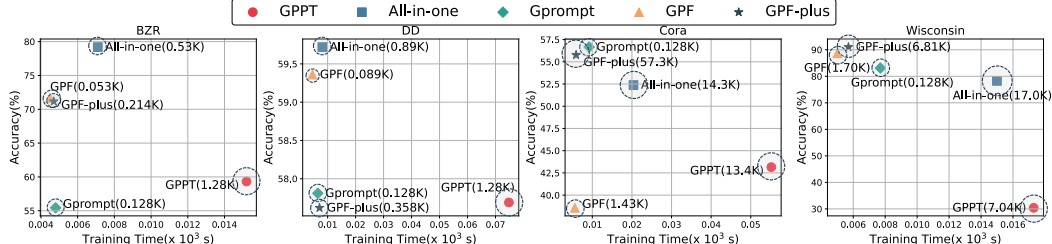

Figure 4: Analysis of accuracy, training time and tunable parameters of various graph prompt methods. Note that the gray area enclosed by the dashed line represents the scale of the tunable parameters.

## 5.5 Adaptability of Graph Prompts on Different Graphs.

Besides the main observations mentioned before, we here further discuss the adaptability of graph prompts on various kinds of graphs. Specifically, considering graph prompts performance on **various domains** as shown in Table 3, we can find that the adaptability to various domains of graph prompt methods, particularly those of the Prompt as Token type, is not always good, where graph prompt methods even may encounter negative transfer issues. GPPT, in particular, faces significant challenges, likely because first-order subgraphs fail to comprehensively capture graph patterns, introducing noise and affecting transferability. For **Large-scale Datasets** like node-level dataset ogbn-arxiv in Table 2, the transferability of graph prompt methods involving subgraph operations, such as GPPT and All-in-one, requires further enhancement. Furthermore, regarding the relatively large graph-level dataset DD in Table 3, all graph prompt methods achieve positive transfer and outperform Pre-train & Fine-tune. Despite this success, the magnitude of improvement indicates that the transferability on graph-level large-scale datasets also requires further enhancement. For **Homophilic & Heterophilic Datasets** in Table 2, most graph prompt methods can handle both homophilic and heterophilic graph datasets effectively, with the exception of GPPT, possibly due to instability caused by the clustering algorithms of GPPT. For **Pure Structural Dataset** like IMDB-BINARY and COLLAB in Table 3, the ability to handle pure structural datasets of graph prompt methods needs further refinement. In detail, some 'Pre-train & Fine-tune' methods can achieve positive transfer on these datasets, while some variants of the All-in-one method even exhibit negative transfer. Further detailed analysis can be referred to Appendix E.

# 6 Conclusion and Future Plan

In this paper, we introduce ProG, a comprehensive benchmark for comparing various graph prompt learning methods with supervised and 'Pre-train & Fine-tune' methods. Our evaluation includes 6 Pre-training methods and 5 graph prompt learning methods across 15 real-world datasets, demonstrating that graph prompt methods generally outperform classical methods. This performance is attributed to the high flexibility and low complexity of graph prompt methods. Our findings highlight the promising future of graph prompt learning research and its challenges. By making ProG open-source, we aim to promote further research and better evaluations of graph prompt learning methods.

As the empirical analysis in this paper, graph prompt learning shows its significant potential beyond task transferring, though our current evaluation focuses on node/graph-level tasks. We view ProG as a long-term project and are committed to its continuous development. Our future plans include expanding to a wider range of graph tasks, adding new datasets, etc. Ultimately, we aim to develop ProG into a robust graph prompt toolbox with advanced features like automated prompt selection.

## Acknowledgments and Disclosure of Funding

This work was supported by National Key Research and Development Program of China Grant No. 2023YFF0725100 and Guangzhou-HKUST(GZ) Joint Funding Scheme 2023A03J0673. The research of Cheng was supported in part by project #MMT-p2-23 of the Shun Hing Institute of Advanced Engineering, The Chinese University of Hong Kong and by grants from the Research Grant Council of the Hong Kong Special Administrative Region, China (No. CUHK 14217622).

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

# A  Detailed Description of Methods

**End-to-end Grpah Models**

- **GCN (Graph Convolutional Network [21]):** A method that utilizes convolution operation on the graph to propagate information from a node to its neighboring nodes, enabling the network to learn a representation for each node based on its local neighborhood.
- **GraphSAGE (Graph Sample and AggregatE [13]:)** A general inductive learning framework that generates node embeddings by sampling and aggregating features from a node's local neighborhood.
- **GAT (Graph Attention Networks [54]):** A graph neural network (GNN) framework that incorporates the attention mechanism. It assigns varying levels of importance to different nodes during the neighborhood information aggregation process, allowing the model to focus on the most informative parts.
- **GT (Graph Transformer [43]):** An adaptation of the neural network architecture that applies the principles of the Transformer model to graph-structured data. It uses masks in the self-attention process to leverage the graph structure and enhance model efficiency.

**Pre-training**

- **DGI[55]:** capitalizes on a self-supervised method for pretraining, which is based on the concept of mutual information (MI). It maximizes the MI between the local augmented instances and the global representation.
- **GraphMAE[14]:** employs the masked autoencoder framework to minimize the difference between the original features of the masked nodes and their reconstructed features. This process enhances the model's ability to understand and embed graph structures effectively.
- **EdgePreGPPT[46]:** is proposed in GPPT to calculate the dot product as the link probability of the node pair.
- **EdgePreGprompt[33]:** is proposed in Gprompt n, for a set of label-free graphs G, we sample several triplets from each graph to increase the similarity between the contextual subgraphs of linked pair while decreasing unlinked pair.
- **GraphCL[65]:** applies different graph augmentations to exploit the structural information on the graphs and aims to maximize the agreement between different augmentations for graph pre-training.
- **SimGRACE[60]:** tries to perturb the graph model parameter spaces and narrow down the gap between different perturbations for the same graph.

**Graph Prompt Methods**

- **GPPT ('Prompt as token') [46]:** introduces a graph prompting function that transforms a standalone node into a token pair composed of a task token, representing a node label, and a structure token, describing the given node. This reformulates the downstream node classification task to resemble edge prediction. As a result, the pre-trained GNNs can be directly applied without extensive fine-tuning to evaluate the linking probability of the token pair and produce the node classification decision. Since GPPT originally is designed for node classification, in this work, to adapt to graph classification, we follow the core thoughts of GPPT and make the following modifications: (1) Introduce $m$ graph task tokens, one for each graph label. (2) Extract $N$ structure tokens from a graph with $N$ nodes, the same as the GPPT for node tasks. (3) Combine each structure token with all graph tokens to form $m$ token pairs. (4) Calculate an $m$-dimension link prediction probability vector for the $m$ structure tokens and choose the class with maximum probability as the prediction result. (6) Aggregate the link prediction results using a voting strategy to determine the graph class. (7) Update the learnable tokens based on the prediction loss.
- **Gprompt ('Prompt as token') [33]:** employs subgraph similarity as a mechanism to unify different pretext and downstream tasks, including link prediction, node classification, and graph classification. A learnable prompt is subsequently tuned for each downstream task.
- **All-in-one ('Prompt as graph') [47]:** involves freezing the parameters of pre-trained graph models and incorporating learnable graph prompts as insert patterns to original induced graphs on the downstream node classification task.
- **GPF ('Prompt as token') [8]:** applies a consistent prompt feature to all nodes, regardless of their contextual differences.

- **GPF-plus ('Prompt as token') [8]:** enhances the method by assigning prompt features based on node type, ensuring uniformity within each category but without considering the context of individual graphs. Note that GPF and GPF-plus inherently are primarily evaluated in graph classification tasks and are not readily adaptable to node classification. In this work, we address this limitation by inducing a subgraph for each node within a node-level graph. This approach effectively reformulates the node classification task into a graph classification task, allowing for the integration of inserted prompt features.

## B    Related Work

### B.1    Graph pre-training

Pre-training of graphs is a crucial step in graph representation learning for pre-training, prompting, and predicting paradigms, which provides a solid foundation for downstream tasks by utilizing existing information to encode graph structures. it aims to learn some general knowledge for the graph model with easily accessible information to reduce the annotation costs of new tasks. Node-level pre-training obtains local structural representations, which can be used for downstream tasks. Comparative learning learns by maximizing the mutual information(MI) between the original view and the self-supervised view [6, 70, 19] The prediction model reconstructs perturbed data [13, 14]. However, predictive models are difficult to capture higher-order information. Edge-level pre-training includes distinguishing whether there are edges between node pairs [20, 34], or reconstructing masked edges by restoring including graph reconstruction [39, 61] and contrastive learning targets either local patches of node features and global graph features [55, 45]. Multi-task pre-training addresses multiple optimization objectives, covering various graph-related aspects. This approach enhances generalization and mitigates negative transfer issues. For instance, Hu et al. [16] pre-trained a GNN at both the node and graph levels, allowing the GNN to learn valuable local and global representations simultaneously. Additionally, some studies have utilized contrastive learning at different levels [18, 62], or jointly optimized both contrastive loss and graph reconstruction error [26].

### B.2    Graph prompt learning

While Pre-training effectively encodes global information and generates valuable graph-level representations, a significant challenge lies in transferring knowledge from a specific pretext task to downstream tasks with substantial gaps, potentially resulting in negative transfer [40]. Instead of fine-tuning a pre-trained model with an adaptive task head, prompt learning aims to reformulate input data to fit the pretext. Many effective prompt methods were initially developed in the field of NLP, encompassing various approaches such as hand-crafted prompts exemplified by GPT-4 [1], discrete prompts like AutoPrompt [27], and trainable prompts in continuous spaces such as Prefix-Tuning [44]. Due to their remarkable success, prompt-based methods have increasingly been applied to the graph domain [4].

Graph prompt learning usually has two formalities: on the one hand, it can be regarded as tokens [3, 51, 12]. For example, GPF takes a prompt token as some additional features are added to the original graph. Based on that, GPF-plus [8] trains several independent basis vectors and utilizes attentive aggregation of these basis vectors with the assistance of several learnable linear projections. It also solved problems when training graphs have different scales and large-scale input graphs. GPPT [46] defines graph prompts as additional tokens that contain task tokens and structure tokens that align downstream node task and link prediction pretext. Furthermore, some works [71] use graph prompt to assist graph pooling operation (a.k.a Readout), like Gprompt [33] inserts the prompt vector to the graph pooling by element-wise multiplication. On the other hand, graph prompt is regarded as graphs [11, 17], Such as All-in-one [47] inserts the token graphs as prompt to the original graph and all links between tokens and original graphs, these links can be pre-calculated by the dot product between one token to another token (inner links) or one token to another original node (cross links).

## C    Detailed Description of Datasets

**Homophilic:** Cora and Citeseer datasets [42] consist of a diverse collection of computer science publications, where each node is characterized by a bag-of-words representation of papers and a

categorical label indicating the paper topic. Pubmed dataset [35] comprises articles related to diabetes from the PubMed database. Each node in this dataset is represented by an attribute vector containing TF/IDF-weighted word frequencies, accompanied by a label specifying the particular type of diabetes discussed in the publication. ogbn-arxiv [15] is a large-scale directed graph representing the citation network between all Computer Science (CS) arXiv papers indexed by MAG [56]. Each node is an arXiv paper and each directed edge indicates that one paper cites another one. Each paper comes with a 128-dimensional feature vector obtained by averaging the embeddings of words in its title and abstract.

**Heterophilic:** Cornell, Texas, and Wisconsin are three subdatasets derived from the WebKB dataset [37], compiled from multiple universities' web pages. Each node within these datasets represents a web page, with edges denoting hyperlinks between pages. The node features are represented as bag-of-words representations of the web pages. Additionally, the web pages are manually categorized into five distinct labels: student, project, course, staff, and faculty. Actor is a co-occurrence network in which each node corresponds to an actor and edges indicate their co-occurrence on the same Wikipedia page [53, 31].

**Social networks:** IMDB-BINARY [63] is a movie collaboration dataset. The nodes represent actors/actresses, the edges represent collaborations between actors/actresses appearing in the same movie, and the graph labels indicate the genre (Action or Romance). COLLAB [63] is a scientific collaboration dataset. The nodes represent researchers, the edges represent the collaboration relationships between researchers, and the graph labels indicate the field of research (High Energy Physics, Condensed Matter Physics, or Astro Physics).

**Biological:** PROTEINS [58] is a collection of protein graphs that include the amino acid sequence, conformation, structure, and features such as active sites of the proteins. The nodes represent the secondary structures, and each edge depicts the neighboring relation in the amino-acid sequence or in 3D space. The nodes belong to three categories, and the graphs belong to two classes. ENZYMES [2] is a dataset of 600 enzymes collected from the BRENDA enzyme database. These enzymes are labeled into 6 categories according to their top-level EC enzyme. D&D [7] consists of graph representations of 1,178 proteins. In each graph, nodes represent amino acids, and there is an edge if they are less than six Angstroms apart. Graphs are labeled according to whether they are enzymes or not.

**Small Molecule:** COX2 [41] is a dataset of molecular structures including 467 cyclooxygenase-2 inhibitors, in which each node is an atom, and each edge represents the chemical bond between atoms, such as single, double, triple or aromatic. All the molecules belong to two categories. BZR [41] is a collection of 405 ligands for benzodiazepine receptors, in which each ligand is represented by a graph. All these ligands belong to 2 categories. MUTAG [22] is a dataset of 188 mutagenic aromatic and heteroaromatic nitro compounds with 7 discrete labels.

# D    Additional Details

**Metrics.** In node and graph classification, **AUROC** and **F1-score** are another two important evaluation metrics. AUROC stands for *Area Under the Receiver Operating Characteristic Curve*. It shows how well the model can distinguish between different classes. A score of 1 means perfect classification, while 0.5 indicates the model is guessing randomly. F1-score combines precision and recall into a single number. Precision measures how many of the predicted positives are actually correct, and recall measures how many of the actual positives the model correctly identified. The F1-score ranges from 0 to 1, with 1 being the best. It is particularly useful in cases where class distributions are imbalanced. In this work, we compute the macro average values for each class. For both AUROC and F1-score, in multi-class classification tasks, each class is treated as the positive class once, while the other classes are treated as negative, and the metrics are averaged.

**Implementation Details.** To ensure a comprehensive evaluation and maintain fairness across a broad spectrum of models, we develop an open-source package named ProG[3].

**Software and Hardware.** For a fair comparison, all methods are implemented by our developed library ProG, which is built upon PyTorch [36], and unless specifically indicated, the encoders for all methods are Graph Convolutional Networks. All experiments are conducted on a Linux server with

---

[3]https://github.com/sheldonresearch/ProG

GPU (NVIDIA GeForce 3090, NVIDIA GeForce 4090, and NVIDIA A800) and CPU (AMD EPYC 7763), using PyTorch 2.0.1 [36], PyG 2.3.1 [9] and Python 3.9.17.

**Hyperparameter Settings.** To ensure a comprehensive evaluation process and maintain fairness, we conduct an in-depth analysis of various graph prompt methods. Then, we carefully design a unified random search strategy utilized for selecting a set of optimized hyperparameters for each prompt method on each dataset, going beyond the default settings. Given the significant variations in hyperparameters across different prompt methods, we focus on optimizing three types of general hyperparameters via random search. Concretely, we search for the optimal learning rate within the range $10^{uniform(-3,-1)}$, and optimal weight decay within $10^{uniform(-5,-6)}$. Additionally, 32, 64, or 128 batch sizes are selected with equal probability in each trial. Moreover, supervised methods and 'Pre-train & Fine-tune' methods in this work also undergo the same random search process for hyperparameter optimization.

# E  Additional Experimental Analysis

## E.1  Adaptivity of Graph Prompts on Different Graphs.

**Various Domains.** GPL methods, particularly those of the Prompt as Token type, exhibit *performance inconsistencies across graph-level datasets from various domains*, w.r.t Table 3. Sometimes, these methods *may encounter negative transfer issues* to varying degrees, except for the BZR and DD datasets. GPPT, in particular, faces significant challenges, likely because first-order subgraphs fail to comprehensively capture graph patterns, introducing noise and affecting transferability.

**Large-scale Datasets.** Referring to Table 2, on the large-scale node-level dataset (i.e., ogbn-arxiv), Gprompt, GPF, and GPF-plus significantly surpass the Supervised and classical 'Pre-train & Fine-tune' methods, demonstrating good scalability. However, the scalability of GPL methods involving subgraph operations, such as GPPT and All-in-one, needs improvement as they fail to fully capture large-scale graph patterns with their subgraph information. Furthermore, regarding the results of the relatively large graph-level dataset (i.e., DD) in Table 3, all GPL methods achieve positive transfer and outperform classical 'Pre-train & Fine-tune' methods. Despite this success, the magnitude of improvement indicates that *the transferability of GPL methods on large-scale datasets requires further enhancement.*

**Homophilic & Heterophilic Datasets.** Drawing from Table 2, most GPL methods can handle both homophilic and heterophilic graph datasets effectively, with the exception of GPPT. GPPT outperforms the Supervised method on homophilic graph datasets but falls short compared to classical 'Pre-train & Fine-tune' methods. On heterophilic datasets, where node attributes differ significantly, GPPT experiences negative transfer issues, possibly due to instability caused by its clustering algorithms.

**Pure Structural Dataset.** Observing Table 3, the prompting capability for purely structural graph datasets, such as IMDB-B and COLLAB, still requires improvement. While classical 'Pre-train & Fine-tune' methods achieve positive transfer on these datasets, some variants of the All-in-one method exhibit negative transfer, highlighting *the need for further refinement in handling purely structural graphs.*

## E.2  Additional Experimental Results with Different Shots and Metrics

The optimal performance of the graph prompt method, along with other methods, under 3-shot and 5-shot learning settings, is displayed in Tables 7- 10. Building on previous experimental outcomes, it is evident that the graph prompt method significantly boosts the effectiveness of various pre-trained graph models in 1/3/5-shot learning settings, thereby enhancing the transferability of pre-trained knowledge in the original 'Pre-train & Fine-tune' method. Specifically, GPF-plus and All-in-one approaches consistently prove to be the most compatible with different pre-trained methods for node and graph tasks, respectively.

Moreover, the experimental results of various methods under 3/5-shot learning settings for three key metrics—Accuracy, F1-score, and AUROC—are comprehensively presented in Tables 11- 28.

## E.3 Curve of The Shot Number and Accuracy

Figure 5 shows the performance changes of different graph prompt methods with increasing shot numbers on node-level dataset Cora and graph-level dataset ENZYMES. The rest baselines are omitted due to their unsatisfactory performance and unstable training process. Specifically, we observe that the performance of various graph prompt methods generally decreases with the increase in shot numbers and may even be surpassed by Supervised methods. This experimental result aligns with the expectation that graph prompt methods are suited for few-shot learning settings because as the number of samples increases and the sample distribution becomes more comprehensive, the effectiveness of graph prompt methods gradually diminishes.

## E.4 Performance with More Kinds of Graph Models.

Table 29 and Table 30 show the performance of various graph prompt methods based on different graph models on the Wisconsin and PROTEINS datasets. Overall, compared to the 'Pre-train & Fine-tune' method, graph prompt methods are still generally effective across various pre-trained graph models. Among these, the improvement is relatively notable for GAT. Introducing graph prompt techniques to the pre-trained GAT significantly enhances the transferability of the pre-trained model, especially on node-level datasets.

## E.5 Other Heat Maps of Other Graph Prompt Methods

Figure 6 displays the heat maps for various graph prompt methods applied to both node-level and graph-level tasks, offering a more intuitive visualization of the enhancements provided by graph prompt methods over the 'Pre-train & Fine-tune' approach. Through an analysis of these heat maps, it becomes evident that, irrespective of the specific graph prompt method or the type of task, node-level pre-training methods consistently outperform in node-level tasks, whereas graph-level pre-training methods exhibit superior performance in graph-level tasks. Moreover, the experimental results provide further evidence of the robust capability of graph prompt methods in effectively addressing the Negative transfer problem.

Table 7: Comparison of different methods across various datasets on 3-shot node classification tasks. The best results for each dataset are highlighted in bold with a dark red background, and the second-best results are underlined with a light red background.

| Methods\Datasets | Cora | Citeseer | Pubmed | Wisconsin | Texas | Actor | Flicker | Ogbn-arxiv |
|---|---|---|---|---|---|---|---|---|
| Supervised | $37.79_{\pm 9.16}$ | $35.18_{\pm 6.86}$ | $57.33_{\pm 4.64}$ | $41.03_{\pm 6.40}$ | $40.78_{\pm 12.55}$ | $18.62_{\pm 3.46}$ | $19.03_{\pm 5.08}$ | $21.01_{\pm 3.44}$ |
| Pre-train & Fine-tune | $51.97_{\pm 2.84}$ | $45.08_{\pm 2.09}$ | $65.40_{\pm 3.00}$ | $42.40_{\pm 7.77}$ | $43.13_{\pm 13.79}$ | $22.11_{\pm 1.97}$ | $27.34_{\pm 6.61}$ | $26.08_{\pm 1.89}$ |
| GPPT | $43.84_{\pm 6.11}$ | $42.34_{\pm 8.31}$ | $67.43_{\pm 2.96}$ | $34.29_{\pm 4.71}$ | $38.90_{\pm 8.86}$ | $21.65_{\pm 3.39}$ | $22.46_{\pm 4.05}$ | $23.35_{\pm 0.70}$ |
| All-in-one | $48.09_{\pm 4.83}$ | $48.09_{\pm 8.18}$ | $65.79_{\pm 5.79}$ | $89.62_{\pm 4.38}$ | $88.69_{\pm 1.08}$ | $24.23_{\pm 1.39}$ | $29.92_{\pm 1.52}$ | $16.79_{\pm 6.81}$ |
| Gprompt | $63.78_{\pm 5.77}$ | $60.00_{\pm 6.18}$ | $66.68_{\pm 3.53}$ | $92.52_{\pm 5.38}$ | $0.50_{\pm 0.41}$ | $29.67_{\pm 2.53}$ | $19.59_{\pm 3.20}$ | $82.07_{\pm 3.19}$ |
| GPF | $34.84_{\pm 19.83}$ | $25.92_{\pm 12.30}$ | $71.20_{\pm 2.82}$ | $98.92_{\pm 0.79}$ | $98.49_{\pm 1.45}$ | $37.44_{\pm 3.43}$ | $29.90_{\pm 3.01}$ | $77.50_{\pm 7.23}$ |
| GPF-plus | $56.38_{\pm 5.37}$ | $72.48_{\pm 5.63}$ | $70.85_{\pm 4.03}$ | $97.44_{\pm 4.52}$ | $98.83_{\pm 0.85}$ | $43.59_{\pm 4.52}$ | $24.71_{\pm 6.77}$ | $65.97_{\pm 1.11}$ |

Table 8: Comparison of different methods across various datasets on 3-shot graph classification tasks.

| Methods\Datasets | IMDB-B | COLLAB | PROTEINS | MUTAG | ENZYMES | COX2 | BZR | DD |
|---|---|---|---|---|---|---|---|---|
| Supervised | $53.33_{\pm 6.61}$ | $50.77_{\pm 2.44}$ | $61.33_{\pm 2.89}$ | $59.47_{\pm 8.34}$ | $15.96_{\pm 1.64}$ | $65.15_{\pm 18.61}$ | $52.35_{\pm 8.12}$ | $59.77_{\pm 1.10}$ |
| Pre-train & Fine-tune | $66.10_{\pm 0.70}$ | $56.10_{\pm 3.46}$ | $62.72_{\pm 2.39}$ | $59.87_{\pm 8.78}$ | $22.71_{\pm 0.86}$ | $69.97_{\pm 13.89}$ | $52.22_{\pm 10.64}$ | $59.70_{\pm 0.98}$ |
| GPPT | $59.48_{\pm 5.42}$ | $50.88_{\pm 6.31}$ | $64.74_{\pm 1.99}$ | $64.13_{\pm 18.31}$ | $19.12_{\pm 2.43}$ | $71.90_{\pm 14.28}$ | $78.46_{\pm 1.57}$ | $59.00_{\pm 6.34}$ |
| All-in-one | $65.67_{\pm 0.58}$ | $57.12_{\pm 1.99}$ | $69.84_{\pm 6.02}$ | $85.60_{\pm 2.22}$ | $23.96_{\pm 0.62}$ | $66.06_{\pm 18.23}$ | $61.98_{\pm 11.32}$ | $58.96_{\pm 5.93}$ |
| Gprompt | $64.35_{\pm 1.21}$ | $54.95_{\pm 9.47}$ | $64.94_{\pm 2.92}$ | $66.53_{\pm 14.84}$ | $22.08_{\pm 3.57}$ | $51.53_{\pm 13.08}$ | $54.63_{\pm 2.95}$ | $55.99_{\pm 7.53}$ |
| GPF | $65.97_{\pm 0.69}$ | $53.87_{\pm 3.44}$ | $63.35_{\pm 2.45}$ | $74.27_{\pm 1.55}$ | $23.87_{\pm 3.45}$ | $65.31_{\pm 19.45}$ | $74.38_{\pm 11.62}$ | $59.07_{\pm 0.65}$ |
| GPF-plus | $64.38_{\pm 2.30}$ | $56.50_{\pm 3.71}$ | $63.55_{\pm 1.85}$ | $75.20_{\pm 3.64}$ | $24.46_{\pm 2.27}$ | $65.25_{\pm 18.07}$ | $71.67_{\pm 14.87}$ | $59.51_{\pm 0.62}$ |

Table 9: Comparison of different methods across various datasets on 5-shot node classification tasks.

| Methods\Datasets | Cora | Citeseer | Pubmed | Wisconsin | Texas | Actor | flicker | Ogbn-arxiv |
|---|---|---|---|---|---|---|---|---|
| Supervised | 50.25 ±8.37 | 41.22 ±6.30 | 67.88 ±2.18 | 39.43 ±5.86 | 43.91 ±6.47 | 21.92 ±1.86 | 24.28 ±4.49 | 22.38 ±3.05 |
| Pre-train & Fine-tune | 62.66 ±3.55 | 39.54 ±3.54 | 70.91 ±4.87 | 42.97 ±8.99 | 47.19 ±7.37 | 22.92 ±1.22 | 35.12 ±4.72 | 28.84 ±3.11 |
| GPPT | 51.98 ±3.43 | 45.77 ±7.41 | 66.97 ±3.70 | 37.00 ±3.19 | 48.82 ±5.15 | 21.58 ±0.84 | 24.32 ±16.36 | 28.90 ±1.64 |
| All-in-one | 30.36 ±13.48 | 27.93 ±10.59 | 46.16 ±15.83 | 87.16 ±3.02 | 73.28 ±9.91 | 21.49 ±3.02 | 27.28 ±6.43 | 13.01 ±6.29 |
| Gprompt | 69.03 ±3.61 | 66.13 ±1.64 | 67.87 ±2.08 | 78.22 ±37.33 | 39.32 ±47.08 | 34.67 ±1.28 | 22.83 ±3.43 | 85.40 ±0.79 |
| GPF | 35.43 ±1.02 | 25.12 ±3.01 | 68.96 ±3.99 | 98.26 ±1.19 | 98.42 ±0.36 | 44.07 ±3.94 | 25.19 ±6.78 | 71.83 ±9.37 |
| GPF-plus | 63.28 ±4.69 | 75.73 ±2.19 | 69.59 ±4.33 | 99.01 ±1.43 | 99.12 ±0.95 | 44.58 ±5.95 | 39.62 ±8.22 | 66.88 ±6.14 |

Table 10: Comparison of different methods across various datasets on 5-shot graph classification tasks.

| Methods\Datasets | IMDB-B | COLLAB | PROTEINS | MUTAG | ENZYMES | COX2 | BZR | DD |
|---|---|---|---|---|---|---|---|---|
| Supervised | 62.60 ±4.01 | 55.23 ±4.26 | 62.90 ±5.03 | 73.47 ±3.92 | 25.67 ±0.48 | 64.99 ±10.42 | 51.48 ±2.29 | 63.59 ±2.86 |
| Pre-train & Fine-tune | 65.40 ±3.33 | 60.72 ±2.09 | 63.33 ±4.13 | 75.33 ±1.89 | 27.46 ±1.29 | 73.19 ±9.53 | 72.96 ±11.98 | 64.71 ±3.22 |
| GPPT | 66.37 ±3.59 | 54.05 ±4.58 | 58.27 ±4.63 | 70.53 ±3.90 | 22.17 ±2.34 | 67.88 ±17.34 | 69.63 ±14.96 | 60.02 ±3.24 |
| All-in-one | 63.62 ±2.30 | 57.86 ±5.88 | 71.37 ±4.89 | 84.67 ±0.73 | 26.71 ±2.17 | 62.95 ±8.57 | 62.78 ±10.18 | 63.44 ±1.35 |
| Gprompt | 66.70 ±3.87 | 60.76 ±5.08 | 62.94 ±1.38 | 73.07 ±2.13 | 21.46 ±2.27 | 53.35 ±7.75 | 59.38 ±14.43 | 58.28 ±2.18 |
| GPF | 67.80 ±5.58 | 59.65 ±6.25 | 63.37 ±4.37 | 74.00 ±3.65 | 27.00 ±0.78 | 66.27 ±14.57 | 61.05 ±11.51 | 61.06 ±2.63 |
| GPF-plus | 68.13 ±3.31 | 60.68 ±4.67 | 63.51 ±2.89 | 73.87 ±3.51 | 26.87 ±1.89 | 72.87 ±10.17 | 71.54 ±14.81 | 64.80 ±3.45 |

Table 11: Node classification performance Accuracy (%) (1-shot)

| Training schemes | Methods | Cora | CiteSeer | Pubmed | Wisconsin | Texas | Actor | Ogbn-arxiv |
|---|---|---|---|---|---|---|---|---|
| self-supervise | GCN | 26.56 ±5.55 | 21.78 ±7.32 | 39.37 ±16.34 | 41.60 ±3.10 | 37.97 ±5.80 | 20.57 ±4.47 | 10.99 ±3.19 |
| Pretrain + Fine-tuning | DGI | 33.15 ±7.84 | 21.64 ±3.92 | 42.01 ±12.54 | 37.49 ±5.13 | 45.31 ±5.01 | 19.76 ±3.53 | 7.21 ±2.91 |
| | GraphMAE | 32.93 ±3.17 | 21.26 ±3.57 | 42.99 ±14.25 | 36.80 ±7.17 | 37.81 ±8.62 | 19.86 ±2.70 | 12.35 ±3.60 |
| | EdgePreGPPT | 38.12 ±5.29 | 18.09 ±5.39 | 46.74 ±14.09 | 35.31 ±9.31 | 47.66 ±2.37 | 19.17 ±2.53 | 16.21 ±3.82 |
| | EdgePreGprompt | 35.57 ±5.83 | 22.28 ±3.80 | 41.50 ±7.54 | 40.69 ±4.13 | 40.62 ±7.95 | 20.74 ±4.12 | 14.83 ±2.38 |
| | GCL | 52.61 ±1.73 | 27.02 ±4.31 | 42.49 ±11.29 | 33.94 ±7.74 | 40.31 ±13.68 | 20.19 ±1.98 | 4.65 ±1.19 |
| | SimGRACE | 40.40 ±4.66 | 35.05 ±4.37 | 37.59 ±8.17 | 37.37 ±3.68 | 46.88 ±4.64 | 19.78 ±1.89 | 8.13 ±3.26 |
| GPPTPrompt | DGI | 30.47 ±3.53 | 37.26 ±6.17 | 35.62 ±7.74 | 29.94 ±10.40 | 29.29 ±14.57 | 21.76 ±2.00 | 3.80 ±6.19 |
| | GraphMAE | 27.39 ±10.26 | 21.54 ±4.00 | 48.31 ±17.72 | 29.83 ±9.34 | 25.04 ±10.38 | 22.58 ±1.97 | 4.35 ±6.01 |
| | EdgePreGPPT | 30.37 ±4.30 | 21.06 ±4.37 | 39.64 ±7.64 | 23.89 ±5.40 | 30.39 ±8.96 | 19.85 ±0.76 | 14.65 ±3.07 |
| | EdgePreGprompt | 25.52 ±4.42 | 21.85 ±4.30 | 46.20 ±10.76 | 30.40 ±6.81 | 22.68 ±12.82 | 21.52 ±1.13 | 2.05 ±1.43 |
| | GCL | 43.15 ±9.44 | 26.73 ±4.12 | 38.34 ±11.59 | 25.03 ±5.37 | 31.81 ±15.33 | 22.51 ±1.73 | 7.15 ±4.12 |
| | SimGRACE | 27.86 ±2.79 | 25.06 ±4.90 | 36.70 ±9.26 | 29.83 ±6.44 | 25.67 ±8.01 | 20.97 ±2.30 | 5.50 ±5.10 |
| ALL-in-one | DGI | 47.52 ±2.50 | 39.37 ±3.12 | 38.74 ±2.15 | 69.54 ±8.98 | 57.38 ±12.82 | 21.03 ±1.96 | 1.93 ±1.48 |
| | GraphMAE | 23.09 ±4.92 | 18.08 ±5.23 | 33.19 ±11.98 | 67.44 ±10.22 | 52.82 ±11.47 | 23.31 ±2.01 | 4.19 ±6.04 |
| | EdgePreGPPT | 49.63 ±5.26 | 35.06 ±2.37 | 40.73 ±11.32 | 78.24 ±16.68 | 58.62 ±5.54 | 21.49 ±1.27 | 5.62 ±3.95 |
| | EdgePreGprompt | 37.39 ±3.31 | 28.85 ±4.32 | 35.53 ±9.07 | 71.40 ±12.32 | 39.71 ±25.31 | 20.49 ±3.90 | 13.01 ±6.29 |
| | GCL | 52.39 ±10.17 | 37.37 ±4.15 | 45.17 ±6.93 | 42.19 ±1.05 | 65.49 ±7.06 | 24.61 ±2.80 | 6.70 ±6.01 |
| | SimGRACE | 35.99 ±2.76 | 40.41 ±2.80 | 30.23 ±7.03 | 67.44 ±8.42 | 59.22 ±20.17 | 21.03 ±2.23 | 5.72 ±1.61 |
| Gprompt | DGI | 36.46 ±5.39 | 36.25 ±10.26 | 33.65 ±5.29 | 74.00 ±4.40 | 31.00 ±37.32 | 23.85 ±3.52 | 48.47 ±5.87 |
| | GraphMAE | 50.58 ±7.34 | 42.84 ±10.59 | 39.74 ±15.35 | 78.51 ±9.67 | 22.34 ±3.57 | 22.34 ±3.57 | 66.66 ±5.04 |
| | EdgePreGPPT | 46.96 ±6.22 | 40.15 ±7.04 | 35.46 ±14.12 | 83.80 ±2.44 | 30.52 ±36.73 | 23.50 ±4.16 | 75.72 ±4.95 |
| | EdgePreGprompt | 48.11 ±9.89 | 48.07 ±5.62 | 33.54 ±16.66 | 81.93 ±2.61 | 33.25 ±40.11 | 19.89 ±1.38 | 70.55 ±7.66 |
| | GCL | 56.66 ±11.22 | 45.81 ±7.04 | 39.37 ±14.95 | 39.57 ±35.56 | 29.55 ±35.56 | 25.26 ±1.10 | 51.20 ±6.40 |
| | SimGRACE | 46.34 ±6.75 | 53.21 ±10.94 | 35.58 ±9.03 | 68.95 ±10.71 | 30.20 ±36.49 | 24.49 ±4.38 | 52.76 ±5.30 |
| GPF | DGI | 27.83 ±18.89 | 16.50 ±4.57 | 38.33 ±8.13 | 74.13 ±1.88 | 78.45 ±2.57 | 28.17 ±4.81 | 17.85 ±8.23 |
| | GraphMAE | 38.57 ±5.41 | 25.61 ±3.27 | 48.52 ±13.23 | 88.33 ±5.48 | 78.91 ±18.31 | 38.37 ±5.82 | 48.00 ±3.95 |
| | EdgePreGPPT | 15.29 ±8.41 | 12.33 ±5.33 | 43.78 ±6.02 | 88.67 ±5.78 | 86.15 ±5.31 | 25.66 ±3.33 | 78.37 ±5.47 |
| | EdgePreGprompt | 26.60 ±13.92 | 31.16 ±8.05 | 48.98 ±11.57 | 82.04 ±4.20 | 86.26 ±2.52 | 25.27 ±5.65 | 54.27 ±11.49 |
| | GCL | 23.16 ±5.11 | 16.77 ±1.39 | 49.99 ±8.86 | 59.50 ±13.30 | 87.40 ±3.40 | 20.68 ±6.70 | 23.55 ±0.45 |
| | SimGRACE | 32.01 ±11.21 | 19.43 ±2.10 | 37.27 ±6.09 | 79.48 ±2.62 | 87.39 ±1.64 | 28.70 ±3.35 | 37.56 ±6.10 |
| GPF-plus | DGI | 17.29 ±6.18 | 26.60 ±13.24 | 34.02 ±11.94 | 76.34 ±6.66 | 79.54 ±1.89 | 22.42 ±9.66 | 16.83 ±10.02 |
| | GraphMAE | 54.26 ±7.48 | 59.67 ±11.87 | 46.64 ±18.57 | 89.17 ±9.79 | 94.09 ±4.40 | 26.58 ±7.84 | 49.81 ±2.62 |
| | EdgePreGPPT | 28.49 ±18.73 | 28.04 ±14.31 | 46.51 ±15.84 | 91.03 ±4.11 | 92.14 ±5.46 | 29.32 ±8.56 | 71.98 ±12.23 |
| | EdgePreGprompt | 55.77 ±10.30 | 49.43 ±8.21 | 42.79 ±18.18 | 85.44 ±2.23 | 87.70 ±3.77 | 22.68 ±3.64 | 57.44 ±6.95 |
| | GCL | 34.18 ±17.71 | 28.86 ±22.88 | 37.02 ±11.29 | 68.57 ±8.77 | 95.83 ±4.19 | 22.82 ±4.99 | 32.11 ±4.86 |
| | SimGRACE | 21.33 ±14.86 | 24.61 ±21.21 | 35.90 ±9.06 | 73.64 ±5.25 | 90.74 ±3.12 | 20.51 ±4.24 | 46.71 ±3.17 |

Table 12: Node classification performance F1-score (1-shot)

| Training schemes | Methods | Cora | CiteSeer | Pubmed | Wisconsin | Texas | Actor | Ogbn-arxiv |
|---|---|---|---|---|---|---|---|---|
| self-supervise | GCN | $16.60_{\pm2.54}$ | $10.81_{\pm4.90}$ | $37.23_{\pm15.48}$ | $26.34_{\pm4.01}$ | $24.05_{\pm5.12}$ | $11.56_{\pm3.08}$ | $7.99_{\pm0.99}$ |
| Pretrain + Fine-tuning | DGI | $24.96_{\pm6.01}$ | $11.01_{\pm5.90}$ | $34.75_{\pm13.75}$ | $26.69_{\pm3.39}$ | $28.90_{\pm6.81}$ | $12.01_{\pm1.43}$ | $3.93_{\pm1.32}$ |
| | GraphMAE | $23.18_{\pm2.85}$ | $10.82_{\pm3.83}$ | $41.03_{\pm13.36}$ | $27.43_{\pm4.47}$ | $23.08_{\pm6.07}$ | $12.71_{\pm1.24}$ | $8.07_{\pm1.08}$ |
| | EdgePreGPPT | $35.92_{\pm4.06}$ | $10.86_{\pm2.29}$ | $40.62_{\pm18.09}$ | $23.56_{\pm3.43}$ | $29.03_{\pm5.16}$ | $14.59_{\pm3.09}$ | $12.13_{\pm2.04}$ |
| | EdgePreGprompt | $28.99_{\pm6.35}$ | $12.39_{\pm3.56}$ | $28.89_{\pm6.74}$ | $26.74_{\pm3.28}$ | $26.81_{\pm6.66}$ | $11.63_{\pm2.64}$ | $10.61_{\pm1.45}$ |
| | GCL | $47.14_{\pm3.15}$ | $21.86_{\pm3.67}$ | $38.30_{\pm10.89}$ | $14.27_{\pm3.90}$ | $24.52_{\pm7.59}$ | $15.91_{\pm0.98}$ | $2.88_{\pm0.51}$ |
| | SimGRACE | $33.22_{\pm3.29}$ | $30.78_{\pm3.91}$ | $32.79_{\pm6.60}$ | $23.71_{\pm2.97}$ | $29.53_{\pm6.44}$ | $15.73_{\pm1.20}$ | $2.88_{\pm0.50}$ |
| GPPTPrompt | DGI | $25.82_{\pm4.78}$ | $33.00_{\pm6.49}$ | $31.92_{\pm8.94}$ | $22.77_{\pm6.29}$ | $23.80_{\pm8.89}$ | $17.59_{\pm1.13}$ | $0.17_{\pm0.26}$ |
| | GraphMAE | $13.18_{\pm6.02}$ | $10.87_{\pm4.43}$ | $46.43_{\pm16.73}$ | $23.74_{\pm5.95}$ | $21.36_{\pm7.80}$ | $13.60_{\pm1.69}$ | $0.27_{\pm0.23}$ |
| | EdgePreGPPT | $28.54_{\pm3.87}$ | $17.62_{\pm4.24}$ | $34.55_{\pm8.09}$ | $20.61_{\pm5.49}$ | $23.84_{\pm5.22}$ | $18.48_{\pm0.57}$ | $9.15_{\pm1.18}$ |
| | EdgePreGprompt | $23.46_{\pm5.11}$ | $19.00_{\pm4.24}$ | $45.52_{\pm10.54}$ | $23.65_{\pm3.96}$ | $19.82_{\pm7.93}$ | $19.39_{\pm1.08}$ | $1.35_{\pm1.07}$ |
| | GCL | $38.99_{\pm8.32}$ | $23.76_{\pm3.97}$ | $36.75_{\pm12.92}$ | $18.08_{\pm6.69}$ | $25.64_{\pm8.12}$ | $19.62_{\pm0.56}$ | $1.52_{\pm0.68}$ |
| | SimGRACE | $21.70_{\pm2.66}$ | $22.13_{\pm4.48}$ | $31.55_{\pm10.44}$ | $21.48_{\pm3.16}$ | $21.75_{\pm5.75}$ | $17.15_{\pm1.75}$ | $0.66_{\pm0.39}$ |
| ALL-in-one | DGI | $34.76_{\pm3.89}$ | $26.67_{\pm4.46}$ | $22.68_{\pm5.29}$ | $52.08_{\pm6.70}$ | $31.14_{\pm20.37}$ | $14.37_{\pm2.36}$ | $0.24_{\pm0.21}$ |
| | GraphMAE | $10.82_{\pm4.45}$ | $7.04_{\pm2.29}$ | $25.46_{\pm9.17}$ | $53.76_{\pm2.54}$ | $37.08_{\pm9.37}$ | $12.68_{\pm2.63}$ | $0.19_{\pm0.26}$ |
| | EdgePreGPPT | $44.16_{\pm4.02}$ | $26.79_{\pm3.27}$ | $36.27_{\pm12.89}$ | $67.68_{\pm10.36}$ | $26.56_{\pm10.81}$ | $14.72_{\pm3.29}$ | $0.98_{\pm0.47}$ |
| | EdgePreGprompt | $24.93_{\pm4.99}$ | $14.58_{\pm5.03}$ | $30.99_{\pm7.62}$ | $62.98_{\pm8.70}$ | $28.81_{\pm17.76}$ | $11.85_{\pm1.48}$ | $0.56_{\pm0.27}$ |
| | GCL | $46.58_{\pm8.42}$ | $29.35_{\pm3.66}$ | $38.05_{\pm6.24}$ | $34.44_{\pm5.73}$ | $43.37_{\pm16.01}$ | $16.05_{\pm3.88}$ | $0.75_{\pm1.07}$ |
| | SimGRACE | $27.35_{\pm1.31}$ | $30.20_{\pm4.44}$ | $24.61_{\pm6.29}$ | $52.59_{\pm6.43}$ | $39.01_{\pm18.76}$ | $13.86_{\pm1.76}$ | $1.28_{\pm0.35}$ |
| Gprompt | DGI | $30.20_{\pm4.21}$ | $33.58_{\pm9.40}$ | $31.89_{\pm5.43}$ | $61.63_{\pm9.08}$ | $23.68_{\pm28.78}$ | $20.12_{\pm3.96}$ | $42.90_{\pm1.91}$ |
| | GraphMAE | $45.91_{\pm6.10}$ | $40.94_{\pm10.71}$ | $39.46_{\pm15.97}$ | $68.64_{\pm8.14}$ | $19.27_{\pm23.52}$ | $20.06_{\pm4.69}$ | $57.88_{\pm3.32}$ |
| | EdgePreGPPT | $44.15_{\pm7.57}$ | $37.31_{\pm6.26}$ | $29.87_{\pm12.41}$ | $74.76_{\pm4.26}$ | $24.14_{\pm29.42}$ | $20.14_{\pm3.34}$ | $69.86_{\pm1.92}$ |
| | EdgePreGprompt | $46.28_{\pm8.46}$ | $42.82_{\pm6.05}$ | $33.57_{\pm16.07}$ | $77.03_{\pm6.40}$ | $25.86_{\pm31.46}$ | $18.67_{\pm1.74}$ | $69.89_{\pm5.15}$ |
| | GCL | $49.86_{\pm10.36}$ | $40.41_{\pm9.12}$ | $38.04_{\pm13.45}$ | $65.53_{\pm6.42}$ | $21.15_{\pm25.84}$ | $22.00_{\pm1.74}$ | $45.32_{\pm3.89}$ |
| | SimGRACE | $38.55_{\pm5.02}$ | $49.65_{\pm11.42}$ | $31.13_{\pm9.15}$ | $61.22_{\pm2.58}$ | $29.20_{\pm35.62}$ | $21.39_{\pm3.95}$ | $46.73_{\pm4.62}$ |
| GPF | DGI | $19.39_{\pm17.53}$ | $4.68_{\pm1.17}$ | $28.70_{\pm9.13}$ | $57.50_{\pm2.20}$ | $68.55_{\pm8.62}$ | $19.25_{\pm10.07}$ | $6.59_{\pm3.79}$ |
| | GraphMAE | $23.79_{\pm5.49}$ | $12.89_{\pm1.99}$ | $45.36_{\pm15.88}$ | $82.97_{\pm6.10}$ | $73.83_{\pm15.04}$ | $31.69_{\pm5.47}$ | $19.47_{\pm3.26}$ |
| | EdgePreGPPT | $3.76_{\pm1.75}$ | $3.59_{\pm1.38}$ | $31.20_{\pm15.26}$ | $78.86_{\pm7.05}$ | $72.46_{\pm12.05}$ | $18.04_{\pm8.30}$ | $63.46_{\pm4.32}$ |
| | EdgePreGprompt | $15.85_{\pm13.63}$ | $18.63_{\pm7.34}$ | $39.90_{\pm9.00}$ | $80.44_{\pm7.43}$ | $78.43_{\pm9.49}$ | $19.72_{\pm8.08}$ | $29.38_{\pm6.81}$ |
| | GCL | $9.76_{\pm4.57}$ | $4.78_{\pm0.34}$ | $38.27_{\pm17.17}$ | $35.78_{\pm8.46}$ | $76.01_{\pm7.84}$ | $11.69_{\pm9.39}$ | $18.18_{\pm1.66}$ |
| | SimGRACE | $18.73_{\pm10.08}$ | $5.90_{\pm1.13}$ | $22.65_{\pm5.69}$ | $70.22_{\pm3.85}$ | $74.43_{\pm7.73}$ | $17.94_{\pm5.36}$ | $14.16_{\pm1.23}$ |
| GPF-plus | DGI | $9.94_{\pm8.40}$ | $13.74_{\pm16.38}$ | $24.21_{\pm9.44}$ | $62.02_{\pm12.59}$ | $67.91_{\pm7.85}$ | $14.51_{\pm10.72}$ | $5.37_{\pm4.20}$ |
| | GraphMAE | $50.19_{\pm10.64}$ | $56.22_{\pm13.99}$ | $42.38_{\pm15.08}$ | $84.72_{\pm15.08}$ | $81.36_{\pm10.39}$ | $19.25_{\pm5.53}$ | $23.38_{\pm1.12}$ |
| | EdgePreGPPT | $19.52_{\pm17.74}$ | $15.32_{\pm19.12}$ | $40.64_{\pm17.93}$ | $85.24_{\pm5.45}$ | $78.97_{\pm3.30}$ | $24.56_{\pm8.79}$ | $60.35_{\pm7.73}$ |
| | EdgePreGprompt | $53.28_{\pm11.46}$ | $48.37_{\pm9.28}$ | $40.02_{\pm18.18}$ | $74.18_{\pm6.55}$ | $75.46_{\pm5.42}$ | $15.58_{\pm4.14}$ | $31.37_{\pm2.50}$ |
| | GCL | $26.14_{\pm21.11}$ | $18.90_{\pm24.01}$ | $33.64_{\pm11.41}$ | $47.94_{\pm17.19}$ | $86.22_{\pm10.29}$ | $16.08_{\pm3.05}$ | $15.78_{\pm1.60}$ |
| | SimGRACE | $14.35_{\pm19.65}$ | $15.23_{\pm20.93}$ | $23.74_{\pm12.56}$ | $52.05_{\pm5.74}$ | $80.81_{\pm8.01}$ | $11.83_{\pm4.68}$ | $21.81_{\pm1.06}$ |

Table 13: Node classification performance AUROC (1-shot)

| Training schemes | Methods | Cora | CiteSeer | Pubmed | Wisconsin | Texas | Actor | Ogbn-arxiv |
|---|---|---|---|---|---|---|---|---|
| self-supervise | GCN | $76.71_{\pm1.76}$ | $63.02_{\pm4.09}$ | $62.01_{\pm18.06}$ | $61.53_{\pm4.96}$ | $62.86_{\pm10.22}$ | $51.16_{\pm1.01}$ | $72.48_{\pm1.34}$ |
| Pretrain + Fine-tuning | DGI | $79.33_{\pm2.82}$ | $65.16_{\pm4.85}$ | $63.47_{\pm13.36}$ | $59.72_{\pm2.84}$ | $65.16_{\pm9.96}$ | $50.13_{\pm0.96}$ | $64.22_{\pm2.71}$ |
| | GraphMAE | $78.93_{\pm0.67}$ | $63.92_{\pm3.46}$ | $66.78_{\pm16.67}$ | $60.83_{\pm5.18}$ | $61.79_{\pm9.63}$ | $51.01_{\pm0.55}$ | $72.86_{\pm1.61}$ |
| | EdgePreGPPT | $78.44_{\pm1.58}$ | $60.74_{\pm3.98}$ | $63.49_{\pm15.75}$ | $58.91_{\pm3.16}$ | $65.05_{\pm9.01}$ | $50.79_{\pm0.65}$ | $77.26_{\pm2.75}$ |
| | EdgePreGprompt | $80.12_{\pm2.93}$ | $65.05_{\pm4.65}$ | $60.89_{\pm13.56}$ | $61.08_{\pm5.22}$ | $62.77_{\pm10.98}$ | $51.47_{\pm1.28}$ | $75.81_{\pm2.26}$ |
| | GCL | $86.34_{\pm1.24}$ | $68.32_{\pm1.24}$ | $57.56_{\pm15.85}$ | $43.79_{\pm4.25}$ | $63.09_{\pm10.20}$ | $51.19_{\pm0.63}$ | $63.36_{\pm1.49}$ |
| | SimGRACE | $76.63_{\pm1.67}$ | $71.91_{\pm2.47}$ | $58.10_{\pm9.13}$ | $57.76_{\pm2.12}$ | $65.06_{\pm9.74}$ | $50.81_{\pm0.66}$ | $56.33_{\pm0.77}$ |
| GPPTPrompt | DGI | $63.97_{\pm5.40}$ | $71.52_{\pm5.15}$ | $51.57_{\pm5.86}$ | $59.89_{\pm2.95}$ | $57.22_{\pm9.90}$ | $50.71_{\pm0.86}$ | $50.07_{\pm0.08}$ |
| | GraphMAE | $78.20_{\pm3.89}$ | $61.64_{\pm4.31}$ | $64.12_{\pm15.44}$ | $55.48_{\pm6.44}$ | $53.95_{\pm3.70}$ | $51.71_{\pm1.74}$ | $59.28_{\pm2.39}$ |
| | EdgePreGPPT | $65.75_{\pm4.04}$ | $54.28_{\pm5.62}$ | $55.12_{\pm11.88}$ | $54.59_{\pm4.65}$ | $50.55_{\pm3.83}$ | $49.94_{\pm0.53}$ | $74.38_{\pm1.59}$ |
| | EdgePreGprompt | $62.60_{\pm2.50}$ | $54.23_{\pm6.14}$ | $61.41_{\pm11.42}$ | $57.58_{\pm2.23}$ | $53.16_{\pm6.75}$ | $50.32_{\pm0.29}$ | $63.71_{\pm2.13}$ |
| | GCL | $72.09_{\pm7.61}$ | $59.06_{\pm3.51}$ | $55.95_{\pm13.50}$ | $54.51_{\pm4.05}$ | $56.59_{\pm7.31}$ | $51.76_{\pm0.95}$ | $54.45_{\pm1.08}$ |
| | SimGRACE | $64.83_{\pm2.80}$ | $59.61_{\pm4.19}$ | $51.84_{\pm12.98}$ | $56.88_{\pm4.39}$ | $53.42_{\pm6.67}$ | $50.48_{\pm0.77}$ | $50.72_{\pm0.43}$ |
| ALL-in-one | DGI | $75.29_{\pm1.07}$ | $71.50_{\pm1.46}$ | $55.69_{\pm1.38}$ | $78.94_{\pm2.46}$ | $64.75_{\pm6.31}$ | $51.05_{\pm0.96}$ | $55.21_{\pm1.45}$ |
| | GraphMAE | $73.99_{\pm5.63}$ | $49.98_{\pm3.07}$ | $53.18_{\pm13.11}$ | $81.15_{\pm2.57}$ | $62.90_{\pm3.67}$ | $50.96_{\pm1.97}$ | $50.21_{\pm1.01}$ |
| | EdgePreGPPT | $83.26_{\pm1.22}$ | $69.82_{\pm1.32}$ | $73.63_{\pm3.06}$ | $87.52_{\pm6.90}$ | $75.38_{\pm10.13}$ | $49.99_{\pm0.28}$ | $54.22_{\pm5.18}$ |
| | EdgePreGprompt | $73.84_{\pm3.01}$ | $62.35_{\pm6.11}$ | $61.26_{\pm7.66}$ | $89.84_{\pm3.19}$ | $67.99_{\pm6.33}$ | $51.48_{\pm2.87}$ | $75.81_{\pm2.83}$ |
| | GCL | $84.34_{\pm4.32}$ | $72.36_{\pm2.82}$ | $68.22_{\pm6.38}$ | $81.46_{\pm3.63}$ | $69.39_{\pm2.56}$ | $47.63_{\pm2.10}$ | $55.77_{\pm4.78}$ |
| | SimGRACE | $72.30_{\pm1.28}$ | $69.88_{\pm0.83}$ | $52.94_{\pm2.50}$ | $81.86_{\pm5.69}$ | $70.92_{\pm7.19}$ | $48.18_{\pm0.97}$ | $65.43_{\pm1.59}$ |
| Gprompt | DGI | $70.70_{\pm2.26}$ | $70.93_{\pm4.80}$ | $52.96_{\pm2.53}$ | $85.91_{\pm2.99}$ | $57.93_{\pm10.19}$ | $52.82_{\pm3.88}$ | $90.60_{\pm1.08}$ |
| | GraphMAE | $80.67_{\pm5.24}$ | $70.72_{\pm5.34}$ | $60.66_{\pm19.63}$ | $93.63_{\pm2.61}$ | $59.57_{\pm11.81}$ | $52.87_{\pm4.78}$ | $94.04_{\pm1.32}$ |
| | EdgePreGPPT | $84.03_{\pm2.26}$ | $67.95_{\pm2.69}$ | $44.60_{\pm13.08}$ | $95.25_{\pm2.14}$ | $60.65_{\pm12.95}$ | $53.55_{\pm3.61}$ | $96.40_{\pm0.74}$ |
| | EdgePreGprompt | $81.90_{\pm4.04}$ | $74.85_{\pm2.68}$ | $58.34_{\pm22.51}$ | $96.17_{\pm1.26}$ | $62.03_{\pm14.44}$ | $50.16_{\pm1.75}$ | $94.39_{\pm1.12}$ |
| | GCL | $83.03_{\pm4.16}$ | $78.33_{\pm5.28}$ | $58.24_{\pm13.76}$ | $88.03_{\pm5.49}$ | $58.33_{\pm10.63}$ | $54.45_{\pm2.93}$ | $92.72_{\pm0.72}$ |
| | SimGRACE | $76.99_{\pm3.17}$ | $84.09_{\pm2.88}$ | $51.91_{\pm12.26}$ | $86.09_{\pm6.13}$ | $61.60_{\pm14.00}$ | $53.63_{\pm4.60}$ | $91.93_{\pm1.25}$ |
| GPF | DGI | $65.74_{\pm10.40}$ | $50.00_{\pm0.00}$ | $51.90_{\pm7.57}$ | $83.72_{\pm4.30}$ | $68.70_{\pm1.83}$ | $58.84_{\pm3.16}$ | $64.74_{\pm2.99}$ |
| | GraphMAE | $73.69_{\pm2.75}$ | $66.16_{\pm0.22}$ | $73.35_{\pm8.02}$ | $98.84_{\pm0.77}$ | $84.20_{\pm10.56}$ | $65.43_{\pm3.58}$ | $77.64_{\pm0.97}$ |
| | EdgePreGPPT | $62.03_{\pm6.87}$ | $57.99_{\pm6.03}$ | $68.25_{\pm7.09}$ | $98.54_{\pm0.99}$ | $84.45_{\pm7.94}$ | $55.90_{\pm6.20}$ | $93.53_{\pm1.06}$ |
| | EdgePreGprompt | $70.83_{\pm8.42}$ | $71.94_{\pm3.31}$ | $71.40_{\pm8.27}$ | $97.48_{\pm0.73}$ | $81.31_{\pm7.35}$ | $56.13_{\pm4.33}$ | $82.60_{\pm2.47}$ |
| | GCL | $74.17_{\pm5.66}$ | $61.21_{\pm7.77}$ | $70.39_{\pm4.76}$ | $62.05_{\pm4.35}$ | $82.48_{\pm5.23}$ | $54.42_{\pm3.73}$ | $73.36_{\pm1.69}$ |
| | SimGRACE | $69.60_{\pm10.54}$ | $62.47_{\pm3.70}$ | $50.73_{\pm3.64}$ | $93.71_{\pm1.37}$ | $85.70_{\pm8.79}$ | $60.48_{\pm2.78}$ | $61.29_{\pm1.33}$ |
| GPF-plus | DGI | $58.07_{\pm8.25}$ | $55.15_{\pm11.69}$ | $55.70_{\pm11.04}$ | $91.39_{\pm4.68}$ | $78.71_{\pm9.15}$ | $57.02_{\pm8.99}$ | $60.73_{\pm3.11}$ |
| | GraphMAE | $86.94_{\pm4.26}$ | $87.43_{\pm3.46}$ | $66.74_{\pm18.62}$ | $99.25_{\pm1.27}$ | $86.83_{\pm8.56}$ | $63.32_{\pm6.43}$ | $80.32_{\pm3.00}$ |
| | EdgePreGPPT | $68.27_{\pm10.69}$ | $59.85_{\pm17.18}$ | $71.27_{\pm10.40}$ | $99.12_{\pm1.13}$ | $83.10_{\pm7.26}$ | $61.72_{\pm5.16}$ | $93.52_{\pm1.09}$ |
| | EdgePreGprompt | $86.33_{\pm4.94}$ | $88.60_{\pm5.18}$ | $63.12_{\pm18.73}$ | $96.59_{\pm3.43}$ | $82.30_{\pm6.38}$ | $61.97_{\pm5.10}$ | $84.36_{\pm0.87}$ |
| | GCL | $80.06_{\pm6.24}$ | $61.58_{\pm14.77}$ | $55.63_{\pm15.68}$ | $77.34_{\pm13.37}$ | $83.65_{\pm8.19}$ | $58.86_{\pm3.80}$ | $70.39_{\pm3.22}$ |
| | SimGRACE | $64.39_{\pm11.25}$ | $64.11_{\pm10.53}$ | $54.79_{\pm10.68}$ | $88.74_{\pm2.84}$ | $86.70_{\pm8.72}$ | $55.75_{\pm4.51}$ | $69.09_{\pm2.34}$ |

Table 14: Node classification performance Accuracy (%) (3-shot)

| Training schemes | Methods | Cora | CiteSeer | Pubmed | Wisconsin | Texas | Actor | Ogbn-arxiv |
|---|---|---|---|---|---|---|---|---|
| self-supervise | GCN | 37.79 ±9.16 | 35.18 ±6.86 | 57.33 ±4.64 | 41.03 ±6.40 | 40.78 ±12.55 | 18.62 ±3.46 | 19.03 ±5.08 |
| Pretrain + Fine-tuning | DGI | 45.84 ±3.29 | 34.91 ±10.07 | 63.00 ±6.83 | 39.43 ±6.93 | 43.13 ±13.79 | 20.27 ±1.70 | 21.85 ±5.37 |
| | GraphMAE | 45.15 ±4.77 | 28.59 ±6.41 | 65.40 ±3.00 | 41.49 ±5.19 | 40.94 ±13.91 | 19.03 ±3.17 | 19.63 ±7.64 |
| | EdgePreGPPT | 51.97 ±2.84 | 33.19 ±6.79 | 64.38 ±3.59 | 42.40 ±7.77 | 34.69 ±12.07 | 20.83 ±2.24 | 20.51 ±4.02 |
| | EdgePredGprompt | 40.33 ±6.62 | 33.39 ±5.51 | 63.49 ±3.17 | 41.37 ±7.28 | 34.53 ±10.09 | 19.29 ±1.95 | 27.34 ±6.61 |
| | GCL | 49.39 ±9.15 | 38.40 ±3.06 | 62.79 ±3.21 | 41.26 ±6.26 | 40.31 ±13.36 | 21.06 ±1.27 | 17.37 ±7.34 |
| | SimGRACE | 43.61 ±5.41 | 45.08 ±2.09 | 62.66 ±3.21 | 40.11 ±8.04 | 40.94 ±13.73 | 22.11 ±1.97 | 16.06 ±2.05 |
| GPPTPrompt | DGI | 37.46 ±6.27 | 42.34 ±8.31 | 44.36 ±3.67 | 34.29 ±4.71 | 37.80 ±8.98 | 21.42 ±3.13 | 14.34 ±14.10 |
| | GraphMAE | 30.93 ±3.64 | 20.76 ±2.64 | 67.43 ±2.96 | 33.60 ±2.30 | 37.64 ±5.88 | 21.65 ±3.39 | 17.24 ±13.25 |
| | EdgePreGPPT | 35.05 ±2.95 | 24.26 ±2.55 | 58.66 ±5.93 | 29.37 ±6.22 | 38.90 ±7.56 | 20.35 ±0.43 | 22.46 ±4.05 |
| | EdgePredGprompt | 27.94 ±5.07 | 23.21 ±2.95 | 64.98 ±5.35 | 33.49 ±5.49 | 36.06 ±12.01 | 19.85 ±0.31 | 18.73 ±4.77 |
| | GCL | 43.84 ±6.11 | 27.09 ±4.57 | 51.30 ±6.35 | 23.89 ±5.35 | 38.90 ±8.86 | 20.60 ±1.10 | 13.04 ±6.44 |
| | SimGRACE | 29.66 ±5.49 | 29.63 ±1.87 | 43.95 ±3.55 | 31.66 ±6.21 | 34.65 ±6.97 | 21.08 ±1.16 | 14.13 ±13.89 |
| ALL-in-one | DGI | 33.80 ±6.36 | 31.08 ±2.95 | 48.09 ±2.81 | 74.71 ±2.35 | 31.94 ±7.00 | 24.23 ±1.39 | 29.92 ±1.52 |
| | GraphMAE | 19.25 ±3.11 | 19.02 ±5.19 | 64.03 ±4.70 | 76.90 ±2.98 | 65.93 ±17.42 | 19.56 ±2.48 | 13.15 ±4.63 |
| | EdgePreGPPT | 48.09 ±4.83 | 34.29 ±1.85 | 58.36 ±4.63 | 89.62 ±4.38 | 88.69 ±1.08 | 20.91 ±3.37 | 14.41 ±5.57 |
| | EdgePredGprompt | 29.54 ±6.25 | 24.63 ±3.57 | 65.79 ±5.79 | 84.29 ±2.32 | 73.89 ±16.92 | 20.70 ±2.58 | 12.91 ±2.08 |
| | GCL | 27.89 ±18.34 | 43.96 ±6.69 | 51.44 ±4.46 | 67.06 ±9.70 | 78.86 ±4.06 | 22.14 ±2.57 | 31.15 ±2.25 |
| | SimGRACE | 25.16 ±11.78 | 48.09 ±8.18 | 47.87 ±3.53 | 76.78 ±1.28 | 82.93 ±5.04 | 21.99 ±1.17 | 15.47 ±4.05 |
| Gprompt | DGI | 37.42 ±7.07 | 41.19 ±5.30 | 38.69 ±1.54 | 71.54 ±10.15 | 0.50 ±0.41 | 26.64 ±4.45 | 19.40 ±5.30 |
| | GraphMAE | 63.78 ±5.77 | 57.15 ±3.90 | 62.47 ±4.29 | 86.48 ±7.48 | 0.50 ±0.41 | 29.21 ±3.27 | 18.90 ±3.10 |
| | EdgePreGPPT | 51.29 ±7.07 | 34.79 ±4.07 | 53.63 ±7.62 | 89.10 ±4.09 | 0.50 ±0.41 | 26.44 ±3.36 | 18.14 ±3.54 |
| | EdgePredGprompt | 53.37 ±6.30 | 56.19 ±2.95 | 66.68 ±3.53 | 92.52 ±5.38 | 0.50 ±0.41 | 21.78 ±1.85 | 19.59 ±3.20 |
| | GCL | 56.61 ±8.02 | 56.28 ±5.89 | 50.33 ±5.18 | 85.83 ±4.53 | 0.50 ±0.41 | 29.67 ±2.53 | 18.94 ±2.93 |
| | SimGRACE | 45.22 ±5.69 | 60.00 ±6.18 | 43.58 ±5.78 | 82.84 ±1.52 | 0.50 ±0.41 | 25.07 ±1.10 | 15.74 ±4.18 |
| GPF | DGI | 25.14 ±19.07 | 25.92 ±12.30 | 38.49 ±14.50 | 76.22 ±5.46 | 44.94 ±4.84 | 25.03 ±1.66 | 29.90 ±3.01 |
| | GraphMAE | 33.93 ±8.01 | 23.91 ±5.11 | 71.20 ±2.82 | 98.92 ±0.79 | 94.67 ±3.28 | 37.44 ±3.43 | 12.22 ±5.44 |
| | EdgePreGPPT | 23.05 ±9.03 | 18.81 ±4.35 | 65.06 ±2.22 | 96.63 ±2.53 | 94.85 ±1.67 | 30.47 ±3.57 | 19.76 ±4.75 |
| | EdgePredGprompt | 32.57 ±9.57 | 21.75 ±2.88 | 67.57 ±14.07 | 94.58 ±5.67 | 98.49 ±1.45 | 25.64 ±8.58 | 15.72 ±5.16 |
| | GCL | 34.84 ±19.83 | 23.02 ±5.56 | 49.15 ±6.46 | 56.93 ±12.66 | 94.66 ±3.87 | 21.95 ±6.54 | 8.90 ±2.80 |
| | SimGRACE | 27.80 ±18.09 | 21.20 ±11.25 | 31.75 ±13.86 | 86.10 ±0.76 | 97.83 ±1.13 | 22.82 ±3.16 | 10.71 ±1.11 |
| GPF-plus | DGI | 20.80 ±13.69 | 24.46 ±17.23 | 51.70 ±11.29 | 85.61 ±8.19 | 72.61 ±22.35 | 23.99 ±6.98 | 16.16 ±6.23 |
| | GraphMAE | 56.38 ±5.37 | 72.48 ±5.63 | 70.85 ±4.03 | 97.44 ±4.52 | 97.00 ±0.68 | 43.59 ±4.52 | 22.50 ±14.10 |
| | EdgePreGPPT | 32.62 ±23.21 | 22.06 ±21.01 | 67.20 ±1.67 | 97.34 ±2.33 | 98.81 ±0.85 | 34.38 ±5.23 | 23.56 ±5.67 |
| | EdgePredGprompt | 52.74 ±7.08 | 71.56 ±8.03 | 68.84 ±4.33 | 96.96 ±4.60 | 97.01 ±1.62 | 30.87 ±6.61 | 24.71 ±6.77 |
| | GCL | 30.34 ±21.41 | 27.17 ±24.65 | 61.23 ±11.39 | 62.76 ±14.68 | 97.18 ±1.93 | 36.87 ±2.45 | 5.42 ±0.82 |
| | SimGRACE | 30.95 ±19.32 | 28.97 ±20.66 | 41.96 ±15.64 | 80.64 ±3.99 | 97.34 ±1.84 | 34.89 ±6.03 | 13.16 ±5.89 |

Table 15: Node classification performance F1-score (3-shot)

| Training schemes | Methods | Cora | CiteSeer | Pubmed | Wisconsin | Texas | Actor | Ogbn-arxiv |
|---|---|---|---|---|---|---|---|---|
| self-supervise | GCN | 37.64 ±6.32 | 28.68 ±8.09 | 53.04 ±6.01 | 34.90 ±5.93 | 29.16 ±7.77 | 13.69 ±2.87 | 15.21 ±2.66 |
| Pretrain + Fine-tuning | DGI | 41.76 ±1.48 | 27.23 ±9.82 | 61.18 ±10.18 | 35.01 ±5.55 | 31.12 ±8.67 | 16.05 ±0.75 | 16.45 ±1.97 |
| | GraphMAE | 45.70 ±6.35 | 20.60 ±5.72 | 64.33 ±3.42 | 34.08 ±4.01 | 28.29 ±8.05 | 16.30 ±2.37 | 14.26 ±4.46 |
| | EdgePreGPPT | 49.76 ±2.52 | 28.14 ±6.04 | 63.62 ±4.81 | 35.03 ±5.65 | 25.55 ±6.04 | 17.79 ±1.62 | 16.68 ±2.65 |
| | EdgePredGprompt | 41.69 ±5.58 | 23.81 ±4.86 | 62.48 ±4.02 | 35.42 ±6.02 | 26.05 ±6.95 | 12.98 ±2.02 | 19.33 ±2.46 |
| | GCL | 47.79 ±9.36 | 34.47 ±3.44 | 63.10 ±2.93 | 22.42 ±6.61 | 29.45 ±8.86 | 19.91 ±1.08 | 7.46 ±3.10 |
| | SimGRACE | 41.71 ±4.09 | 42.01 ±1.56 | 63.14 ±3.43 | 34.28 ±5.18 | 29.55 ±7.96 | 17.92 ±1.61 | 11.68 ±1.37 |
| GPPTPrompt | DGI | 33.94 ±7.56 | 36.13 ±8.65 | 44.13 ±3.80 | 30.91 ±2.41 | 34.81 ±7.09 | 19.30 ±2.98 | 3.50 ±2.51 |
| | GraphMAE | 25.31 ±6.18 | 6.91 ±2.93 | 67.61 ±2.62 | 29.51 ±3.03 | 33.87 ±3.38 | 18.03 ±1.97 | 6.06 ±5.21 |
| | EdgePreGPPT | 32.77 ±2.69 | 19.54 ±1.23 | 58.07 ±6.41 | 27.20 ±3.72 | 33.65 ±5.16 | 19.32 ±0.53 | 17.24 ±1.62 |
| | EdgePredGprompt | 20.10 ±4.36 | 20.60 ±1.98 | 62.73 ±4.96 | 29.19 ±4.29 | 30.98 ±7.36 | 18.99 ±0.70 | 15.47 ±2.95 |
| | GCL | 41.25 ±4.50 | 25.71 ±4.25 | 50.27 ±6.65 | 15.65 ±5.40 | 34.61 ±6.65 | 19.81 ±1.02 | 5.96 ±3.82 |
| | SimGRACE | 27.19 ±4.01 | 27.09 ±2.43 | 42.79 ±3.17 | 26.79 ±4.55 | 32.38 ±5.73 | 19.02 ±0.50 | 4.01 ±2.59 |
| ALL-in-one | DGI | 22.05 ±4.15 | 25.31 ±3.49 | 42.82 ±4.66 | 63.22 ±7.20 | 37.71 ±11.00 | 11.04 ±2.36 | 15.79 ±0.87 |
| | GraphMAE | 8.10 ±2.43 | 7.36 ±2.81 | 62.24 ±5.80 | 51.62 ±2.63 | 58.91 ±14.27 | 11.52 ±0.97 | 9.72 ±3.20 |
| | EdgePreGPPT | 41.54 ±1.81 | 27.24 ±3.58 | 56.20 ±5.02 | 80.48 ±6.96 | 82.89 ±2.60 | 9.51 ±3.89 | 9.61 ±4.38 |
| | EdgePredGprompt | 20.33 ±9.09 | 12.09 ±3.36 | 63.18 ±7.52 | 79.59 ±6.58 | 68.71 ±14.50 | 7.89 ±2.38 | 9.96 ±1.24 |
| | GCL | 14.85 ±18.09 | 33.38 ±6.65 | 50.52 ±3.48 | 61.79 ±11.38 | 69.09 ±6.96 | 16.40 ±1.47 | 14.33 ±0.41 |
| | SimGRACE | 12.71 ±11.42 | 40.22 ±8.61 | 44.79 ±2.93 | 64.88 ±4.35 | 79.46 ±5.10 | 13.70 ±2.29 | 10.07 ±1.80 |
| Gprompt | DGI | 32.92 ±6.82 | 37.71 ±5.52 | 37.90 ±1.53 | 60.12 ±10.43 | 0.20 ±0.16 | 26.39 ±4.24 | 14.59 ±2.78 |
| | GraphMAE | 59.30 ±7.36 | 53.60 ±3.06 | 61.33 ±4.81 | 79.06 ±11.48 | 0.20 ±0.16 | 29.02 ±3.53 | 14.78 ±1.50 |
| | EdgePreGPPT | 46.21 ±6.52 | 29.81 ±3.69 | 48.94 ±10.06 | 80.60 ±7.17 | 0.20 ±0.16 | 24.73 ±3.32 | 15.68 ±2.29 |
| | EdgePredGprompt | 52.43 ±6.41 | 53.88 ±3.50 | 65.02 ±2.80 | 91.03 ±5.30 | 0.20 ±0.16 | 20.40 ±1.65 | 15.56 ±1.69 |
| | GCL | 51.54 ±8.49 | 52.46 ±5.08 | 49.16 ±11.12 | 65.08 ±11.12 | 0.20 ±0.16 | 28.21 ±2.34 | 14.80 ±1.34 |
| | SimGRACE | 40.83 ±3.88 | 56.25 ±7.01 | 42.59 ±5.51 | 76.68 ±3.65 | 0.20 ±0.16 | 25.41 ±1.35 | 12.58 ±1.91 |
| GPF | DGI | 12.93 ±16.42 | 12.92 ±13.22 | 32.76 ±17.33 | 67.09 ±7.72 | 46.87 ±5.91 | 15.68 ±9.94 | 14.93 ±1.60 |
| | GraphMAE | 19.59 ±5.22 | 14.48 ±5.04 | 70.15 ±2.68 | 94.99 ±4.12 | 83.16 ±10.27 | 31.56 ±8.62 | 5.27 ±5.58 |
| | EdgePreGPPT | 9.90 ±9.04 | 7.40 ±3.59 | 63.04 ±3.09 | 80.32 ±6.04 | 85.55 ±8.73 | 22.38 ±8.76 | 14.77 ±2.22 |
| | EdgePredGprompt | 21.59 ±5.95 | 11.79 ±3.50 | 62.02 ±21.60 | 93.30 ±5.43 | 86.36 ±10.64 | 22.04 ±11.41 | 11.51 ±3.42 |
| | GCL | 18.90 ±20.40 | 8.10 ±4.95 | 47.10 ±8.36 | 30.08 ±6.91 | 81.66 ±11.83 | 14.24 ±8.36 | 4.28 ±1.13 |
| | SimGRACE | 13.21 ±14.54 | 10.26 ±8.85 | 25.30 ±17.26 | 77.51 ±2.92 | 85.34 ±10.18 | 15.37 ±7.26 | 5.41 ±1.24 |
| GPF-plus | DGI | 10.87 ±13.23 | 14.35 ±19.57 | 43.25 ±20.47 | 78.25 ±14.06 | 63.25 ±15.75 | 14.70 ±3.64 | 10.41 ±3.06 |
| | GraphMAE | 55.76 ±3.81 | 70.45 ±6.54 | 69.78 ±3.58 | 97.33 ±2.96 | 83.31 ±9.41 | 45.02 ±4.47 | 12.22 ±4.77 |
| | EdgePreGPPT | 20.90 ±24.90 | 15.30 ±23.70 | 65.43 ±1.28 | 94.18 ±5.80 | 86.54 ±10.00 | 32.74 ±6.14 | 14.90 ±2.10 |
| | EdgePredGprompt | 52.13 ±7.14 | 68.79 ±9.94 | 67.98 ±3.69 | 94.78 ±4.60 | 85.14 ±8.73 | 30.59 ±7.43 | 15.43 ±2.65 |
| | GCL | 21.84 ±23.15 | 18.50 ±28.28 | 59.42 ±13.43 | 47.29 ±15.14 | 84.24 ±8.20 | 30.93 ±5.62 | 4.37 ±1.56 |
| | SimGRACE | 20.11 ±22.26 | 18.43 ±24.95 | 27.58 ±21.15 | 61.92 ±12.87 | 85.24 ±9.54 | 30.11 ±6.73 | 6.91 ±2.30 |

Table 16: Node classification performance AUROC (3-shot)

| Training schemes | Methods | Cora | CiteSeer | Pubmed | Wisconsin | Texas | Actor | Ogbn-arxiv |
|---|---|---|---|---|---|---|---|---|
| self-supervise | GCN | 86.20 $_{\pm2.47}$ | 76.31 $_{\pm3.45}$ | 80.24 $_{\pm2.82}$ | 66.44 $_{\pm6.12}$ | 59.85 $_{\pm12.05}$ | 51.48 $_{\pm0.74}$ | 59.17 $_{\pm1.26}$ |
| Pretrain + Fine-tuning | DGI | 84.23 $_{\pm2.75}$ | 74.32 $_{\pm4.21}$ | 82.28 $_{\pm5.43}$ | 67.49 $_{\pm3.80}$ | 55.28 $_{\pm10.97}$ | 51.93 $_{\pm1.20}$ | 61.28 $_{\pm1.40}$ |
|  | GraphMAE | 86.80 $_{\pm2.35}$ | 75.07 $_{\pm2.48}$ | 84.70 $_{\pm1.28}$ | 66.58 $_{\pm6.51}$ | 58.48 $_{\pm11.37}$ | 51.78 $_{\pm1.01}$ | 59.12 $_{\pm1.66}$ |
|  | EdgePreGPPT | 85.62 $_{\pm1.41}$ | 72.39 $_{\pm3.60}$ | 83.03 $_{\pm2.89}$ | 66.70 $_{\pm4.04}$ | 54.49 $_{\pm12.28}$ | 52.25 $_{\pm0.89}$ | 60.55 $_{\pm1.73}$ |
|  | EdgePredGprompt | 85.85 $_{\pm0.79}$ | 76.73 $_{\pm2.26}$ | 83.40 $_{\pm2.74}$ | 67.27 $_{\pm4.41}$ | 58.43 $_{\pm10.76}$ | 51.24 $_{\pm0.96}$ | 61.30 $_{\pm1.07}$ |
|  | GCL | 87.71 $_{\pm1.59}$ | 75.10 $_{\pm2.66}$ | 81.83 $_{\pm2.15}$ | 67.86 $_{\pm7.86}$ | 57.10 $_{\pm12.53}$ | 52.03 $_{\pm1.09}$ | 54.66 $_{\pm2.69}$ |
|  | SimGRACE | 80.39 $_{\pm2.84}$ | 76.94 $_{\pm1.30}$ | 79.56 $_{\pm3.24}$ | 67.34 $_{\pm5.82}$ | 59.02 $_{\pm11.26}$ | 51.60 $_{\pm0.77}$ | 56.28 $_{\pm1.14}$ |
| GPPTPrompt | DGI | 70.88 $_{\pm2.54}$ | 73.97 $_{\pm5.46}$ | 61.07 $_{\pm2.93}$ | 66.79 $_{\pm2.66}$ | 63.59 $_{\pm7.91}$ | 51.34 $_{\pm1.54}$ | 50.34 $_{\pm0.68}$ |
|  | GraphMAE | 80.91 $_{\pm2.38}$ | 69.11 $_{\pm3.46}$ | 83.10 $_{\pm1.03}$ | 63.77 $_{\pm3.61}$ | 60.95 $_{\pm3.13}$ | 51.22 $_{\pm1.38}$ | 51.84 $_{\pm3.86}$ |
|  | EdgePreGPPT | 68.71 $_{\pm2.02}$ | 59.01 $_{\pm2.73}$ | 74.22 $_{\pm1.78}$ | 57.60 $_{\pm4.72}$ | 58.20 $_{\pm4.72}$ | 50.23 $_{\pm0.31}$ | 58.95 $_{\pm1.29}$ |
|  | EdgePreGprompt | 68.29 $_{\pm3.52}$ | 58.07 $_{\pm2.40}$ | 81.15 $_{\pm4.81}$ | 61.93 $_{\pm5.66}$ | 61.91 $_{\pm5.28}$ | 50.27 $_{\pm0.63}$ | 56.16 $_{\pm2.47}$ |
|  | GCL | 75.26 $_{\pm3.86}$ | 60.58 $_{\pm4.04}$ | 71.43 $_{\pm2.47}$ | 53.79 $_{\pm2.21}$ | 61.73 $_{\pm5.50}$ | 50.96 $_{\pm0.66}$ | 50.69 $_{\pm1.68}$ |
|  | SimGRACE | 66.90 $_{\pm4.18}$ | 65.72 $_{\pm2.09}$ | 62.58 $_{\pm2.08}$ | 66.87 $_{\pm2.38}$ | 60.35 $_{\pm7.62}$ | 50.53 $_{\pm0.57}$ | 50.07 $_{\pm0.24}$ |
| ALL-in-one | DGI | 71.15 $_{\pm3.33}$ | 65.99 $_{\pm0.90}$ | 66.83 $_{\pm1.02}$ | 81.86 $_{\pm0.82}$ | 46.05 $_{\pm2.66}$ | 52.80 $_{\pm2.23}$ | 55.31 $_{\pm0.33}$ |
|  | GraphMAE | 77.89 $_{\pm3.94}$ | 63.31 $_{\pm2.05}$ | 84.13 $_{\pm1.99}$ | 81.57 $_{\pm1.86}$ | 66.49 $_{\pm5.26}$ | 50.76 $_{\pm1.17}$ | 55.43 $_{\pm0.07}$ |
|  | EdgePreGPPT | 80.90 $_{\pm2.72}$ | 68.74 $_{\pm3.28}$ | 76.23 $_{\pm0.81}$ | 96.17 $_{\pm3.82}$ | 75.39 $_{\pm1.30}$ | 52.98 $_{\pm1.21}$ | 56.37 $_{\pm0.72}$ |
|  | EdgePreGprompt | 77.50 $_{\pm2.96}$ | 69.84 $_{\pm1.80}$ | 85.03 $_{\pm1.73}$ | 91.35 $_{\pm2.44}$ | 72.04 $_{\pm6.05}$ | 49.89 $_{\pm1.95}$ | 54.08 $_{\pm0.15}$ |
|  | GCL | 65.18 $_{\pm11.80}$ | 77.64 $_{\pm2.06}$ | 73.05 $_{\pm1.97}$ | 82.74 $_{\pm3.64}$ | 72.60 $_{\pm1.20}$ | 51.67 $_{\pm1.31}$ | 53.03 $_{\pm0.27}$ |
|  | SimGRACE | 59.44 $_{\pm9.08}$ | 78.11 $_{\pm1.17}$ | 64.76 $_{\pm2.62}$ | 84.99 $_{\pm2.40}$ | 92.99 $_{\pm1.95}$ | 51.64 $_{\pm2.25}$ | 51.35 $_{\pm0.09}$ |
| Gprompt | DGI | 62.24 $_{\pm2.99}$ | 75.87 $_{\pm3.70}$ | 55.46 $_{\pm1.89}$ | 87.72 $_{\pm5.97}$ | 49.44 $_{\pm8.34}$ | 55.78 $_{\pm3.19}$ | 52.89 $_{\pm2.11}$ |
|  | GraphMAE | 80.27 $_{\pm5.61}$ | 74.38 $_{\pm3.36}$ | 84.37 $_{\pm1.75}$ | 96.34 $_{\pm2.81}$ | 49.44 $_{\pm8.34}$ | 58.52 $_{\pm2.66}$ | 53.56 $_{\pm1.76}$ |
|  | EdgePreGPPT | 81.41 $_{\pm5.40}$ | 65.99 $_{\pm4.06}$ | 72.84 $_{\pm0.48}$ | 98.59 $_{\pm1.16}$ | 49.44 $_{\pm8.34}$ | 56.86 $_{\pm2.69}$ | 56.05 $_{\pm3.09}$ |
|  | EdgePreGprompt | 83.15 $_{\pm3.91}$ | 82.01 $_{\pm1.78}$ | 84.45 $_{\pm1.80}$ | 99.32 $_{\pm0.66}$ | 49.44 $_{\pm8.34}$ | 51.33 $_{\pm2.32}$ | 54.94 $_{\pm1.15}$ |
|  | GCL | 83.25 $_{\pm3.22}$ | 83.20 $_{\pm0.97}$ | 71.35 $_{\pm3.39}$ | 91.20 $_{\pm4.59}$ | 49.44 $_{\pm8.34}$ | 59.78 $_{\pm3.41}$ | 53.58 $_{\pm1.71}$ |
|  | SimGRACE | 79.02 $_{\pm2.55}$ | 86.57 $_{\pm1.86}$ | 61.89 $_{\pm6.64}$ | 94.09 $_{\pm3.00}$ | 49.44 $_{\pm8.34}$ | 56.13 $_{\pm1.59}$ | 52.64 $_{\pm1.18}$ |
| GPF | DGI | 60.20 $_{\pm13.06}$ | 59.79 $_{\pm13.56}$ | 59.52 $_{\pm12.99}$ | 84.92 $_{\pm3.03}$ | 71.79 $_{\pm11.35}$ | 54.39 $_{\pm2.39}$ | 55.76 $_{\pm1.34}$ |
|  | GraphMAE | 82.70 $_{\pm2.62}$ | 72.25 $_{\pm4.60}$ | 87.91 $_{\pm1.01}$ | 99.59 $_{\pm0.81}$ | 91.54 $_{\pm9.54}$ | 67.24 $_{\pm4.73}$ | 55.48 $_{\pm0.92}$ |
|  | EdgePreGPPT | 65.44 $_{\pm9.07}$ | 67.13 $_{\pm1.95}$ | 81.07 $_{\pm0.78}$ | 99.62 $_{\pm0.64}$ | 84.29 $_{\pm6.85}$ | 58.43 $_{\pm3.84}$ | 58.43 $_{\pm1.11}$ |
|  | EdgePreGprompt | 83.18 $_{\pm3.22}$ | 69.63 $_{\pm7.04}$ | 85.38 $_{\pm6.75}$ | 99.56 $_{\pm0.58}$ | 87.93 $_{\pm6.62}$ | 56.19 $_{\pm3.18}$ | 54.51 $_{\pm2.74}$ |
|  | GCL | 73.76 $_{\pm9.73}$ | 68.34 $_{\pm10.91}$ | 75.10 $_{\pm2.04}$ | 52.13 $_{\pm1.68}$ | 87.77 $_{\pm8.39}$ | 56.96 $_{\pm4.69}$ | 51.48 $_{\pm0.80}$ |
|  | SimGRACE | 63.96 $_{\pm7.80}$ | 63.26 $_{\pm11.65}$ | 58.71 $_{\pm6.49}$ | 95.66 $_{\pm0.57}$ | 95.66 $_{\pm9.06}$ | 55.06 $_{\pm3.04}$ | 49.92 $_{\pm0.31}$ |
| GPF-plus | DGI | 55.30 $_{\pm13.20}$ | 55.81 $_{\pm15.74}$ | 66.54 $_{\pm15.31}$ | 93.25 $_{\pm5.92}$ | 77.44 $_{\pm7.96}$ | 57.09 $_{\pm3.17}$ | 55.62 $_{\pm1.15}$ |
|  | GraphMAE | 87.26 $_{\pm3.14}$ | 91.53 $_{\pm1.55}$ | 87.12 $_{\pm1.71}$ | 99.99 $_{\pm0.02}$ | 86.23 $_{\pm6.11}$ | 72.39 $_{\pm1.82}$ | 58.01 $_{\pm1.44}$ |
|  | EdgePreGPPT | 73.87 $_{\pm10.93}$ | 56.16 $_{\pm16.39}$ | 82.80 $_{\pm1.90}$ | 99.75 $_{\pm0.50}$ | 87.25 $_{\pm8.99}$ | 67.15 $_{\pm3.76}$ | 57.48 $_{\pm1.34}$ |
|  | EdgePreGprompt | 85.81 $_{\pm3.03}$ | 91.47 $_{\pm1.96}$ | 85.13 $_{\pm2.57}$ | 99.73 $_{\pm0.51}$ | 88.28 $_{\pm9.02}$ | 65.17 $_{\pm4.06}$ | 56.91 $_{\pm1.33}$ |
|  | GCL | 80.93 $_{\pm6.55}$ | 61.66 $_{\pm15.01}$ | 80.00 $_{\pm7.05}$ | 78.70 $_{\pm14.44}$ | 81.41 $_{\pm2.06}$ | 69.40 $_{\pm1.84}$ | 51.15 $_{\pm0.58}$ |
|  | SimGRACE | 66.36 $_{\pm16.56}$ | 60.91 $_{\pm15.05}$ | 61.66 $_{\pm12.28}$ | 90.60 $_{\pm4.75}$ | 91.16 $_{\pm9.14}$ | 66.44 $_{\pm4.39}$ | 49.94 $_{\pm1.04}$ |

Table 17: Node classification performance Accuracy (%) (5-shot)

| Training schemes | Methods | Cora | CiteSeer | Pubmed | Wisconsin | Texas | Actor | Ogbn-arxiv |
|---|---|---|---|---|---|---|---|---|
| self-supervise | GCN | 50.25 $_{\pm8.37}$ | 41.22 $_{\pm6.30}$ | 67.88 $_{\pm2.18}$ | 39.43 $_{\pm5.86}$ | 43.91 $_{\pm6.47}$ | 21.92 $_{\pm1.86}$ | 22.38 $_{\pm3.05}$ |
| Pretrain + Fine-tuning | DGI | 48.79 $_{\pm8.51}$ | 35.91 $_{\pm4.94}$ | 61.44 $_{\pm7.93}$ | 40.00 $_{\pm6.31}$ | 47.19 $_{\pm7.37}$ | 21.48 $_{\pm1.71}$ | 12.48 $_{\pm6.15}$ |
|  | GraphMAE | 54.09 $_{\pm4.21}$ | 32.75 $_{\pm6.35}$ | 70.04 $_{\pm4.57}$ | 40.69 $_{\pm6.73}$ | 43.44 $_{\pm8.48}$ | 22.92 $_{\pm1.22}$ | 24.08 $_{\pm1.45}$ |
|  | EdgePreGPPT | 54.34 $_{\pm3.78}$ | 38.97 $_{\pm7.35}$ | 70.91 $_{\pm4.87}$ | 40.69 $_{\pm6.77}$ | 44.53 $_{\pm11.05}$ | 21.46 $_{\pm1.00}$ | 28.84 $_{\pm3.11}$ |
|  | EdgePredGprompt | 49.04 $_{\pm8.99}$ | 32.08 $_{\pm8.24}$ | 70.44 $_{\pm5.04}$ | 41.03 $_{\pm6.58}$ | 45.78 $_{\pm8.07}$ | 22.46 $_{\pm1.99}$ | 25.94 $_{\pm2.63}$ |
|  | GCL | 62.66 $_{\pm3.55}$ | 39.54 $_{\pm3.54}$ | 66.84 $_{\pm5.78}$ | 42.97 $_{\pm8.99}$ | 46.41 $_{\pm5.47}$ | 21.99 $_{\pm1.61}$ | 13.42 $_{\pm3.73}$ |
|  | SimGRACE | 45.13 $_{\pm7.81}$ | 38.90 $_{\pm6.29}$ | 62.65 $_{\pm6.29}$ | 41.49 $_{\pm5.77}$ | 46.09 $_{\pm8.71}$ | 22.77 $_{\pm1.56}$ | 9.45 $_{\pm1.62}$ |
| GPPTPrompt | DGI | 43.68 $_{\pm7.12}$ | 45.77 $_{\pm7.41}$ | 47.39 $_{\pm10.22}$ | 36.29 $_{\pm3.97}$ | 48.82 $_{\pm5.15}$ | 20.91 $_{\pm1.36}$ | 8.85 $_{\pm6.21}$ |
|  | GraphMAE | 31.50 $_{\pm11.89}$ | 19.93 $_{\pm6.60}$ | 66.97 $_{\pm3.70}$ | 37.00 $_{\pm3.19}$ | 42.68 $_{\pm6.91}$ | 22.10 $_{\pm0.86}$ | 8.90 $_{\pm6.25}$ |
|  | EdgePreGPPT | 32.94 $_{\pm2.82}$ | 26.93 $_{\pm1.56}$ | 63.45 $_{\pm3.63}$ | 34.00 $_{\pm5.69}$ | 36.85 $_{\pm3.24}$ | 20.42 $_{\pm0.81}$ | 28.90 $_{\pm1.64}$ |
|  | EdgePreGprompt | 30.55 $_{\pm4.59}$ | 28.46 $_{\pm4.25}$ | 63.33 $_{\pm4.18}$ | 32.57 $_{\pm3.31}$ | 39.21 $_{\pm6.09}$ | 20.07 $_{\pm0.45}$ | 4.01 $_{\pm0.69}$ |
|  | GCL | 51.98 $_{\pm3.43}$ | 26.68 $_{\pm2.95}$ | 57.59 $_{\pm2.89}$ | 30.43 $_{\pm8.83}$ | 45.20 $_{\pm2.20}$ | 21.25 $_{\pm0.78}$ | 14.15 $_{\pm5.26}$ |
|  | SimGRACE | 34.93 $_{\pm2.55}$ | 29.51 $_{\pm2.60}$ | 42.87 $_{\pm1.57}$ | 33.00 $_{\pm5.27}$ | 41.42 $_{\pm6.80}$ | 21.58 $_{\pm0.84}$ | 9.11 $_{\pm3.50}$ |
| ALL-in-one | DGI | 30.45 $_{\pm0.19}$ | 21.82 $_{\pm4.73}$ | 43.60 $_{\pm5.24}$ | 79.95 $_{\pm4.01}$ | 62.50 $_{\pm12.39}$ | 19.83 $_{\pm3.14}$ | 2.65 $_{\pm0.09}$ |
|  | GraphMAE | 25.02 $_{\pm7.58}$ | 19.38 $_{\pm4.38}$ | 46.16 $_{\pm15.83}$ | 82.43 $_{\pm2.84}$ | 66.47 $_{\pm17.32}$ | 15.03 $_{\pm3.13}$ | 0.24 $_{\pm0.00}$ |
|  | EdgePreGPPT | 30.36 $_{\pm13.48}$ | 25.83 $_{\pm9.32}$ | 37.61 $_{\pm20.58}$ | 88.55 $_{\pm3.52}$ | 47.98 $_{\pm27.78}$ | 21.49 $_{\pm3.02}$ | 1.59 $_{\pm1.30}$ |
|  | EdgePreGprompt | 21.59 $_{\pm7.30}$ | 18.75 $_{\pm1.86}$ | 39.36 $_{\pm11.45}$ | 87.16 $_{\pm3.02}$ | 46.32 $_{\pm29.89}$ | 21.23 $_{\pm5.64}$ | 13.01 $_{\pm6.29}$ |
|  | GCL | 25.63 $_{\pm17.68}$ | 27.93 $_{\pm10.59}$ | 42.86 $_{\pm7.21}$ | 84.75 $_{\pm8.86}$ | 73.28 $_{\pm9.91}$ | 20.83 $_{\pm8.04}$ | 6.70 $_{\pm6.01}$ |
|  | SimGRACE | 15.20 $_{\pm9.51}$ | 25.19 $_{\pm12.95}$ | 21.59 $_{\pm2.58}$ | 78.90 $_{\pm2.26}$ | 64.63 $_{\pm25.59}$ | 21.26 $_{\pm6.24}$ | 4.80 $_{\pm5.17}$ |
| Gprompt | DGI | 27.81 $_{\pm10.28}$ | 52.84 $_{\pm2.36}$ | 37.83 $_{\pm5.71}$ | 57.24 $_{\pm27.27}$ | 34.47 $_{\pm41.15}$ | 26.07 $_{\pm4.20}$ | 56.59 $_{\pm4.05}$ |
|  | GraphMAE | 66.82 $_{\pm3.98}$ | 60.07 $_{\pm3.98}$ | 66.66 $_{\pm2.25}$ | 74.89 $_{\pm35.83}$ | 36.73 $_{\pm43.95}$ | 30.71 $_{\pm2.12}$ | 76.98 $_{\pm3.88}$ |
|  | EdgePreGPPT | 53.15 $_{\pm2.85}$ | 36.15 $_{\pm5.00}$ | 62.18 $_{\pm4.88}$ | 78.22 $_{\pm37.33}$ | 37.59 $_{\pm45.03}$ | 28.76 $_{\pm3.95}$ | 85.40 $_{\pm0.79}$ |
|  | EdgePreGprompt | 62.63 $_{\pm3.26}$ | 65.34 $_{\pm4.60}$ | 67.87 $_{\pm2.08}$ | 78.01 $_{\pm37.38}$ | 38.45 $_{\pm46.03}$ | 21.86 $_{\pm2.13}$ | 81.85 $_{\pm3.16}$ |
|  | GCL | 69.03 $_{\pm3.61}$ | 61.27 $_{\pm5.37}$ | 57.21 $_{\pm2.78}$ | 65.72 $_{\pm31.30}$ | 36.38 $_{\pm43.47}$ | 34.67 $_{\pm1.28}$ | 57.52 $_{\pm4.19}$ |
|  | SimGRACE | 51.27 $_{\pm6.05}$ | 66.13 $_{\pm1.64}$ | 44.78 $_{\pm6.52}$ | 68.34 $_{\pm32.42}$ | 39.32 $_{\pm47.08}$ | 30.06 $_{\pm3.36}$ | 57.36 $_{\pm3.68}$ |
| GPF | DGI | 29.57 $_{\pm20.89}$ | 24.55 $_{\pm12.61}$ | 52.43 $_{\pm6.94}$ | 83.21 $_{\pm5.19}$ | 89.30 $_{\pm1.58}$ | 35.40 $_{\pm8.78}$ | 20.72 $_{\pm0.26}$ |
|  | GraphMAE | 35.43 $_{\pm1.02}$ | 25.12 $_{\pm3.01}$ | 68.96 $_{\pm3.99}$ | 96.30 $_{\pm5.12}$ | 92.44 $_{\pm3.04}$ | 44.07 $_{\pm3.94}$ | 45.20 $_{\pm2.03}$ |
|  | EdgePreGPPT | 15.39 $_{\pm8.19}$ | 18.11 $_{\pm2.78}$ | 58.67 $_{\pm3.21}$ | 98.26 $_{\pm1.19}$ | 89.47 $_{\pm1.91}$ | 30.99 $_{\pm4.68}$ | 71.83 $_{\pm9.37}$ |
|  | EdgePreGprompt | 31.58 $_{\pm18.16}$ | 24.55 $_{\pm8.28}$ | 66.25 $_{\pm8.53}$ | 96.30 $_{\pm5.12}$ | 90.18 $_{\pm1.57}$ | 31.92 $_{\pm5.75}$ | 28.60 $_{\pm3.11}$ |
|  | GCL | 28.60 $_{\pm11.19}$ | 17.69 $_{\pm1.35}$ | 52.47 $_{\pm6.73}$ | 69.15 $_{\pm21.44}$ | 98.42 $_{\pm0.36}$ | 29.13 $_{\pm2.63}$ | 23.90 $_{\pm0.12}$ |
|  | SimGRACE | 18.94 $_{\pm12.60}$ | 22.45 $_{\pm3.45}$ | 40.35 $_{\pm1.71}$ | 92.29 $_{\pm3.36}$ | 91.73 $_{\pm3.84}$ | 30.28 $_{\pm2.64}$ | 33.77 $_{\pm8.48}$ |
| GPF-plus | DGI | 27.23 $_{\pm14.61}$ | 26.31 $_{\pm11.48}$ | 47.02 $_{\pm14.51}$ | 83.86 $_{\pm17.24}$ | 96.18 $_{\pm4.12}$ | 36.03 $_{\pm7.49}$ | 16.86 $_{\pm3.30}$ |
|  | GraphMAE | 63.28 $_{\pm4.69}$ | 75.73 $_{\pm2.19}$ | 69.59 $_{\pm4.33}$ | 99.01 $_{\pm1.43}$ | 99.12 $_{\pm0.95}$ | 44.58 $_{\pm5.95}$ | 47.79 $_{\pm1.09}$ |
|  | EdgePreGPPT | 22.44 $_{\pm23.88}$ | 13.63 $_{\pm5.27}$ | 66.43 $_{\pm3.28}$ | 98.52 $_{\pm2.07}$ | 96.18 $_{\pm4.12}$ | 37.15 $_{\pm8.48}$ | 66.88 $_{\pm6.14}$ |
|  | EdgePreGprompt | 66.22 $_{\pm6.20}$ | 64.49 $_{\pm14.12}$ | 68.10 $_{\pm4.56}$ | 98.64 $_{\pm2.14}$ | 97.74 $_{\pm2.47}$ | 41.98 $_{\pm4.70}$ | 51.25 $_{\pm1.31}$ |
|  | GCL | 47.71 $_{\pm22.44}$ | 29.16 $_{\pm16.66}$ | 64.53 $_{\pm4.15}$ | 70.98 $_{\pm19.27}$ | 99.12 $_{\pm0.95}$ | 36.99 $_{\pm6.59}$ | 25.74 $_{\pm1.70}$ |
|  | SimGRACE | 27.79 $_{\pm21.51}$ | 28.37 $_{\pm22.54}$ | 50.25 $_{\pm10.04}$ | 89.57 $_{\pm4.22}$ | 95.83 $_{\pm3.85}$ | 38.23 $_{\pm2.34}$ | 44.70 $_{\pm2.69}$ |

Table 18: Node classification performance F1-score (5-shot)

| Training schemes | Methods | Cora | CiteSeer | Pubmed | Wisconsin | Texas | Actor | Ogbn-arxiv |
|---|---|---|---|---|---|---|---|---|
| self-supervise | GCN | $51.42_{\pm8.70}$ | $34.20_{\pm6.12}$ | $67.45_{\pm2.25}$ | $34.31_{\pm4.41}$ | $35.02_{\pm8.10}$ | $16.89_{\pm1.20}$ | $14.76_{\pm1.13}$ |
| Pretrain + Fine-tuning | DGI | $47.53_{\pm9.44}$ | $28.71_{\pm2.76}$ | $59.33_{\pm10.23}$ | $34.71_{\pm5.28}$ | $38.80_{\pm7.04}$ | $16.49_{\pm1.42}$ | $4.78_{\pm2.53}$ |
| | GraphMAE | $55.17_{\pm5.27}$ | $29.67_{\pm6.94}$ | $69.58_{\pm4.63}$ | $34.69_{\pm5.63}$ | $34.57_{\pm9.23}$ | $16.92_{\pm1.76}$ | $12.97_{\pm0.32}$ |
| | EdgePreGPPT | $55.50_{\pm3.62}$ | $31.31_{\pm8.86}$ | $70.31_{\pm4.65}$ | $34.86_{\pm4.83}$ | $36.63_{\pm7.69}$ | $19.42_{\pm1.02}$ | $20.19_{\pm1.27}$ |
| | EdgePredGprompt | $49.18_{\pm8.50}$ | $27.04_{\pm6.83}$ | $69.46_{\pm4.79}$ | $34.90_{\pm5.77}$ | $36.38_{\pm7.04}$ | $18.28_{\pm1.91}$ | $18.51_{\pm1.18}$ |
| | GCL | $61.32_{\pm3.43}$ | $35.95_{\pm3.18}$ | $66.16_{\pm5.43}$ | $33.81_{\pm3.28}$ | $26.66_{\pm5.78}$ | $20.44_{\pm1.45}$ | $6.96_{\pm1.07}$ |
| | SimGRACE | $45.33_{\pm4.88}$ | $34.74_{\pm7.40}$ | $62.66_{\pm6.39}$ | $35.86_{\pm4.86}$ | $36.86_{\pm7.72}$ | $20.63_{\pm1.13}$ | $4.74_{\pm0.48}$ |
| GPPTPrompt | DGI | $40.79_{\pm7.28}$ | $39.86_{\pm7.94}$ | $44.67_{\pm12.25}$ | $32.76_{\pm3.66}$ | $39.86_{\pm4.07}$ | $19.28_{\pm0.85}$ | $0.39_{\pm0.27}$ |
| | GraphMAE | $24.55_{\pm7.97}$ | $7.49_{\pm4.08}$ | $66.55_{\pm3.62}$ | $33.86_{\pm3.88}$ | $36.66_{\pm2.42}$ | $19.92_{\pm0.84}$ | $0.42_{\pm0.29}$ |
| | EdgePreGPPT | $32.50_{\pm2.42}$ | $19.95_{\pm1.98}$ | $62.42_{\pm4.78}$ | $30.57_{\pm4.59}$ | $33.93_{\pm2.76}$ | $19.89_{\pm0.56}$ | $18.11_{\pm1.03}$ |
| | EdgePredGprompt | $22.20_{\pm3.24}$ | $25.96_{\pm4.33}$ | $61.22_{\pm5.46}$ | $30.39_{\pm2.77}$ | $35.84_{\pm3.35}$ | $19.22_{\pm0.33}$ | $2.56_{\pm1.55}$ |
| | GCL | $50.60_{\pm2.51}$ | $25.00_{\pm2.05}$ | $57.27_{\pm2.84}$ | $18.78_{\pm5.00}$ | $35.83_{\pm1.96}$ | $20.34_{\pm0.74}$ | $3.03_{\pm1.03}$ |
| | SimGRACE | $31.56_{\pm2.75}$ | $28.27_{\pm1.68}$ | $40.87_{\pm2.90}$ | $27.06_{\pm3.81}$ | $36.11_{\pm2.81}$ | $20.45_{\pm0.51}$ | $1.63_{\pm0.40}$ |
| ALL-in-one | DGI | $6.67_{\pm0.03}$ | $10.68_{\pm7.84}$ | $26.81_{\pm9.85}$ | $60.73_{\pm6.34}$ | $31.59_{\pm18.97}$ | $10.66_{\pm3.88}$ | $0.21_{\pm0.04}$ |
| | GraphMAE | $6.77_{\pm3.23}$ | $8.66_{\pm4.25}$ | $33.97_{\pm19.42}$ | $58.02_{\pm3.64}$ | $49.27_{\pm9.03}$ | $8.31_{\pm2.80}$ | $0.01_{\pm0.00}$ |
| | EdgePreGPPT | $15.19_{\pm16.25}$ | $13.46_{\pm10.23}$ | $31.47_{\pm24.42}$ | $78.97_{\pm11.06}$ | $29.95_{\pm20.92}$ | $10.10_{\pm3.63}$ | $0.53_{\pm0.63}$ |
| | EdgePredGprompt | $4.99_{\pm1.38}$ | $5.28_{\pm0.47}$ | $27.49_{\pm15.02}$ | $68.69_{\pm3.29}$ | $25.44_{\pm17.47}$ | $8.08_{\pm2.12}$ | $0.56_{\pm0.27}$ |
| | GCL | $14.88_{\pm18.86}$ | $14.98_{\pm12.03}$ | $31.71_{\pm13.56}$ | $74.33_{\pm11.49}$ | $49.70_{\pm21.87}$ | $10.10_{\pm6.01}$ | $0.75_{\pm1.07}$ |
| | SimGRACE | $7.21_{\pm7.14}$ | $13.71_{\pm14.05}$ | $13.42_{\pm3.26}$ | $66.26_{\pm6.68}$ | $45.47_{\pm21.83}$ | $11.42_{\pm5.14}$ | $0.52_{\pm0.70}$ |
| Gprompt | DGI | $19.96_{\pm9.68}$ | $47.84_{\pm2.39}$ | $37.26_{\pm5.45}$ | $48.86_{\pm23.99}$ | $29.50_{\pm36.22}$ | $24.02_{\pm4.65}$ | $49.69_{\pm1.68}$ |
| | GraphMAE | $61.57_{\pm5.38}$ | $56.39_{\pm3.60}$ | $65.31_{\pm2.36}$ | $70.18_{\pm34.72}$ | $36.05_{\pm43.75}$ | $30.08_{\pm1.88}$ | $66.82_{\pm2.68}$ |
| | EdgePreGPPT | $51.03_{\pm3.08}$ | $31.15_{\pm3.86}$ | $60.74_{\pm4.30}$ | $73.65_{\pm36.66}$ | $34.07_{\pm41.83}$ | $27.29_{\pm3.43}$ | $79.73_{\pm1.29}$ |
| | EdgePredGprompt | $60.77_{\pm3.39}$ | $63.42_{\pm5.11}$ | $66.46_{\pm2.30}$ | $77.08_{\pm37.97}$ | $34.45_{\pm41.99}$ | $21.03_{\pm2.04}$ | $79.20_{\pm1.95}$ |
| | GCL | $64.22_{\pm4.08}$ | $57.65_{\pm5.74}$ | $56.66_{\pm2.35}$ | $51.82_{\pm25.62}$ | $31.38_{\pm38.22}$ | $32.99_{\pm1.00}$ | $49.82_{\pm2.81}$ |
| | SimGRACE | $45.98_{\pm5.22}$ | $63.84_{\pm1.37}$ | $42.20_{\pm5.64}$ | $62.12_{\pm30.55}$ | $38.47_{\pm46.70}$ | $29.63_{\pm3.89}$ | $53.41_{\pm2.08}$ |
| GPF | DGI | $18.44_{\pm18.83}$ | $13.25_{\pm11.04}$ | $47.23_{\pm9.28}$ | $75.78_{\pm14.28}$ | $77.01_{\pm10.57}$ | $28.58_{\pm14.86}$ | $11.02_{\pm0.99}$ |
| | GraphMAE | $23.17_{\pm2.53}$ | $12.78_{\pm1.23}$ | $67.88_{\pm4.07}$ | $87.75_{\pm13.24}$ | $82.81_{\pm10.55}$ | $43.09_{\pm4.07}$ | $16.76_{\pm2.76}$ |
| | EdgePreGPPT | $3.69_{\pm1.67}$ | $5.09_{\pm0.66}$ | $53.91_{\pm7.41}$ | $94.37_{\pm8.31}$ | $80.86_{\pm9.83}$ | $23.04_{\pm9.22}$ | $55.01_{\pm8.05}$ |
| | EdgePredGprompt | $20.17_{\pm13.79}$ | $12.79_{\pm6.88}$ | $62.47_{\pm11.25}$ | $89.10_{\pm11.52}$ | $79.88_{\pm10.50}$ | $30.44_{\pm9.87}$ | $18.77_{\pm3.29}$ |
| | GCL | $11.74_{\pm6.74}$ | $5.99_{\pm2.19}$ | $46.46_{\pm14.28}$ | $51.01_{\pm22.81}$ | $66.44_{\pm6.95}$ | $22.59_{\pm5.87}$ | $18.74_{\pm0.78}$ |
| | SimGRACE | $10.17_{\pm12.35}$ | $9.91_{\pm2.56}$ | $29.36_{\pm8.75}$ | $71.38_{\pm2.97}$ | $81.14_{\pm11.49}$ | $22.30_{\pm8.60}$ | $12.31_{\pm4.13}$ |
| GPF-plus | DGI | $13.92_{\pm14.65}$ | $12.41_{\pm13.45}$ | $40.73_{\pm16.83}$ | $71.22_{\pm13.69}$ | $84.85_{\pm10.28}$ | $32.05_{\pm9.40}$ | $6.42_{\pm2.47}$ |
| | GraphMAE | $61.55_{\pm4.90}$ | $74.86_{\pm2.27}$ | $68.34_{\pm4.25}$ | $94.48_{\pm8.79}$ | $86.93_{\pm10.70}$ | $44.28_{\pm6.46}$ | $24.14_{\pm2.16}$ |
| | EdgePreGPPT | $15.91_{\pm26.40}$ | $4.26_{\pm1.11}$ | $61.79_{\pm2.65}$ | $93.18_{\pm10.98}$ | $82.10_{\pm9.29}$ | $35.11_{\pm10.76}$ | $56.15_{\pm4.28}$ |
| | EdgePredGprompt | $65.96_{\pm5.07}$ | $57.16_{\pm18.91}$ | $66.11_{\pm4.85}$ | $93.46_{\pm10.79}$ | $85.56_{\pm9.53}$ | $39.84_{\pm6.02}$ | $32.07_{\pm2.05}$ |
| | GCL | $38.84_{\pm26.82}$ | $18.53_{\pm21.72}$ | $63.06_{\pm3.88}$ | $65.13_{\pm17.26}$ | $86.59_{\pm11.04}$ | $33.23_{\pm7.80}$ | $19.46_{\pm1.47}$ |
| | SimGRACE | $19.60_{\pm23.99}$ | $18.27_{\pm26.81}$ | $36.93_{\pm16.65}$ | $69.27_{\pm9.83}$ | $84.63_{\pm10.23}$ | $35.78_{\pm3.00}$ | $24.02_{\pm2.29}$ |

Table 19: Node classification performance AUROC (5-shot)

| Training schemes | Methods | Cora | CiteSeer | Pubmed | Wisconsin | Texas | Actor | Ogbn-arxiv |
|---|---|---|---|---|---|---|---|---|
| self-supervise | GCN | $88.39_{\pm2.56}$ | $78.16_{\pm2.95}$ | $84.93_{\pm2.77}$ | $65.96_{\pm5.03}$ | $66.32_{\pm8.89}$ | $51.44_{\pm0.62}$ | $78.83_{\pm1.37}$ |
| Pretrain + Fine-tuning | DGI | $84.60_{\pm5.40}$ | $74.29_{\pm3.78}$ | $81.26_{\pm3.74}$ | $67.40_{\pm4.97}$ | $66.62_{\pm8.12}$ | $51.78_{\pm0.48}$ | $62.86_{\pm4.35}$ |
| | GraphMAE | $90.30_{\pm1.62}$ | $77.04_{\pm3.75}$ | $86.09_{\pm2.54}$ | $65.32_{\pm5.26}$ | $69.36_{\pm9.63}$ | $51.73_{\pm0.46}$ | $75.65_{\pm0.78}$ |
| | EdgePreGPPT | $86.88_{\pm1.90}$ | $75.17_{\pm2.72}$ | $86.23_{\pm2.32}$ | $67.71_{\pm3.54}$ | $69.36_{\pm8.52}$ | $51.34_{\pm0.65}$ | $83.86_{\pm1.25}$ |
| | EdgePredGprompt | $87.41_{\pm2.15}$ | $73.79_{\pm5.07}$ | $85.66_{\pm3.26}$ | $66.04_{\pm5.12}$ | $66.70_{\pm8.92}$ | $52.40_{\pm0.91}$ | $82.34_{\pm1.45}$ |
| | GCL | $90.49_{\pm1.49}$ | $76.54_{\pm1.56}$ | $83.07_{\pm3.73}$ | $57.71_{\pm8.14}$ | $65.57_{\pm7.78}$ | $52.68_{\pm0.69}$ | $66.70_{\pm0.95}$ |
| | SimGRACE | $85.45_{\pm2.29}$ | $76.95_{\pm2.79}$ | $80.86_{\pm3.17}$ | $67.73_{\pm4.47}$ | $66.11_{\pm7.69}$ | $52.52_{\pm0.92}$ | $60.36_{\pm0.72}$ |
| GPPTPrompt | DGI | $76.38_{\pm6.02}$ | $76.59_{\pm3.18}$ | $63.76_{\pm10.00}$ | $69.27_{\pm1.49}$ | $66.52_{\pm3.33}$ | $51.27_{\pm1.04}$ | $50.09_{\pm0.25}$ |
| | GraphMAE | $84.77_{\pm3.18}$ | $69.69_{\pm1.62}$ | $82.56_{\pm1.84}$ | $66.53_{\pm4.44}$ | $63.43_{\pm2.99}$ | $50.92_{\pm0.44}$ | $63.55_{\pm2.93}$ |
| | EdgePreGPPT | $69.44_{\pm2.15}$ | $60.14_{\pm1.47}$ | $77.28_{\pm3.58}$ | $62.04_{\pm3.27}$ | $61.22_{\pm2.13}$ | $50.85_{\pm0.37}$ | $85.67_{\pm0.67}$ |
| | EdgePredGprompt | $69.60_{\pm0.83}$ | $62.44_{\pm2.17}$ | $81.98_{\pm4.11}$ | $63.83_{\pm2.92}$ | $65.86_{\pm2.34}$ | $50.32_{\pm0.54}$ | $71.01_{\pm5.38}$ |
| | GCL | $82.06_{\pm2.24}$ | $61.54_{\pm2.42}$ | $76.26_{\pm1.36}$ | $55.23_{\pm2.54}$ | $62.83_{\pm4.62}$ | $50.87_{\pm0.69}$ | $58.53_{\pm1.53}$ |
| | SimGRACE | $70.54_{\pm2.07}$ | $64.49_{\pm1.54}$ | $60.67_{\pm1.80}$ | $66.38_{\pm0.42}$ | $62.69_{\pm1.92}$ | $51.00_{\pm0.54}$ | $51.46_{\pm0.26}$ |
| ALL-in-one | DGI | $42.65_{\pm0.41}$ | $57.45_{\pm7.01}$ | $57.38_{\pm8.87}$ | $85.53_{\pm2.93}$ | $68.87_{\pm6.32}$ | $51.24_{\pm2.32}$ | $54.76_{\pm0.75}$ |
| | GraphMAE | $80.75_{\pm0.64}$ | $57.95_{\pm1.98}$ | $81.32_{\pm2.25}$ | $83.91_{\pm5.14}$ | $72.30_{\pm3.24}$ | $53.43_{\pm0.71}$ | $53.46_{\pm2.41}$ |
| | EdgePreGPPT | $62.42_{\pm11.37}$ | $62.25_{\pm7.23}$ | $63.45_{\pm13.43}$ | $95.30_{\pm4.46}$ | $78.68_{\pm10.06}$ | $51.90_{\pm0.91}$ | $54.22_{\pm1.57}$ |
| | EdgePredGprompt | $77.24_{\pm3.51}$ | $64.14_{\pm1.01}$ | $75.97_{\pm3.33}$ | $91.72_{\pm1.62}$ | $80.10_{\pm10.86}$ | $50.40_{\pm3.16}$ | $75.81_{\pm2.83}$ |
| | GCL | $65.99_{\pm10.83}$ | $65.27_{\pm11.57}$ | $69.76_{\pm4.24}$ | $88.87_{\pm4.34}$ | $76.09_{\pm6.05}$ | $54.21_{\pm2.36}$ | $55.77_{\pm4.78}$ |
| | SimGRACE | $59.53_{\pm3.53}$ | $64.55_{\pm7.97}$ | $47.71_{\pm3.86}$ | $86.55_{\pm2.81}$ | $83.01_{\pm10.84}$ | $52.44_{\pm1.68}$ | $53.56_{\pm7.86}$ |
| Gprompt | DGI | $70.76_{\pm4.67}$ | $80.01_{\pm2.22}$ | $54.50_{\pm2.51}$ | $84.53_{\pm17.37}$ | $65.41_{\pm9.24}$ | $54.02_{\pm2.61}$ | $92.66_{\pm0.73}$ |
| | GraphMAE | $72.59_{\pm7.56}$ | $76.04_{\pm5.23}$ | $82.88_{\pm2.18}$ | $89.40_{\pm19.62}$ | $66.34_{\pm10.38}$ | $56.23_{\pm2.46}$ | $96.45_{\pm0.14}$ |
| | EdgePreGPPT | $85.57_{\pm0.39}$ | $68.97_{\pm4.56}$ | $76.57_{\pm1.08}$ | $89.57_{\pm19.71}$ | $66.48_{\pm10.55}$ | $58.85_{\pm2.98}$ | $98.40_{\pm0.08}$ |
| | EdgePredGprompt | $88.49_{\pm0.88}$ | $84.82_{\pm2.89}$ | $83.58_{\pm1.29}$ | $89.71_{\pm19.78}$ | $66.52_{\pm10.61}$ | $52.51_{\pm2.39}$ | $97.79_{\pm0.15}$ |
| | GCL | $88.70_{\pm2.05}$ | $81.00_{\pm6.61}$ | $74.92_{\pm2.53}$ | $83.64_{\pm16.83}$ | $66.33_{\pm10.37}$ | $63.03_{\pm3.07}$ | $94.54_{\pm0.37}$ |
| | SimGRACE | $78.28_{\pm3.96}$ | $89.06_{\pm0.86}$ | $59.22_{\pm5.76}$ | $86.22_{\pm18.17}$ | $66.68_{\pm10.79}$ | $58.88_{\pm3.13}$ | $95.27_{\pm0.28}$ |
| GPF | DGI | $62.20_{\pm15.24}$ | $62.32_{\pm17.75}$ | $67.14_{\pm11.86}$ | $89.86_{\pm8.25}$ | $78.39_{\pm1.66}$ | $65.97_{\pm7.26}$ | $74.59_{\pm0.97}$ |
| | GraphMAE | $80.34_{\pm2.03}$ | $67.33_{\pm3.16}$ | $86.64_{\pm2.02}$ | $97.72_{\pm3.33}$ | $89.25_{\pm8.10}$ | $73.06_{\pm2.24}$ | $74.39_{\pm0.76}$ |
| | EdgePreGPPT | $49.95_{\pm0.10}$ | $52.76_{\pm6.92}$ | $74.91_{\pm5.59}$ | $99.98_{\pm0.03}$ | $83.65_{\pm7.64}$ | $59.31_{\pm5.01}$ | $90.96_{\pm1.32}$ |
| | EdgePredGprompt | $73.56_{\pm5.48}$ | $76.82_{\pm4.28}$ | $85.50_{\pm1.95}$ | $97.15_{\pm4.98}$ | $86.81_{\pm8.42}$ | $60.43_{\pm6.30}$ | $75.36_{\pm2.61}$ |
| | GCL | $79.04_{\pm5.41}$ | $64.25_{\pm7.12}$ | $70.90_{\pm6.31}$ | $73.84_{\pm18.96}$ | $83.62_{\pm4.75}$ | $57.39_{\pm3.99}$ | $75.19_{\pm0.54}$ |
| | SimGRACE | $60.98_{\pm9.22}$ | $64.93_{\pm3.80}$ | $58.21_{\pm3.99}$ | $90.38_{\pm1.52}$ | $90.43_{\pm8.89}$ | $61.68_{\pm2.02}$ | $60.89_{\pm3.56}$ |
| GPF-plus | DGI | $60.30_{\pm14.39}$ | $54.40_{\pm11.09}$ | $73.47_{\pm9.02}$ | $95.86_{\pm3.95}$ | $89.08_{\pm8.43}$ | $68.06_{\pm2.70}$ | $67.37_{\pm3.03}$ |
| | GraphMAE | $89.99_{\pm1.39}$ | $92.09_{\pm1.11}$ | $85.69_{\pm3.12}$ | $99.93_{\pm0.14}$ | $87.29_{\pm7.92}$ | $73.10_{\pm2.68}$ | $81.99_{\pm0.56}$ |
| | EdgePreGPPT | $65.26_{\pm14.38}$ | $55.10_{\pm2.95}$ | $82.94_{\pm2.55}$ | $99.96_{\pm0.05}$ | $86.89_{\pm8.72}$ | $70.17_{\pm4.50}$ | $93.73_{\pm1.05}$ |
| | EdgePredGprompt | $91.09_{\pm1.03}$ | $82.22_{\pm12.12}$ | $83.84_{\pm3.91}$ | $99.96_{\pm0.08}$ | $83.88_{\pm7.87}$ | $72.25_{\pm1.48}$ | $81.54_{\pm1.05}$ |
| | GCL | $84.11_{\pm7.92}$ | $61.55_{\pm15.26}$ | $81.38_{\pm3.75}$ | $91.95_{\pm9.63}$ | $86.24_{\pm7.72}$ | $69.74_{\pm1.87}$ | $77.42_{\pm2.14}$ |
| | SimGRACE | $66.55_{\pm14.49}$ | $63.11_{\pm14.51}$ | $65.07_{\pm10.97}$ | $88.17_{\pm7.54}$ | $89.58_{\pm9.03}$ | $68.81_{\pm1.60}$ | $69.60_{\pm2.37}$ |

Table 20: Graph classification performance Accuracy (%) (1-shot)

| Training schemes | Methods | IMDB-B | COLLAB | PROTEINS | MUTAG | ENZYMES | COX2 | BZR | DD |
|---|---|---|---|---|---|---|---|---|---|
| self-supervise | GCN | $57.30_{\pm0.98}$ | $47.23_{\pm0.61}$ | $56.36_{\pm7.97}$ | $65.20_{\pm6.70}$ | $20.58_{\pm2.00}$ | $27.08_{\pm11.94}$ | $25.80_{\pm6.53}$ | $55.33_{\pm6.22}$ |
| Pretrain + Fine-tuning | DGI | $57.32_{\pm0.90}$ | $42.22_{\pm0.73}$ | $60.00_{\pm4.48}$ | $64.13_{\pm7.90}$ | $17.83_{\pm1.88}$ | $29.44_{\pm9.68}$ | $26.48_{\pm7.61}$ | $57.15_{\pm4.32}$ |
| | GraphMAE | $57.70_{\pm1.13}$ | $48.10_{\pm0.23}$ | $62.40_{\pm1.94}$ | $65.20_{\pm5.00}$ | $22.21_{\pm2.79}$ | $28.47_{\pm14.72}$ | $25.80_{\pm6.53}$ | $53.59_{\pm6.93}$ |
| | EdgePreGPPT | $57.20_{\pm0.85}$ | $47.14_{\pm0.55}$ | $58.27_{\pm10.66}$ | $64.27_{\pm4.73}$ | $19.79_{\pm2.17}$ | $27.83_{\pm13.44}$ | $34.69_{\pm8.50}$ | $52.82_{\pm9.38}$ |
| | EdgePreGprompt | $57.35_{\pm0.92}$ | $47.20_{\pm0.53}$ | $61.84_{\pm2.59}$ | $62.67_{\pm2.67}$ | $19.75_{\pm2.33}$ | $27.13_{\pm12.05}$ | $29.44_{\pm11.20}$ | $56.16_{\pm5.10}$ |
| | GCL | $57.75_{\pm1.02}$ | $39.62_{\pm0.63}$ | $63.44_{\pm3.64}$ | $65.07_{\pm8.38}$ | $21.21_{\pm0.87}$ | $53.14_{\pm21.32}$ | $29.07_{\pm7.00}$ | $55.50_{\pm5.83}$ |
| | SimGRACE | $57.33_{\pm0.96}$ | $46.89_{\pm0.42}$ | $60.07_{\pm3.21}$ | $65.47_{\pm5.89}$ | $19.71_{\pm1.76}$ | $76.19_{\pm5.41}$ | $28.46_{\pm6.49}$ | $53.23_{\pm9.71}$ |
| GPPTPrompt | DGI | $49.07_{\pm10.36}$ | $39.34_{\pm9.11}$ | $60.81_{\pm1.55}$ | $51.33_{\pm15.87}$ | $20.29_{\pm1.40}$ | $78.23_{\pm1.38}$ | $44.07_{\pm22.42}$ | $53.65_{\pm10.00}$ |
| | GraphMAE | $50.15_{\pm0.75}$ | $29.46_{\pm13.65}$ | $60.72_{\pm1.70}$ | $44.80_{\pm15.52}$ | $20.37_{\pm1.96}$ | $68.63_{\pm20.51}$ | $55.99_{\pm19.28}$ | $57.69_{\pm6.89}$ |
| | EdgePreGPPT | $49.38_{\pm10.29}$ | $36.47_{\pm7.88}$ | $60.92_{\pm2.47}$ | $42.80_{\pm12.98}$ | $20.87_{\pm2.42}$ | $73.99_{\pm9.79}$ | $49.81_{\pm20.17}$ | $53.69_{\pm6.88}$ |
| | EdgePreGprompt | $50.15_{\pm0.75}$ | $40.22_{\pm9.56}$ | $57.03_{\pm4.55}$ | $37.87_{\pm10.43}$ | $22.08_{\pm3.42}$ | $72.28_{\pm13.22}$ | $50.06_{\pm18.97}$ | $55.33_{\pm8.51}$ |
| | GCL | $45.70_{\pm8.20}$ | $47.18_{\pm5.93}$ | $59.24_{\pm1.01}$ | $60.40_{\pm15.43}$ | $21.29_{\pm3.79}$ | $68.36_{\pm21.05}$ | $59.32_{\pm11.22}$ | $56.26_{\pm8.20}$ |
| | SimGRACE | $46.03_{\pm10.29}$ | $41.11_{\pm8.47}$ | $55.42_{\pm8.81}$ | $52.67_{\pm17.12}$ | $20.83_{\pm3.47}$ | $62.31_{\pm19.42}$ | $59.20_{\pm15.06}$ | $55.88_{\pm7.81}$ |
| ALL-in-one | DGI | $60.07_{\pm4.81}$ | $39.56_{\pm5.00}$ | $62.58_{\pm7.07}$ | $75.73_{\pm7.75}$ | $23.96_{\pm1.43}$ | $50.72_{\pm9.93}$ | $79.20_{\pm1.65}$ | $55.97_{\pm6.52}$ |
| | GraphMAE | $52.62_{\pm3.04}$ | $40.82_{\pm14.63}$ | $66.49_{\pm6.26}$ | $72.27_{\pm8.87}$ | $23.21_{\pm1.72}$ | $56.68_{\pm7.38}$ | $75.37_{\pm5.99}$ | $58.77_{\pm1.05}$ |
| | EdgePreGPPT | $59.12_{\pm0.77}$ | $42.74_{\pm4.65}$ | $65.71_{\pm5.49}$ | $79.87_{\pm5.34}$ | $20.92_{\pm2.04}$ | $60.27_{\pm16.97}$ | $64.20_{\pm19.74}$ | $56.24_{\pm2.46}$ |
| | EdgePreGprompt | $53.78_{\pm2.82}$ | $42.87_{\pm6.19}$ | $61.82_{\pm7.53}$ | $75.47_{\pm5.00}$ | $21.88_{\pm0.56}$ | $49.06_{\pm5.53}$ | $66.60_{\pm17.36}$ | $57.60_{\pm4.37}$ |
| | GCL | $58.75_{\pm0.80}$ | $51.66_{\pm0.26}$ | $64.36_{\pm7.30}$ | $70.53_{\pm8.58}$ | $19.46_{\pm2.85}$ | $52.55_{\pm13.51}$ | $50.93_{\pm16.83}$ | $59.72_{\pm1.52}$ |
| | SimGRACE | $58.83_{\pm0.85}$ | $47.60_{\pm0.39}$ | $61.17_{\pm1.73}$ | $65.47_{\pm6.91}$ | $22.50_{\pm1.56}$ | $76.14_{\pm5.51}$ | $69.88_{\pm18.06}$ | $58.26_{\pm1.18}$ |
| Gprompt | DGI | $50.47_{\pm10.10}$ | $47.29_{\pm7.78}$ | $56.61_{\pm7.93}$ | $63.33_{\pm14.36}$ | $20.50_{\pm1.79}$ | $45.52_{\pm16.98}$ | $55.43_{\pm13.69}$ | $56.18_{\pm6.13}$ |
| | GraphMAE | $54.75_{\pm12.43}$ | $36.39_{\pm7.72}$ | $57.66_{\pm12.56}$ | $68.80_{\pm4.76}$ | $19.54_{\pm1.99}$ | $43.91_{\pm6.64}$ | $47.16_{\pm4.72}$ | $55.22_{\pm6.40}$ |
| | EdgePreGPPT | $51.18_{\pm11.11}$ | $46.70_{\pm5.74}$ | $59.17_{\pm11.26}$ | $52.13_{\pm5.80}$ | $19.71_{\pm4.46}$ | $50.08_{\pm8.00}$ | $45.06_{\pm15.93}$ | $51.04_{\pm4.82}$ |
| | EdgePreGprompt | $51.57_{\pm11.87}$ | $40.53_{\pm12.02}$ | $55.55_{\pm8.17}$ | $73.60_{\pm4.76}$ | $19.67_{\pm3.08}$ | $54.64_{\pm9.94}$ | $51.36_{\pm15.55}$ | $57.20_{\pm5.54}$ |
| | GCL | $50.50_{\pm10.42}$ | $45.54_{\pm9.05}$ | $55.51_{\pm10.73}$ | $56.00_{\pm13.79}$ | $19.83_{\pm2.19}$ | $44.40_{\pm5.74}$ | $46.42_{\pm20.67}$ | $52.65_{\pm9.17}$ |
| | SimGRACE | $50.40_{\pm10.54}$ | $48.25_{\pm13.64}$ | $57.53_{\pm11.05}$ | $64.67_{\pm7.92}$ | $22.29_{\pm3.50}$ | $47.02_{\pm5.59}$ | $52.90_{\pm11.76}$ | $57.81_{\pm2.68}$ |
| GPF | DGI | $52.85_{\pm8.91}$ | $42.75_{\pm7.10}$ | $59.17_{\pm3.63}$ | $63.07_{\pm7.22}$ | $22.00_{\pm1.25}$ | $27.94_{\pm13.65}$ | $70.56_{\pm15.46}$ | $59.36_{\pm1.18}$ |
| | GraphMAE | $49.27_{\pm7.77}$ | $37.23_{\pm17.95}$ | $58.65_{\pm8.49}$ | $65.73_{\pm6.91}$ | $20.71_{\pm1.92}$ | $40.43_{\pm10.43}$ | $67.84_{\pm20.28}$ | $55.80_{\pm6.34}$ |
| | EdgePreGPPT | $59.35_{\pm1.02}$ | $37.53_{\pm5.19}$ | $62.54_{\pm2.55}$ | $66.40_{\pm5.93}$ | $22.04_{\pm1.48}$ | $27.40_{\pm12.58}$ | $24.75_{\pm8.01}$ | $43.86_{\pm5.37}$ |
| | EdgePreGprompt | $59.65_{\pm5.06}$ | $41.44_{\pm0.52}$ | $61.82_{\pm2.61}$ | $68.40_{\pm5.09}$ | $22.17_{\pm1.48}$ | $65.79_{\pm17.72}$ | $60.49_{\pm24.20}$ | $55.56_{\pm5.07}$ |
| | GCL | $57.73_{\pm0.79}$ | $47.42_{\pm1.22}$ | $63.91_{\pm3.26}$ | $59.20_{\pm7.01}$ | $21.13_{\pm2.11}$ | $37.05_{\pm9.39}$ | $71.67_{\pm14.71}$ | $58.43_{\pm1.17}$ |
| | SimGRACE | $58.30_{\pm0.77}$ | $41.04_{\pm0.24}$ | $63.35_{\pm3.69}$ | $66.93_{\pm5.05}$ | $21.79_{\pm2.40}$ | $33.99_{\pm11.05}$ | $27.47_{\pm5.92}$ | $58.03_{\pm1.63}$ |
| GPF-plus | DGI | $57.87_{\pm6.19}$ | $43.98_{\pm7.17}$ | $61.26_{\pm3.06}$ | $62.53_{\pm6.86}$ | $18.71_{\pm1.19}$ | $26.70_{\pm11.19}$ | $67.90_{\pm14.60}$ | $48.79_{\pm9.14}$ |
| | GraphMAE | $55.00_{\pm5.81}$ | $40.32_{\pm9.49}$ | $62.49_{\pm2.05}$ | $62.13_{\pm8.71}$ | $22.92_{\pm1.64}$ | $33.78_{\pm11.61}$ | $68.40_{\pm20.02}$ | $56.31_{\pm4.93}$ |
| | EdgePreGPPT | $56.65_{\pm4.08}$ | $41.41_{\pm0.73}$ | $63.06_{\pm2.55}$ | $65.07_{\pm4.41}$ | $21.50_{\pm2.37}$ | $27.83_{\pm11.54}$ | $29.57_{\pm7.69}$ | $57.62_{\pm2.42}$ |
| | EdgePreGprompt | $55.35_{\pm3.91}$ | $40.08_{\pm2.95}$ | $61.33_{\pm2.81}$ | $65.20_{\pm6.04}$ | $19.42_{\pm1.88}$ | $28.85_{\pm15.48}$ | $57.22_{\pm18.17}$ | $55.90_{\pm5.56}$ |
| | GCL | $57.10_{\pm1.34}$ | $46.89_{\pm0.24}$ | $59.75_{\pm7.95}$ | $64.00_{\pm7.89}$ | $18.79_{\pm1.46}$ | $25.90_{\pm9.58}$ | $71.17_{\pm14.92}$ | $57.56_{\pm2.54}$ |
| | SimGRACE | $57.93_{\pm1.62}$ | $47.24_{\pm0.29}$ | $62.92_{\pm2.78}$ | $61.33_{\pm3.84}$ | $20.29_{\pm1.62}$ | $27.08_{\pm7.86}$ | $28.46_{\pm6.49}$ | $57.11_{\pm3.42}$ |

Table 21: Graph classification performance F1-score (1-shot)

| Training schemes | Methods | IMDB-B | COLLAB | PROTEINS | MUTAG | ENZYMES | COX2 | BZR | DD |
|---|---|---|---|---|---|---|---|---|---|
| self-supervise | GCN | $54.62_{\pm1.12}$ | $41.10_{\pm0.39}$ | $46.69_{\pm10.82}$ | $63.47_{\pm6.36}$ | $15.25_{\pm3.96}$ | $22.78_{\pm10.69}$ | $23.71_{\pm8.23}$ | $44.74_{\pm4.23}$ |
| Pretrain + Fine-tuning | DGI | $54.60_{\pm1.00}$ | $38.53_{\pm0.34}$ | $54.82_{\pm3.34}$ | $61.97_{\pm7.76}$ | $10.76_{\pm4.28}$ | $27.09_{\pm11.48}$ | $24.34_{\pm9.21}$ | $46.15_{\pm5.41}$ |
| | GraphMAE | $55.20_{\pm1.24}$ | $41.71_{\pm0.17}$ | $52.05_{\pm7.26}$ | $63.41_{\pm4.44}$ | $19.17_{\pm3.42}$ | $23.63_{\pm12.40}$ | $23.71_{\pm8.23}$ | $46.25_{\pm7.84}$ |
| | EdgePreGPPT | $54.39_{\pm0.95}$ | $41.10_{\pm0.37}$ | $55.82_{\pm10.61}$ | $60.94_{\pm3.46}$ | $12.89_{\pm3.54}$ | $23.08_{\pm11.30}$ | $33.12_{\pm7.45}$ | $36.31_{\pm5.78}$ |
| | EdgePreGprompt | $54.62_{\pm1.07}$ | $41.14_{\pm0.41}$ | $59.73_{\pm1.34}$ | $59.05_{\pm1.33}$ | $13.72_{\pm4.13}$ | $22.91_{\pm10.95}$ | $27.09_{\pm12.51}$ | $45.49_{\pm4.58}$ |
| | GCL | $55.24_{\pm1.07}$ | $36.27_{\pm0.63}$ | $56.25_{\pm7.55}$ | $63.37_{\pm8.64}$ | $16.78_{\pm1.91}$ | $39.11_{\pm4.29}$ | $27.67_{\pm8.70}$ | $48.68_{\pm6.42}$ |
| | SimGRACE | $54.69_{\pm1.09}$ | $40.92_{\pm0.37}$ | $52.67_{\pm7.14}$ | $63.70_{\pm5.32}$ | $14.15_{\pm2.49}$ | $45.06_{\pm1.93}$ | $27.05_{\pm8.20}$ | $37.84_{\pm7.14}$ |
| GPPTPrompt | DGI | $41.17_{\pm12.71}$ | $27.05_{\pm13.23}$ | $46.05_{\pm10.61}$ | $41.76_{\pm15.84}$ | $17.26_{\pm2.39}$ | $44.68_{\pm1.17}$ | $33.93_{\pm14.06}$ | $43.61_{\pm5.55}$ |
| | GraphMAE | $33.40_{\pm0.33}$ | $14.61_{\pm5.30}$ | $46.64_{\pm11.32}$ | $39.44_{\pm18.49}$ | $17.61_{\pm2.22}$ | $40.48_{\pm7.24}$ | $44.38_{\pm9.91}$ | $50.34_{\pm5.80}$ |
| | EdgePreGPPT | $44.16_{\pm6.70}$ | $21.35_{\pm9.96}$ | $47.07_{\pm11.95}$ | $38.03_{\pm16.75}$ | $17.02_{\pm2.90}$ | $43.67_{\pm0.88}$ | $40.73_{\pm9.69}$ | $51.50_{\pm6.54}$ |
| | EdgePreGprompt | $33.40_{\pm0.33}$ | $18.92_{\pm3.19}$ | $43.34_{\pm8.00}$ | $31.42_{\pm13.60}$ | $19.87_{\pm2.99}$ | $43.73_{\pm0.75}$ | $41.78_{\pm10.08}$ | $45.12_{\pm7.86}$ |
| | GCL | $39.08_{\pm10.25}$ | $42.87_{\pm7.70}$ | $41.15_{\pm7.80}$ | $53.15_{\pm16.82}$ | $19.62_{\pm3.92}$ | $40.11_{\pm7.97}$ | $45.58_{\pm3.38}$ | $50.02_{\pm8.57}$ |
| | SimGRACE | $43.18_{\pm8.95}$ | $33.88_{\pm13.05}$ | $40.87_{\pm7.11}$ | $46.54_{\pm18.13}$ | $18.87_{\pm3.37}$ | $41.86_{\pm9.35}$ | $49.40_{\pm8.41}$ | $46.82_{\pm6.89}$ |
| ALL-in-one | DGI | $56.82_{\pm6.07}$ | $35.40_{\pm5.55}$ | $60.66_{\pm6.94}$ | $74.53_{\pm7.04}$ | $14.48_{\pm3.58}$ | $44.46_{\pm4.45}$ | $62.11_{\pm7.06}$ | $48.28_{\pm7.29}$ |
| | GraphMAE | $45.83_{\pm5.38}$ | $18.76_{\pm5.47}$ | $64.27_{\pm4.78}$ | $69.78_{\pm8.83}$ | $19.66_{\pm3.11}$ | $49.43_{\pm3.96}$ | $53.85_{\pm3.84}$ | $56.70_{\pm1.89}$ |
| | EdgePreGPPT | $57.29_{\pm0.74}$ | $37.07_{\pm5.56}$ | $64.68_{\pm5.35}$ | $78.57_{\pm4.92}$ | $12.95_{\pm3.18}$ | $49.62_{\pm10.42}$ | $51.66_{\pm12.74}$ | $55.10_{\pm1.49}$ |
| | EdgePreGprompt | $48.44_{\pm4.51}$ | $34.64_{\pm5.55}$ | $60.04_{\pm8.57}$ | $73.31_{\pm5.46}$ | $12.50_{\pm3.12}$ | $45.57_{\pm5.70}$ | $44.06_{\pm5.14}$ | $48.13_{\pm4.31}$ |
| | GCL | $56.83_{\pm0.76}$ | $47.78_{\pm0.10}$ | $62.99_{\pm7.19}$ | $64.90_{\pm11.24}$ | $12.01_{\pm5.16}$ | $46.65_{\pm6.50}$ | $41.89_{\pm8.44}$ | $43.55_{\pm8.21}$ |
| | SimGRACE | $56.88_{\pm0.80}$ | $41.64_{\pm0.13}$ | $53.18_{\pm7.57}$ | $58.57_{\pm11.26}$ | $12.23_{\pm2.42}$ | $45.03_{\pm1.86}$ | $46.16_{\pm7.12}$ | $39.55_{\pm5.05}$ |
| Gprompt | DGI | $48.68_{\pm9.78}$ | $42.80_{\pm9.19}$ | $55.95_{\pm7.78}$ | $61.15_{\pm13.98}$ | $18.68_{\pm2.94}$ | $38.30_{\pm12.89}$ | $44.61_{\pm5.71}$ | $49.81_{\pm1.61}$ |
| | GraphMAE | $52.10_{\pm13.61}$ | $17.64_{\pm2.56}$ | $55.24_{\pm12.01}$ | $64.58_{\pm3.26}$ | $18.36_{\pm2.20}$ | $37.75_{\pm10.47}$ | $43.38_{\pm3.73}$ | $50.47_{\pm3.41}$ |
| | EdgePreGPPT | $49.33_{\pm10.58}$ | $43.20_{\pm8.14}$ | $58.30_{\pm10.88}$ | $50.70_{\pm6.00}$ | $18.20_{\pm5.07}$ | $44.54_{\pm3.28}$ | $39.06_{\pm9.23}$ | $50.78_{\pm5.00}$ |
| | EdgePreGprompt | $50.43_{\pm11.93}$ | $36.62_{\pm12.55}$ | $54.29_{\pm7.32}$ | $71.38_{\pm3.64}$ | $17.17_{\pm4.25}$ | $46.26_{\pm5.14}$ | $43.73_{\pm9.27}$ | $48.18_{\pm4.55}$ |
| | GCL | $48.91_{\pm10.12}$ | $40.78_{\pm10.09}$ | $53.98_{\pm9.93}$ | $53.39_{\pm14.36}$ | $18.26_{\pm2.77}$ | $42.26_{\pm4.15}$ | $38.58_{\pm11.82}$ | $50.85_{\pm8.14}$ |
| | SimGRACE | $48.78_{\pm10.20}$ | $43.35_{\pm10.75}$ | $55.51_{\pm10.10}$ | $60.58_{\pm6.08}$ | $19.52_{\pm3.36}$ | $44.68_{\pm4.01}$ | $44.81_{\pm6.73}$ | $52.80_{\pm3.60}$ |
| GPF | DGI | $50.50_{\pm7.88}$ | $34.56_{\pm6.02}$ | $49.27_{\pm10.07}$ | $62.02_{\pm6.87}$ | $15.08_{\pm1.44}$ | $23.42_{\pm11.99}$ | $44.77_{\pm3.37}$ | $39.53_{\pm5.01}$ |
| | GraphMAE | $42.98_{\pm8.29}$ | $17.19_{\pm6.84}$ | $52.62_{\pm9.40}$ | $59.14_{\pm10.41}$ | $13.10_{\pm3.28}$ | $37.75_{\pm10.47}$ | $41.84_{\pm7.65}$ | $48.52_{\pm7.11}$ |
| | EdgePreGPPT | $55.67_{\pm0.84}$ | $34.09_{\pm5.32}$ | $57.01_{\pm5.79}$ | $58.18_{\pm3.05}$ | $16.57_{\pm1.64}$ | $22.68_{\pm10.50}$ | $21.77_{\pm9.16}$ | $34.22_{\pm10.09}$ |
| | EdgePreGprompt | $56.22_{\pm6.17}$ | $38.14_{\pm0.44}$ | $56.91_{\pm6.21}$ | $63.90_{\pm4.05}$ | $17.34_{\pm2.45}$ | $43.08_{\pm4.88}$ | $39.86_{\pm11.54}$ | $47.44_{\pm4.83}$ |
| | GCL | $55.23_{\pm0.77}$ | $38.04_{\pm0.46}$ | $56.08_{\pm7.40}$ | $57.99_{\pm6.96}$ | $15.97_{\pm3.75}$ | $35.89_{\pm9.97}$ | $48.83_{\pm5.30}$ | $40.86_{\pm4.89}$ |
| | SimGRACE | $56.19_{\pm0.68}$ | $37.69_{\pm0.21}$ | $55.50_{\pm9.14}$ | $58.38_{\pm2.44}$ | $14.39_{\pm3.45}$ | $31.82_{\pm12.08}$ | $26.02_{\pm7.56}$ | $39.13_{\pm4.20}$ |
| GPF-plus | DGI | $53.13_{\pm10.49}$ | $37.59_{\pm1.42}$ | $54.74_{\pm7.01}$ | $61.19_{\pm6.31}$ | $13.03_{\pm1.21}$ | $22.69_{\pm10.52}$ | $46.57_{\pm4.62}$ | $33.21_{\pm4.87}$ |
| | GraphMAE | $48.23_{\pm10.59}$ | $18.94_{\pm3.14}$ | $52.88_{\pm6.59}$ | $61.01_{\pm8.89}$ | $18.39_{\pm2.76}$ | $30.90_{\pm11.56}$ | $44.87_{\pm9.19}$ | $46.24_{\pm4.86}$ |
| | EdgePreGPPT | $50.88_{\pm7.65}$ | $37.29_{\pm2.27}$ | $57.58_{\pm7.28}$ | $62.03_{\pm2.92}$ | $17.40_{\pm2.32}$ | $24.17_{\pm10.38}$ | $28.06_{\pm9.23}$ | $40.06_{\pm6.06}$ |
| | EdgePreGprompt | $50.07_{\pm6.94}$ | $37.43_{\pm1.84}$ | $54.79_{\pm2.74}$ | $63.20_{\pm5.31}$ | $14.44_{\pm1.44}$ | $24.60_{\pm14.35}$ | $42.77_{\pm1.25}$ | $45.59_{\pm4.64}$ |
| | GCL | $54.24_{\pm1.69}$ | $38.53_{\pm0.20}$ | $57.46_{\pm6.94}$ | $62.31_{\pm7.93}$ | $13.66_{\pm2.70}$ | $22.64_{\pm10.42}$ | $48.71_{\pm5.51}$ | $39.37_{\pm4.68}$ |
| | SimGRACE | $55.55_{\pm2.03}$ | $41.24_{\pm0.31}$ | $54.80_{\pm6.88}$ | $59.38_{\pm2.64}$ | $14.74_{\pm2.94}$ | $24.79_{\pm9.43}$ | $27.05_{\pm8.20}$ | $39.51_{\pm4.96}$ |

Table 22: Graph classification performance AUROC (1-shot)

| Training schemes | Methods | IMDB-B | COLLAB | PROTEINS | MUTAG | ENZYMES | COX2 | BZR | DD |
|---|---|---|---|---|---|---|---|---|---|
| self-supervise | GCN | $67.05_{\pm1.01}$ | $54.23_{\pm0.34}$ | $57.88_{\pm1.72}$ | $71.68_{\pm1.25}$ | $53.49_{\pm1.11}$ | $48.39_{\pm1.89}$ | $51.13_{\pm1.38}$ | $49.60_{\pm2.94}$ |
| Pretrain + Fine-tuning | DGI | $67.06_{\pm1.00}$ | $74.13_{\pm1.27}$ | $56.87_{\pm2.74}$ | $71.45_{\pm1.55}$ | $51.76_{\pm2.18}$ | $52.35_{\pm7.49}$ | $53.79_{\pm1.12}$ | $49.32_{\pm3.05}$ |
| | GraphMAE | $66.91_{\pm1.18}$ | $55.16_{\pm0.20}$ | $59.87_{\pm0.78}$ | $69.81_{\pm1.39}$ | $53.43_{\pm0.98}$ | $49.46_{\pm1.08}$ | $51.01_{\pm0.87}$ | $50.66_{\pm4.90}$ |
| | EdgePreGPPT | $67.50_{\pm1.06}$ | $54.41_{\pm0.31}$ | $58.29_{\pm3.43}$ | $72.42_{\pm1.64}$ | $53.62_{\pm1.61}$ | $46.88_{\pm1.16}$ | $48.35_{\pm3.28}$ | $52.27_{\pm1.89}$ |
| | EdgePreGprompt | $66.92_{\pm0.96}$ | $54.36_{\pm0.30}$ | $58.24_{\pm1.27}$ | $71.58_{\pm0.75}$ | $53.30_{\pm1.75}$ | $49.62_{\pm0.75}$ | $55.48_{\pm2.09}$ | $49.97_{\pm3.16}$ |
| | GCL | $67.11_{\pm1.09}$ | $68.95_{\pm1.48}$ | $59.68_{\pm1.80}$ | $71.07_{\pm1.87}$ | $54.35_{\pm0.92}$ | $51.84_{\pm2.24}$ | $53.34_{\pm2.75}$ | $53.14_{\pm5.70}$ |
| | SimGRACE | $66.95_{\pm0.99}$ | $53.79_{\pm0.47}$ | $59.80_{\pm1.06}$ | $71.49_{\pm1.82}$ | $54.20_{\pm1.36}$ | $49.38_{\pm1.24}$ | $54.01_{\pm2.14}$ | $49.20_{\pm0.84}$ |
| GPPTPrompt | DGI | $48.58_{\pm10.92}$ | $58.79_{\pm4.61}$ | $71.14_{\pm3.06}$ | $51.11_{\pm31.50}$ | $53.09_{\pm1.46}$ | $53.40_{\pm0.84}$ | $48.98_{\pm7.81}$ | $53.82_{\pm11.83}$ |
| | GraphMAE | $50.27_{\pm0.74}$ | $51.39_{\pm4.26}$ | $70.25_{\pm3.45}$ | $36.15_{\pm27.67}$ | $52.89_{\pm1.24}$ | $53.91_{\pm1.86}$ | $54.53_{\pm11.66}$ | $58.36_{\pm8.51}$ |
| | EdgePreGPPT | $48.81_{\pm11.41}$ | $56.39_{\pm0.58}$ | $70.36_{\pm3.79}$ | $33.54_{\pm24.51}$ | $52.98_{\pm1.23}$ | $50.31_{\pm5.34}$ | $51.48_{\pm6.83}$ | $54.80_{\pm8.30}$ |
| | EdgePreGprompt | $50.27_{\pm0.74}$ | $54.48_{\pm1.95}$ | $65.03_{\pm10.25}$ | $24.49_{\pm23.32}$ | $53.83_{\pm2.13}$ | $51.51_{\pm2.95}$ | $53.38_{\pm9.22}$ | $56.26_{\pm10.88}$ |
| | GCL | $45.42_{\pm8.63}$ | $67.97_{\pm6.02}$ | $69.82_{\pm6.05}$ | $65.58_{\pm22.89}$ | $53.36_{\pm2.88}$ | $54.61_{\pm3.27}$ | $51.68_{\pm3.28}$ | $56.31_{\pm9.89}$ |
| | SimGRACE | $45.58_{\pm10.97}$ | $61.78_{\pm7.34}$ | $62.48_{\pm15.87}$ | $48.86_{\pm29.41}$ | $53.23_{\pm2.42}$ | $55.27_{\pm7.83}$ | $58.65_{\pm6.19}$ | $56.15_{\pm9.67}$ |
| ALL-in-one | DGI | $69.12_{\pm1.04}$ | $66.27_{\pm11.54}$ | $75.07_{\pm0.76}$ | $90.86_{\pm1.09}$ | $57.42_{\pm1.15}$ | $50.21_{\pm1.87}$ | $57.93_{\pm12.50}$ | $60.56_{\pm0.91}$ |
| | GraphMAE | $65.65_{\pm1.22}$ | $50.00_{\pm0.00}$ | $73.97_{\pm0.80}$ | $86.98_{\pm0.90}$ | $54.66_{\pm0.90}$ | $57.09_{\pm6.45}$ | $56.21_{\pm11.12}$ | $59.65_{\pm2.74}$ |
| | EdgePreGPPT | $65.44_{\pm0.92}$ | $64.21_{\pm9.84}$ | $75.51_{\pm0.71}$ | $91.34_{\pm0.84}$ | $55.67_{\pm1.65}$ | $61.50_{\pm1.13}$ | $59.90_{\pm1.41}$ | $57.44_{\pm0.59}$ |
| | EdgePreGprompt | $68.48_{\pm1.11}$ | $51.35_{\pm0.30}$ | $73.73_{\pm0.94}$ | $86.05_{\pm1.03}$ | $55.15_{\pm0.59}$ | $57.02_{\pm10.92}$ | $54.72_{\pm1.48}$ | $56.95_{\pm4.42}$ |
| | GCL | $65.20_{\pm1.06}$ | $63.82_{\pm0.26}$ | $77.76_{\pm0.74}$ | $85.11_{\pm0.98}$ | $53.58_{\pm1.38}$ | $66.43_{\pm2.88}$ | $49.56_{\pm2.28}$ | $55.04_{\pm1.51}$ |
| | SimGRACE | $66.33_{\pm0.99}$ | $53.09_{\pm0.21}$ | $57.32_{\pm1.26}$ | $74.91_{\pm1.00}$ | $54.75_{\pm0.64}$ | $45.81_{\pm0.74}$ | $55.96_{\pm0.33}$ | $50.64_{\pm1.32}$ |
| Gprompt | DGI | $53.74_{\pm19.12}$ | $73.04_{\pm8.22}$ | $60.36_{\pm0.55}$ | $69.64_{\pm13.88}$ | $55.68_{\pm2.50}$ | $49.63_{\pm0.56}$ | $49.08_{\pm3.10}$ | $52.16_{\pm4.08}$ |
| | GraphMAE | $51.48_{\pm9.49}$ | $44.86_{\pm0.14}$ | $55.65_{\pm7.44}$ | $70.27_{\pm4.19}$ | $55.28_{\pm3.35}$ | $56.97_{\pm3.76}$ | $51.24_{\pm5.39}$ | $52.47_{\pm6.09}$ |
| | EdgePreGPPT | $55.21_{\pm19.21}$ | $69.85_{\pm10.59}$ | $60.31_{\pm14.12}$ | $59.50_{\pm10.08}$ | $53.75_{\pm2.67}$ | $53.86_{\pm6.62}$ | $49.52_{\pm5.86}$ | $52.73_{\pm6.44}$ |
| | EdgePreGprompt | $53.58_{\pm19.20}$ | $73.09_{\pm5.29}$ | $57.72_{\pm7.86}$ | $79.17_{\pm2.18}$ | $55.53_{\pm3.27}$ | $48.74_{\pm2.54}$ | $50.81_{\pm7.50}$ | $54.96_{\pm5.21}$ |
| | GCL | $53.54_{\pm18.02}$ | $66.59_{\pm11.21}$ | $57.80_{\pm12.67}$ | $71.02_{\pm3.35}$ | $55.06_{\pm3.09}$ | $54.08_{\pm4.37}$ | $47.73_{\pm6.12}$ | $52.92_{\pm6.67}$ |
| | SimGRACE | $53.21_{\pm18.90}$ | $66.59_{\pm8.84}$ | $59.42_{\pm13.20}$ | $66.20_{\pm4.91}$ | $55.23_{\pm3.64}$ | $53.95_{\pm3.70}$ | $49.34_{\pm3.63}$ | $52.15_{\pm6.51}$ |
| GPF | DGI | $55.96_{\pm12.77}$ | $50.91_{\pm1.05}$ | $59.34_{\pm0.55}$ | $71.40_{\pm1.24}$ | $53.13_{\pm1.13}$ | $48.97_{\pm0.73}$ | $47.84_{\pm2.93}$ | $49.47_{\pm0.63}$ |
| | GraphMAE | $61.68_{\pm14.66}$ | $50.00_{\pm0.00}$ | $59.21_{\pm1.29}$ | $73.95_{\pm2.06}$ | $51.73_{\pm1.39}$ | $52.21_{\pm1.49}$ | $46.87_{\pm1.80}$ | $49.36_{\pm4.14}$ |
| | EdgePreGPPT | $69.73_{\pm3.23}$ | $64.68_{\pm11.19}$ | $61.63_{\pm2.41}$ | $68.41_{\pm1.95}$ | $54.48_{\pm1.10}$ | $48.80_{\pm2.39}$ | $51.30_{\pm2.60}$ | $49.08_{\pm3.86}$ |
| | EdgePreGprompt | $69.06_{\pm1.34}$ | $71.37_{\pm1.86}$ | $58.36_{\pm1.20}$ | $74.51_{\pm1.18}$ | $54.01_{\pm0.86}$ | $50.42_{\pm0.62}$ | $58.34_{\pm0.87}$ | $52.14_{\pm2.73}$ |
| | GCL | $67.60_{\pm1.02}$ | $54.30_{\pm0.39}$ | $59.15_{\pm1.73}$ | $68.50_{\pm2.13}$ | $53.76_{\pm1.38}$ | $59.21_{\pm1.87}$ | $56.57_{\pm1.65}$ | $49.30_{\pm1.63}$ |
| | SimGRACE | $66.95_{\pm1.01}$ | $73.55_{\pm0.37}$ | $60.00_{\pm1.43}$ | $68.25_{\pm1.28}$ | $55.25_{\pm1.53}$ | $55.80_{\pm0.69}$ | $52.30_{\pm1.91}$ | $46.17_{\pm0.78}$ |
| GPF-plus | DGI | $68.77_{\pm1.50}$ | $72.06_{\pm4.49}$ | $58.37_{\pm2.32}$ | $70.12_{\pm1.06}$ | $53.83_{\pm0.61}$ | $48.71_{\pm0.50}$ | $51.49_{\pm1.31}$ | $48.76_{\pm1.56}$ |
| | GraphMAE | $68.98_{\pm1.49}$ | $50.00_{\pm0.00}$ | $59.58_{\pm1.13}$ | $72.03_{\pm2.16}$ | $52.68_{\pm1.28}$ | $47.21_{\pm1.67}$ | $53.26_{\pm1.91}$ | $49.42_{\pm2.18}$ |
| | EdgePreGPPT | $69.57_{\pm2.53}$ | $69.92_{\pm1.65}$ | $63.58_{\pm3.31}$ | $72.02_{\pm0.86}$ | $55.66_{\pm1.62}$ | $45.11_{\pm0.89}$ | $55.06_{\pm4.29}$ | $44.29_{\pm3.43}$ |
| | EdgePreGprompt | $68.39_{\pm1.07}$ | $69.85_{\pm1.36}$ | $57.57_{\pm1.65}$ | $73.37_{\pm1.78}$ | $52.19_{\pm1.18}$ | $51.14_{\pm2.27}$ | $56.22_{\pm5.10}$ | $49.74_{\pm1.32}$ |
| | GCL | $67.01_{\pm1.22}$ | $54.03_{\pm0.40}$ | $59.78_{\pm1.47}$ | $71.25_{\pm1.88}$ | $52.59_{\pm2.06}$ | $51.25_{\pm2.24}$ | $56.70_{\pm1.44}$ | $48.94_{\pm2.26}$ |
| | SimGRACE | $67.19_{\pm1.54}$ | $53.04_{\pm0.17}$ | $59.66_{\pm1.32}$ | $68.33_{\pm0.76}$ | $52.47_{\pm1.76}$ | $51.79_{\pm1.73}$ | $53.39_{\pm2.81}$ | $49.75_{\pm1.52}$ |

Table 23: Graph classification performance Accuracy (%) (3-shot)

| Training schemes | Methods | IMDB-B | COLLAB | PROTEINS | MUTAG | ENZYMES | COX2 | BZR | DD |
|---|---|---|---|---|---|---|---|---|---|
| self-supervise | GCN | $53.33_{\pm6.61}$ | $50.77_{\pm2.44}$ | $61.33_{\pm2.89}$ | $59.47_{\pm8.34}$ | $15.96_{\pm1.64}$ | $65.15_{\pm18.61}$ | $52.35_{\pm8.12}$ | $59.77_{\pm1.10}$ |
| Pretrain + Fine-tuning | DGI | $53.33_{\pm6.61}$ | $56.10_{\pm3.46}$ | $61.33_{\pm2.75}$ | $59.87_{\pm8.78}$ | $21.71_{\pm0.81}$ | $51.96_{\pm13.00}$ | $52.22_{\pm10.64}$ | $59.70_{\pm0.98}$ |
| | GraphMAE | $53.33_{\pm6.61}$ | $49.11_{\pm16.81}$ | $61.19_{\pm1.57}$ | $44.00_{\pm13.56}$ | $15.04_{\pm1.86}$ | $60.11_{\pm18.73}$ | $38.52_{\pm17.15}$ | $57.90_{\pm2.69}$ |
| | EdgePreGPPT | $63.43_{\pm2.65}$ | $47.40_{\pm15.87}$ | $62.72_{\pm2.39}$ | $43.47_{\pm13.44}$ | $21.96_{\pm2.45}$ | $49.81_{\pm9.44}$ | $43.70_{\pm17.89}$ | $58.94_{\pm0.66}$ |
| | EdgePreGprompt | $53.33_{\pm6.61}$ | $54.67_{\pm1.34}$ | $60.67_{\pm2.32}$ | $59.20_{\pm7.05}$ | $17.58_{\pm1.45}$ | $35.82_{\pm14.38}$ | $29.07_{\pm9.83}$ | $59.45_{\pm9.10}$ |
| | GCL | $62.22_{\pm1.38}$ | $55.27_{\pm2.61}$ | $62.07_{\pm2.39}$ | $54.80_{\pm5.97}$ | $22.00_{\pm1.71}$ | $31.90_{\pm8.27}$ | $51.23_{\pm11.67}$ | $55.54_{\pm6.26}$ |
| | SimGRACE | $66.10_{\pm0.70}$ | $55.38_{\pm3.58}$ | $60.09_{\pm0.63}$ | $54.00_{\pm7.03}$ | $22.71_{\pm0.86}$ | $69.97_{\pm13.89}$ | $36.67_{\pm1.80}$ | $58.17_{\pm2.79}$ |
| GPPTPrompt | DGI | $50.33_{\pm0.92}$ | $38.40_{\pm8.13}$ | $60.36_{\pm8.99}$ | $64.13_{\pm18.31}$ | $17.67_{\pm2.05}$ | $56.84_{\pm28.02}$ | $67.53_{\pm23.14}$ | $56.11_{\pm7.51}$ |
| | GraphMAE | $50.23_{\pm0.95}$ | $36.37_{\pm7.89}$ | $56.94_{\pm6.67}$ | $47.87_{\pm17.55}$ | $17.63_{\pm1.97}$ | $69.38_{\pm19.31}$ | $70.49_{\pm17.22}$ | $56.50_{\pm8.48}$ |
| | EdgePreGPPT | $59.48_{\pm5.42}$ | $38.45_{\pm9.13}$ | $64.74_{\pm1.99}$ | $52.00_{\pm16.93}$ | $19.12_{\pm2.43}$ | $69.87_{\pm18.34}$ | $58.95_{\pm12.79}$ | $52.14_{\pm6.23}$ |
| | EdgePreGprompt | $51.85_{\pm2.10}$ | $36.11_{\pm7.73}$ | $60.76_{\pm1.52}$ | $62.00_{\pm18.69}$ | $18.71_{\pm5.10}$ | $52.17_{\pm15.58}$ | $77.96_{\pm2.43}$ | $57.94_{\pm7.47}$ |
| | GCL | $50.43_{\pm11.80}$ | $50.88_{\pm6.31}$ | $60.31_{\pm0.57}$ | $48.93_{\pm18.86}$ | $17.25_{\pm1.19}$ | $71.90_{\pm14.28}$ | $78.46_{\pm1.57}$ | $56.28_{\pm7.21}$ |
| | SimGRACE | $50.12_{\pm12.86}$ | $41.87_{\pm8.73}$ | $55.93_{\pm6.16}$ | $57.87_{\pm20.52}$ | $15.62_{\pm2.32}$ | $58.28_{\pm20.18}$ | $67.47_{\pm19.05}$ | $59.00_{\pm6.34}$ |
| ALL-in-one | DGI | $64.28_{\pm0.75}$ | $52.63_{\pm8.14}$ | $69.84_{\pm6.02}$ | $82.40_{\pm3.57}$ | $22.87_{\pm0.93}$ | $52.17_{\pm12.81}$ | $59.81_{\pm15.62}$ | $54.95_{\pm6.52}$ |
| | GraphMAE | $63.88_{\pm0.73}$ | $52.09_{\pm0.33}$ | $65.69_{\pm3.31}$ | $82.00_{\pm4.02}$ | $21.04_{\pm2.51}$ | $53.83_{\pm7.02}$ | $61.98_{\pm11.32}$ | $56.56_{\pm4.54}$ |
| | EdgePreGPPT | $63.80_{\pm1.07}$ | $55.73_{\pm3.59}$ | $65.62_{\pm7.36}$ | $85.60_{\pm2.22}$ | $22.17_{\pm2.17}$ | $50.19_{\pm10.89}$ | $54.26_{\pm13.30}$ | $52.19_{\pm6.17}$ |
| | EdgePreGprompt | $63.90_{\pm1.57}$ | $51.69_{\pm8.39}$ | $61.30_{\pm2.47}$ | $73.60_{\pm4.23}$ | $23.25_{\pm1.11}$ | $60.21_{\pm8.86}$ | $54.44_{\pm17.27}$ | $58.96_{\pm5.93}$ |
| | GCL | $65.67_{\pm0.58}$ | $57.12_{\pm1.99}$ | $65.57_{\pm2.24}$ | $58.93_{\pm10.23}$ | $23.96_{\pm0.62}$ | $52.17_{\pm14.65}$ | $58.64_{\pm4.86}$ | $52.65_{\pm5.87}$ |
| | SimGRACE | $64.20_{\pm1.29}$ | $55.48_{\pm3.48}$ | $62.36_{\pm1.86}$ | $72.40_{\pm4.01}$ | $22.58_{\pm1.18}$ | $66.06_{\pm18.23}$ | $61.30_{\pm16.21}$ | $53.23_{\pm6.95}$ |
| Gprompt | DGI | $58.95_{\pm9.88}$ | $55.27_{\pm8.86}$ | $62.43_{\pm4.09}$ | $54.93_{\pm17.15}$ | $20.50_{\pm2.36}$ | $50.29_{\pm7.71}$ | $49.69_{\pm6.79}$ | $53.84_{\pm5.72}$ |
| | GraphMAE | $59.17_{\pm10.00}$ | $36.11_{\pm7.73}$ | $61.98_{\pm4.45}$ | $64.40_{\pm16.46}$ | $21.42_{\pm2.71}$ | $44.83_{\pm9.16}$ | $48.15_{\pm8.09}$ | $52.89_{\pm5.27}$ |
| | EdgePreGPPT | $64.35_{\pm1.21}$ | $53.20_{\pm7.90}$ | $64.94_{\pm2.92}$ | $53.60_{\pm14.41}$ | $18.50_{\pm3.69}$ | $45.47_{\pm7.03}$ | $54.63_{\pm2.95}$ | $53.61_{\pm2.31}$ |
| | EdgePreGprompt | $59.30_{\pm10.17}$ | $54.95_{\pm9.47}$ | $62.02_{\pm3.15}$ | $66.53_{\pm14.84}$ | $21.42_{\pm3.01}$ | $50.56_{\pm9.27}$ | $49.88_{\pm12.32}$ | $52.55_{\pm6.16}$ |
| | GCL | $59.85_{\pm9.50}$ | $52.52_{\pm9.71}$ | $58.49_{\pm9.20}$ | $52.40_{\pm20.58}$ | $21.42_{\pm0.77}$ | $48.15_{\pm9.42}$ | $54.26_{\pm9.10}$ | $55.61_{\pm3.21}$ |
| | SimGRACE | $60.00_{\pm9.95}$ | $53.45_{\pm7.18}$ | $60.27_{\pm4.44}$ | $56.40_{\pm13.37}$ | $22.08_{\pm3.57}$ | $51.53_{\pm13.08}$ | $43.21_{\pm8.84}$ | $55.99_{\pm7.53}$ |
| GPF | DGI | $63.53_{\pm2.47}$ | $49.84_{\pm7.48}$ | $61.39_{\pm2.63}$ | $48.67_{\pm15.53}$ | $16.63_{\pm3.49}$ | $65.31_{\pm19.45}$ | $61.79_{\pm21.19}$ | $59.07_{\pm0.65}$ |
| | GraphMAE | $62.80_{\pm2.90}$ | $37.01_{\pm13.81}$ | $62.72_{\pm3.07}$ | $55.87_{\pm12.48}$ | $18.29_{\pm2.39}$ | $53.51_{\pm13.09}$ | $51.91_{\pm8.73}$ | $57.15_{\pm5.68}$ |
| | EdgePreGPPT | $65.25_{\pm2.65}$ | $51.91_{\pm8.13}$ | $63.35_{\pm2.45}$ | $74.27_{\pm1.55}$ | $19.92_{\pm2.19}$ | $44.50_{\pm4.46}$ | $54.63_{\pm10.59}$ | $51.59_{\pm5.62}$ |
| | EdgePreGprompt | $64.05_{\pm1.03}$ | $50.37_{\pm7.25}$ | $62.49_{\pm2.18}$ | $55.60_{\pm13.42}$ | $23.08_{\pm3.11}$ | $61.72_{\pm11.67}$ | $74.38_{\pm11.62}$ | $56.37_{\pm6.77}$ |
| | GCL | $63.25_{\pm2.36}$ | $53.87_{\pm3.44}$ | $62.90_{\pm2.52}$ | $54.00_{\pm12.02}$ | $22.38_{\pm1.93}$ | $49.33_{\pm11.40}$ | $50.19_{\pm4.33}$ | $52.34_{\pm6.89}$ |
| | SimGRACE | $65.97_{\pm0.69}$ | $53.23_{\pm4.59}$ | $60.92_{\pm1.65}$ | $50.13_{\pm13.88}$ | $23.87_{\pm3.45}$ | $62.31_{\pm8.87}$ | $25.62_{\pm8.25}$ | $57.54_{\pm4.65}$ |
| GPF-plus | DGI | $62.45_{\pm2.52}$ | $52.14_{\pm7.67}$ | $62.16_{\pm2.14}$ | $75.20_{\pm3.64}$ | $21.92_{\pm0.74}$ | $65.25_{\pm18.07}$ | $60.86_{\pm16.47}$ | $59.43_{\pm0.52}$ |
| | GraphMAE | $61.97_{\pm2.88}$ | $36.87_{\pm13.90}$ | $61.85_{\pm1.85}$ | $59.33_{\pm7.66}$ | $17.08_{\pm1.68}$ | $48.20_{\pm19.15}$ | $40.99_{\pm10.64}$ | $57.18_{\pm5.31}$ |
| | EdgePreGPPT | $64.38_{\pm2.30}$ | $54.63_{\pm7.14}$ | $62.99_{\pm1.94}$ | $72.40_{\pm2.00}$ | $23.62_{\pm2.56}$ | $49.44_{\pm11.92}$ | $36.85_{\pm19.33}$ | $58.85_{\pm1.33}$ |
| | EdgePreGprompt | $64.00_{\pm3.54}$ | $50.77_{\pm9.01}$ | $60.38_{\pm2.47}$ | $50.27_{\pm18.19}$ | $24.46_{\pm2.27}$ | $52.87_{\pm12.00}$ | $50.06_{\pm16.36}$ | $58.54_{\pm2.38}$ |
| | GCL | $63.25_{\pm2.63}$ | $56.50_{\pm3.71}$ | $60.56_{\pm1.94}$ | $74.27_{\pm4.59}$ | $19.00_{\pm2.42}$ | $51.58_{\pm11.78}$ | $71.67_{\pm14.87}$ | $53.99_{\pm6.38}$ |
| | SimGRACE | $63.55_{\pm2.25}$ | $52.72_{\pm6.39}$ | $60.07_{\pm0.97}$ | $50.67_{\pm17.37}$ | $22.17_{\pm2.30}$ | $63.86_{\pm10.00}$ | $25.62_{\pm8.25}$ | $59.51_{\pm0.62}$ |

Table 24: Graph classification performance F1-score (3-shot)

| Training schemes | Methods | IMDB-B | COLLAB | PROTEINS | MUTAG | ENZYMES | COX2 | BZR | DD |
|---|---|---|---|---|---|---|---|---|---|
| self-supervise | GCN | $39.88_{\pm13.07}$ | $40.73_{\pm1.00}$ | $51.01_{\pm8.18}$ | $57.79_{\pm8.44}$ | $11.97_{\pm0.92}$ | $42.57_{\pm5.89}$ | $47.71_{\pm5.27}$ | $41.05_{\pm7.70}$ |
| Pretrain + Fine-tuning | DGI | $39.88_{\pm13.07}$ | $56.10_{\pm3.90}$ | $50.89_{\pm8.01}$ | $58.19_{\pm8.98}$ | $14.68_{\pm2.46}$ | $47.30_{\pm8.88}$ | $46.39_{\pm5.60}$ | $41.31_{\pm8.21}$ |
| | GraphMAE | $39.88_{\pm13.08}$ | $47.71_{\pm19.43}$ | $51.90_{\pm7.98}$ | $37.98_{\pm16.20}$ | $10.19_{\pm2.54}$ | $43.74_{\pm6.72}$ | $33.85_{\pm14.41}$ | $40.24_{\pm6.08}$ |
| | EdgePreGPPT | $60.66_{\pm4.52}$ | $35.28_{\pm13.12}$ | $54.53_{\pm4.29}$ | $37.20_{\pm15.67}$ | $15.47_{\pm3.45}$ | $45.43_{\pm4.85}$ | $39.01_{\pm11.37}$ | $40.78_{\pm7.17}$ |
| | EdgePreGprompt | $39.88_{\pm13.07}$ | $35.69_{\pm7.57}$ | $49.31_{\pm7.52}$ | $57.28_{\pm6.35}$ | $12.99_{\pm2.98}$ | $33.02_{\pm14.02}$ | $26.66_{\pm11.36}$ | $48.50_{\pm10.32}$ |
| | GCL | $60.72_{\pm1.89}$ | $54.45_{\pm4.00}$ | $52.75_{\pm6.77}$ | $53.13_{\pm5.12}$ | $17.00_{\pm3.88}$ | $30.17_{\pm7.80}$ | $45.33_{\pm7.28}$ | $51.94_{\pm4.43}$ |
| | SimGRACE | $65.91_{\pm0.61}$ | $54.77_{\pm3.76}$ | $48.85_{\pm6.10}$ | $52.10_{\pm6.02}$ | $16.13_{\pm2.28}$ | $45.75_{\pm2.10}$ | $36.07_{\pm1.29}$ | $50.54_{\pm6.18}$ |
| GPPTPrompt | DGI | $33.47_{\pm0.41}$ | $25.72_{\pm11.59}$ | $56.81_{\pm8.37}$ | $56.46_{\pm18.74}$ | $7.26_{\pm4.98}$ | $33.97_{\pm12.73}$ | $38.84_{\pm10.65}$ | $52.76_{\pm6.80}$ |
| | GraphMAE | $33.43_{\pm0.42}$ | $17.63_{\pm2.61}$ | $44.10_{\pm8.25}$ | $41.20_{\pm19.70}$ | $7.70_{\pm5.86}$ | $41.47_{\pm5.35}$ | $42.54_{\pm3.25}$ | $53.64_{\pm7.98}$ |
| | EdgePreGPPT | $54.15_{\pm12.01}$ | $28.60_{\pm16.18}$ | $61.46_{\pm1.10}$ | $49.73_{\pm17.65}$ | $12.26_{\pm6.37}$ | $41.87_{\pm4.54}$ | $49.38_{\pm6.09}$ | $50.03_{\pm5.25}$ |
| | EdgePreGprompt | $36.01_{\pm4.61}$ | $17.54_{\pm2.57}$ | $51.33_{\pm7.17}$ | $61.15_{\pm18.28}$ | $11.46_{\pm8.27}$ | $41.89_{\pm5.55}$ | $46.42_{\pm4.52}$ | $53.54_{\pm7.77}$ |
| | GCL | $46.73_{\pm14.15}$ | $49.62_{\pm6.77}$ | $41.87_{\pm8.55}$ | $41.46_{\pm20.04}$ | $9.53_{\pm5.93}$ | $43.94_{\pm0.52}$ | $44.66_{\pm1.04}$ | $53.74_{\pm6.42}$ |
| | SimGRACE | $47.67_{\pm15.00}$ | $38.51_{\pm10.60}$ | $54.02_{\pm4.98}$ | $55.79_{\pm19.08}$ | $6.44_{\pm2.25}$ | $42.49_{\pm8.52}$ | $42.36_{\pm6.71}$ | $56.28_{\pm5.60}$ |
| ALL-in-one | DGI | $63.99_{\pm0.86}$ | $50.43_{\pm9.08}$ | $68.85_{\pm5.72}$ | $81.20_{\pm3.18}$ | $14.63_{\pm1.31}$ | $43.21_{\pm7.33}$ | $53.56_{\pm10.74}$ | $43.35_{\pm5.95}$ |
| | GraphMAE | $62.94_{\pm1.54}$ | $22.83_{\pm0.10}$ | $62.48_{\pm1.62}$ | $79.84_{\pm3.94}$ | $15.61_{\pm1.28}$ | $46.34_{\pm2.82}$ | $55.20_{\pm8.12}$ | $54.23_{\pm4.13}$ |
| | EdgePreGPPT | $62.99_{\pm2.30}$ | $55.56_{\pm4.18}$ | $64.50_{\pm6.98}$ | $83.88_{\pm2.07}$ | $17.11_{\pm3.24}$ | $42.10_{\pm6.16}$ | $44.01_{\pm4.11}$ | $49.69_{\pm6.19}$ |
| | EdgePreGprompt | $62.88_{\pm3.31}$ | $49.83_{\pm10.02}$ | $52.05_{\pm8.59}$ | $71.06_{\pm3.19}$ | $17.70_{\pm1.81}$ | $50.66_{\pm4.25}$ | $47.08_{\pm8.47}$ | $55.21_{\pm4.94}$ |
| | GCL | $65.60_{\pm0.57}$ | $56.55_{\pm1.88}$ | $62.62_{\pm3.77}$ | $57.24_{\pm11.36}$ | $22.37_{\pm1.31}$ | $45.27_{\pm9.63}$ | $49.39_{\pm2.68}$ | $50.28_{\pm4.51}$ |
| | SimGRACE | $63.50_{\pm2.41}$ | $55.34_{\pm3.89}$ | $60.36_{\pm1.11}$ | $69.33_{\pm2.34}$ | $18.20_{\pm1.04}$ | $41.93_{\pm5.18}$ | $49.00_{\pm9.10}$ | $49.06_{\pm6.95}$ |
| Gprompt | DGI | $58.08_{\pm10.23}$ | $55.32_{\pm8.74}$ | $58.05_{\pm3.97}$ | $52.54_{\pm17.09}$ | $18.81_{\pm1.44}$ | $44.13_{\pm4.49}$ | $45.80_{\pm7.78}$ | $50.97_{\pm5.00}$ |
| | GraphMAE | $58.01_{\pm10.19}$ | $17.54_{\pm2.57}$ | $56.27_{\pm5.10}$ | $60.26_{\pm15.18}$ | $19.39_{\pm1.66}$ | $42.42_{\pm7.00}$ | $43.14_{\pm4.47}$ | $49.67_{\pm2.97}$ |
| | EdgePreGPPT | $63.89_{\pm1.11}$ | $53.51_{\pm7.87}$ | $61.58_{\pm3.09}$ | $49.72_{\pm12.38}$ | $16.84_{\pm3.73}$ | $42.05_{\pm5.59}$ | $47.32_{\pm2.37}$ | $52.83_{\pm1.88}$ |
| | EdgePreGprompt | $58.51_{\pm10.62}$ | $55.39_{\pm8.86}$ | $56.27_{\pm2.08}$ | $64.64_{\pm13.86}$ | $18.56_{\pm3.11}$ | $43.87_{\pm4.26}$ | $44.57_{\pm9.37}$ | $49.71_{\pm7.25}$ |
| | GCL | $58.83_{\pm10.22}$ | $52.04_{\pm10.22}$ | $57.08_{\pm8.81}$ | $51.26_{\pm19.96}$ | $19.60_{\pm1.34}$ | $44.29_{\pm5.52}$ | $47.69_{\pm4.92}$ | $54.82_{\pm2.66}$ |
| | SimGRACE | $59.21_{\pm10.45}$ | $52.82_{\pm8.34}$ | $55.53_{\pm7.37}$ | $52.05_{\pm11.07}$ | $21.63_{\pm3.19}$ | $43.90_{\pm7.83}$ | $40.90_{\pm6.25}$ | $53.39_{\pm6.72}$ |
| GPF | DGI | $62.80_{\pm2.95}$ | $47.97_{\pm9.36}$ | $51.68_{\pm7.67}$ | $44.82_{\pm12.44}$ | $11.70_{\pm4.44}$ | $44.13_{\pm3.16}$ | $40.70_{\pm4.95}$ | $40.29_{\pm5.61}$ |
| | GraphMAE | $61.83_{\pm3.65}$ | $17.49_{\pm5.13}$ | $55.21_{\pm7.59}$ | $52.99_{\pm14.80}$ | $15.77_{\pm1.71}$ | $45.77_{\pm7.77}$ | $47.24_{\pm5.79}$ | $54.79_{\pm4.60}$ |
| | EdgePreGPPT | $63.82_{\pm3.68}$ | $50.28_{\pm10.09}$ | $56.68_{\pm3.32}$ | $68.47_{\pm1.51}$ | $18.77_{\pm2.43}$ | $42.92_{\pm3.39}$ | $48.61_{\pm4.66}$ | $49.22_{\pm6.62}$ |
| | EdgePreGprompt | $63.59_{\pm0.99}$ | $48.60_{\pm9.17}$ | $53.75_{\pm5.47}$ | $51.63_{\pm14.57}$ | $16.62_{\pm1.75}$ | $43.32_{\pm4.50}$ | $48.42_{\pm3.50}$ | $53.86_{\pm7.48}$ |
| | GCL | $62.47_{\pm2.84}$ | $53.42_{\pm3.86}$ | $55.24_{\pm8.26}$ | $50.93_{\pm14.25}$ | $15.40_{\pm2.55}$ | $45.37_{\pm9.89}$ | $47.00_{\pm2.24}$ | $48.53_{\pm5.59}$ |
| | SimGRACE | $65.85_{\pm0.61}$ | $52.54_{\pm5.59}$ | $49.67_{\pm6.21}$ | $47.58_{\pm13.28}$ | $18.36_{\pm4.95}$ | $51.59_{\pm4.59}$ | $22.84_{\pm9.35}$ | $55.12_{\pm4.27}$ |
| GPF-plus | DGI | $60.90_{\pm3.22}$ | $50.49_{\pm9.05}$ | $53.28_{\pm5.40}$ | $73.17_{\pm3.16}$ | $17.51_{\pm1.75}$ | $43.17_{\pm5.33}$ | $46.21_{\pm5.60}$ | $40.98_{\pm7.58}$ |
| | GraphMAE | $60.30_{\pm3.23}$ | $17.43_{\pm5.19}$ | $59.82_{\pm4.60}$ | $57.10_{\pm7.36}$ | $12.55_{\pm0.72}$ | $37.20_{\pm9.82}$ | $38.17_{\pm8.53}$ | $52.72_{\pm3.75}$ |
| | EdgePreGPPT | $63.35_{\pm3.33}$ | $53.82_{\pm8.44}$ | $59.39_{\pm1.65}$ | $65.96_{\pm1.23}$ | $18.73_{\pm2.85}$ | $46.05_{\pm10.35}$ | $31.44_{\pm17.20}$ | $41.26_{\pm6.70}$ |
| | EdgePreGprompt | $63.65_{\pm3.51}$ | $48.46_{\pm10.27}$ | $49.63_{\pm6.76}$ | $48.24_{\pm17.45}$ | $19.80_{\pm4.12}$ | $47.82_{\pm9.03}$ | $43.31_{\pm12.09}$ | $44.08_{\pm7.41}$ |
| | GCL | $62.49_{\pm3.11}$ | $56.38_{\pm3.66}$ | $49.25_{\pm7.67}$ | $72.21_{\pm3.78}$ | $15.66_{\pm2.56}$ | $47.78_{\pm9.88}$ | $43.52_{\pm1.30}$ | $43.72_{\pm6.72}$ |
| | SimGRACE | $62.91_{\pm2.78}$ | $51.40_{\pm8.33}$ | $48.38_{\pm6.37}$ | $48.42_{\pm16.33}$ | $16.25_{\pm3.30}$ | $50.53_{\pm3.93}$ | $22.82_{\pm9.32}$ | $40.63_{\pm6.86}$ |

Table 25: Graph classification performance AUROC (3-shot)

| Training schemes | Methods | IMDB-B | COLLAB | PROTEINS | MUTAG | ENZYMES | COX2 | BZR | DD |
|---|---|---|---|---|---|---|---|---|---|
| self-supervise | GCN | $53.85_{\pm7.70}$ | $54.29_{\pm0.68}$ | $58.83_{\pm2.09}$ | $73.36_{\pm0.52}$ | $51.34_{\pm1.42}$ | $53.03_{\pm0.66}$ | $55.36_{\pm2.82}$ | $50.97_{\pm1.94}$ |
| Pretrain + Fine-tuning | DGI | $53.85_{\pm7.71}$ | $79.16_{\pm1.29}$ | $59.21_{\pm1.87}$ | $72.49_{\pm0.69}$ | $55.06_{\pm0.72}$ | $59.34_{\pm3.68}$ | $55.52_{\pm3.89}$ | $50.91_{\pm1.85}$ |
| | GraphMAE | $53.86_{\pm7.71}$ | $72.58_{\pm11.44}$ | $59.12_{\pm1.77}$ | $73.70_{\pm2.39}$ | $51.64_{\pm1.15}$ | $56.52_{\pm2.87}$ | $57.30_{\pm2.71}$ | $50.87_{\pm1.75}$ |
| | EdgePreGPPT | $68.90_{\pm1.00}$ | $53.70_{\pm2.76}$ | $59.61_{\pm1.05}$ | $63.78_{\pm11.40}$ | $56.63_{\pm1.88}$ | $58.62_{\pm4.95}$ | $53.24_{\pm1.06}$ | $50.87_{\pm1.74}$ |
| | EdgePreGprompt | $53.88_{\pm7.76}$ | $54.66_{\pm1.29}$ | $57.43_{\pm1.26}$ | $72.67_{\pm0.71}$ | $52.81_{\pm1.23}$ | $56.43_{\pm2.54}$ | $52.85_{\pm3.05}$ | $55.71_{\pm3.36}$ |
| | GCL | $66.91_{\pm0.96}$ | $77.24_{\pm5.27}$ | $59.63_{\pm1.25}$ | $71.94_{\pm1.69}$ | $55.22_{\pm0.62}$ | $55.65_{\pm1.39}$ | $52.36_{\pm4.70}$ | $56.27_{\pm3.63}$ |
| | SimGRACE | $69.07_{\pm0.74}$ | $78.72_{\pm0.81}$ | $54.56_{\pm4.50}$ | $71.53_{\pm1.53}$ | $57.45_{\pm1.25}$ | $54.90_{\pm1.47}$ | $53.79_{\pm5.86}$ | $58.14_{\pm3.45}$ |
| GPPTPrompt | DGI | $50.70_{\pm0.98}$ | $59.06_{\pm5.42}$ | $63.21_{\pm13.02}$ | $69.64_{\pm24.34}$ | $51.97_{\pm1.01}$ | $51.15_{\pm3.38}$ | $53.46_{\pm4.77}$ | $58.16_{\pm7.86}$ |
| | GraphMAE | $50.02_{\pm1.20}$ | $55.37_{\pm1.65}$ | $64.79_{\pm12.01}$ | $36.64_{\pm30.88}$ | $51.88_{\pm0.85}$ | $51.90_{\pm3.51}$ | $57.33_{\pm2.22}$ | $57.48_{\pm8.59}$ |
| | EdgePreGPPT | $60.13_{\pm5.46}$ | $59.32_{\pm8.93}$ | $68.26_{\pm2.20}$ | $53.05_{\pm23.65}$ | $52.97_{\pm1.80}$ | $54.30_{\pm1.55}$ | $53.75_{\pm1.34}$ | $54.71_{\pm6.13}$ |
| | EdgePreGprompt | $51.09_{\pm3.03}$ | $55.18_{\pm1.91}$ | $68.92_{\pm2.46}$ | $66.15_{\pm23.60}$ | $51.81_{\pm3.18}$ | $50.12_{\pm2.64}$ | $56.67_{\pm1.07}$ | $59.80_{\pm7.91}$ |
| | GCL | $50.36_{\pm13.14}$ | $70.37_{\pm8.98}$ | $71.44_{\pm3.56}$ | $41.39_{\pm32.92}$ | $51.66_{\pm0.70}$ | $54.79_{\pm2.46}$ | $55.63_{\pm1.51}$ | $58.02_{\pm8.00}$ |
| | SimGRACE | $50.30_{\pm13.72}$ | $63.01_{\pm7.00}$ | $57.69_{\pm9.30}$ | $59.52_{\pm26.16}$ | $49.66_{\pm2.26}$ | $53.90_{\pm4.39}$ | $53.94_{\pm3.10}$ | $60.31_{\pm6.86}$ |
| ALL-in-one | DGI | $68.29_{\pm0.65}$ | $81.02_{\pm0.32}$ | $78.78_{\pm0.89}$ | $91.14_{\pm1.01}$ | $55.50_{\pm0.41}$ | $50.11_{\pm4.43}$ | $64.10_{\pm1.82}$ | $47.72_{\pm1.00}$ |
| | GraphMAE | $67.69_{\pm0.61}$ | $50.00_{\pm0.00}$ | $71.38_{\pm4.27}$ | $90.04_{\pm1.80}$ | $54.06_{\pm0.38}$ | $51.31_{\pm6.59}$ | $62.34_{\pm1.48}$ | $59.05_{\pm3.75}$ |
| | EdgePreGPPT | $67.39_{\pm0.61}$ | $80.54_{\pm1.13}$ | $71.08_{\pm8.07}$ | $92.40_{\pm0.68}$ | $55.76_{\pm1.18}$ | $47.76_{\pm1.24}$ | $48.35_{\pm3.93}$ | $54.96_{\pm0.53}$ |
| | EdgePreGprompt | $68.01_{\pm0.66}$ | $77.37_{\pm3.38}$ | $59.05_{\pm4.06}$ | $85.58_{\pm1.01}$ | $55.64_{\pm1.24}$ | | $64.23_{\pm1.73}$ | $60.92_{\pm1.89}$ |
| | GCL | $68.55_{\pm0.68}$ | $81.56_{\pm1.31}$ | $68.66_{\pm3.13}$ | $75.79_{\pm1.97}$ | $54.68_{\pm0.82}$ | $58.33_{\pm4.46}$ | $54.20_{\pm4.83}$ | $55.30_{\pm3.48}$ |
| | SimGRACE | $67.71_{\pm0.63}$ | $79.47_{\pm0.36}$ | $62.54_{\pm0.80}$ | $85.93_{\pm1.67}$ | $54.94_{\pm0.85}$ | $47.16_{\pm4.60}$ | $54.11_{\pm1.26}$ | $58.19_{\pm0.80}$ |
| Gprompt | DGI | $61.28_{\pm6.92}$ | $72.35_{\pm6.43}$ | $60.00_{\pm4.15}$ | $59.82_{\pm8.05}$ | $53.15_{\pm3.14}$ | $49.01_{\pm1.79}$ | $50.69_{\pm8.24}$ | $52.10_{\pm4.06}$ |
| | GraphMAE | $56.22_{\pm7.41}$ | $44.90_{\pm9.12}$ | $53.63_{\pm3.55}$ | $64.43_{\pm15.28}$ | $52.29_{\pm3.02}$ | $55.66_{\pm5.14}$ | $50.42_{\pm1.52}$ | $55.22_{\pm1.63}$ |
| | EdgePreGPPT | $63.01_{\pm2.64}$ | $74.04_{\pm6.87}$ | $65.82_{\pm4.00}$ | $56.24_{\pm15.55}$ | $51.97_{\pm3.30}$ | $49.66_{\pm7.34}$ | $50.37_{\pm2.68}$ | $57.12_{\pm1.01}$ |
| | EdgePreGprompt | $61.76_{\pm6.92}$ | $74.20_{\pm7.18}$ | $63.20_{\pm4.24}$ | $68.36_{\pm15.60}$ | $54.30_{\pm3.58}$ | $51.46_{\pm3.88}$ | $52.11_{\pm4.43}$ | $55.65_{\pm4.85}$ |
| | GCL | $62.10_{\pm7.10}$ | $72.58_{\pm8.88}$ | $55.52_{\pm9.16}$ | $54.21_{\pm19.23}$ | $52.15_{\pm2.03}$ | $57.07_{\pm3.48}$ | $52.27_{\pm2.39}$ | $57.24_{\pm4.44}$ |
| | SimGRACE | $63.49_{\pm7.53}$ | $73.85_{\pm5.74}$ | $58.07_{\pm7.27}$ | $56.06_{\pm18.58}$ | $53.72_{\pm4.10}$ | $53.97_{\pm4.68}$ | $49.80_{\pm7.28}$ | $56.78_{\pm6.19}$ |
| GPF | DGI | $68.17_{\pm1.10}$ | $76.98_{\pm3.82}$ | $60.16_{\pm1.20}$ | $49.38_{\pm22.29}$ | $47.07_{\pm1.84}$ | $48.08_{\pm2.39}$ | $57.22_{\pm3.09}$ | $48.85_{\pm1.22}$ |
| | GraphMAE | $68.64_{\pm0.80}$ | $50.00_{\pm0.00}$ | $60.27_{\pm1.58}$ | $75.23_{\pm2.86}$ | $52.79_{\pm1.96}$ | $52.96_{\pm5.14}$ | $59.08_{\pm2.59}$ | $57.07_{\pm5.22}$ |
| | EdgePreGPPT | $72.39_{\pm1.43}$ | $78.68_{\pm3.14}$ | $59.70_{\pm1.16}$ | $71.74_{\pm1.36}$ | $53.23_{\pm0.59}$ | $55.95_{\pm3.53}$ | $57.16_{\pm1.15}$ | $55.12_{\pm1.37}$ |
| | EdgePreGprompt | $64.81_{\pm4.48}$ | $76.99_{\pm3.95}$ | $59.65_{\pm1.05}$ | $75.64_{\pm1.21}$ | $55.95_{\pm1.04}$ | $61.22_{\pm3.30}$ | $57.14_{\pm3.83}$ | $60.99_{\pm3.42}$ |
| | GCL | $68.50_{\pm0.89}$ | $79.36_{\pm0.64}$ | $58.20_{\pm2.58}$ | $75.42_{\pm2.40}$ | $55.17_{\pm1.18}$ | $57.66_{\pm3.49}$ | $56.26_{\pm0.72}$ | $54.38_{\pm2.29}$ |
| | SimGRACE | $68.88_{\pm0.73}$ | $79.18_{\pm0.70}$ | $59.03_{\pm1.13}$ | $54.34_{\pm11.05}$ | $56.49_{\pm0.82}$ | $59.26_{\pm4.32}$ | $53.33_{\pm0.93}$ | $58.78_{\pm0.91}$ |
| GPF-plus | DGI | $67.30_{\pm2.04}$ | $78.76_{\pm3.06}$ | $59.98_{\pm1.03}$ | $78.67_{\pm3.24}$ | $55.76_{\pm0.99}$ | $58.01_{\pm1.54}$ | $55.60_{\pm4.64}$ | $53.17_{\pm0.34}$ |
| | GraphMAE | $66.86_{\pm1.34}$ | $50.00_{\pm0.00}$ | $61.20_{\pm1.57}$ | $72.94_{\pm0.86}$ | $49.96_{\pm0.80}$ | $47.18_{\pm2.64}$ | $54.12_{\pm12.01}$ | $59.32_{\pm3.11}$ |
| | EdgePreGPPT | $68.32_{\pm1.40}$ | $79.72_{\pm3.57}$ | $61.31_{\pm2.41}$ | $68.83_{\pm3.06}$ | $55.99_{\pm0.45}$ | $61.66_{\pm1.49}$ | $51.94_{\pm1.06}$ | $49.50_{\pm2.07}$ |
| | EdgePreGprompt | $66.08_{\pm4.10}$ | $78.31_{\pm2.86}$ | $58.73_{\pm2.17}$ | $51.73_{\pm17.01}$ | $57.28_{\pm2.04}$ | $59.88_{\pm2.24}$ | $52.69_{\pm2.74}$ | $55.53_{\pm2.48}$ |
| | GCL | $67.76_{\pm1.67}$ | $78.53_{\pm0.70}$ | $57.90_{\pm2.40}$ | $77.86_{\pm2.73}$ | $53.24_{\pm1.85}$ | $61.30_{\pm2.87}$ | $50.97_{\pm1.94}$ | $52.80_{\pm2.36}$ |
| | SimGRACE | $67.38_{\pm2.08}$ | $79.41_{\pm0.77}$ | $53.05_{\pm5.96}$ | $49.04_{\pm15.66}$ | $55.30_{\pm1.21}$ | $59.92_{\pm3.03}$ | $52.68_{\pm1.00}$ | $51.35_{\pm2.70}$ |

Table 26: Graph classification performance Accuracy (%) (5-shot)

| Training schemes | Methods | IMDB-B | COLLAB | PROTEINS | MUTAG | ENZYMES | COX2 | BZR | DD |
|---|---|---|---|---|---|---|---|---|---|
| self-supervise | GCN | $62.60_{\pm4.01}$ | $55.23_{\pm4.26}$ | $62.90_{\pm5.03}$ | $73.47_{\pm3.92}$ | $25.67_{\pm0.48}$ | $64.99_{\pm10.42}$ | $51.48_{\pm2.29}$ | $63.59_{\pm2.86}$ |
| Pretrain + Fine-tuning | DGI | $50.80_{\pm0.88}$ | $60.72_{\pm2.09}$ | $62.47_{\pm4.25}$ | $74.53_{\pm3.71}$ | $25.17_{\pm1.67}$ | $68.36_{\pm11.83}$ | $24.94_{\pm7.80}$ | $58.94_{\pm9.66}$ |
| | GraphMAE | $55.75_{\pm5.48}$ | $54.40_{\pm5.15}$ | $61.21_{\pm6.92}$ | $67.33_{\pm5.61}$ | $26.13_{\pm1.09}$ | $68.42_{\pm11.71}$ | $38.27_{\pm18.11}$ | $64.71_{\pm3.22}$ |
| | EdgePreGPPT | $60.97_{\pm10.07}$ | $57.44_{\pm3.75}$ | $62.85_{\pm4.63}$ | $70.80_{\pm3.05}$ | $27.04_{\pm1.75}$ | $69.38_{\pm10.91}$ | $58.40_{\pm12.79}$ | $64.52_{\pm2.97}$ |
| | EdgePreGprompt | $62.88_{\pm4.02}$ | $58.11_{\pm3.76}$ | $60.25_{\pm4.96}$ | $73.20_{\pm2.44}$ | $27.46_{\pm1.29}$ | $73.19_{\pm9.53}$ | $34.20_{\pm16.57}$ | $62.70_{\pm3.37}$ |
| | GCL | $65.40_{\pm3.33}$ | $48.28_{\pm6.30}$ | $63.33_{\pm4.13}$ | $75.33_{\pm1.89}$ | $24.63_{\pm1.52}$ | $55.44_{\pm3.84}$ | $72.96_{\pm11.98}$ | $63.84_{\pm2.06}$ |
| | SimGRACE | $60.65_{\pm4.54}$ | $48.40_{\pm5.90}$ | $63.08_{\pm4.66}$ | $71.73_{\pm2.33}$ | $24.79_{\pm1.83}$ | $61.50_{\pm16.06}$ | $70.80_{\pm16.29}$ | $62.25_{\pm3.23}$ |
| GPPTPrompt | DGI | $49.90_{\pm0.84}$ | $48.54_{\pm9.22}$ | $54.49_{\pm9.48}$ | $70.53_{\pm3.90}$ | $19.54_{\pm3.17}$ | $47.08_{\pm25.71}$ | $54.75_{\pm19.07}$ | $52.65_{\pm9.39}$ |
| | GraphMAE | $49.60_{\pm0.75}$ | $33.61_{\pm16.35}$ | $58.27_{\pm4.63}$ | $69.20_{\pm3.30}$ | $22.17_{\pm2.34}$ | $56.14_{\pm25.93}$ | $69.63_{\pm14.96}$ | $57.07_{\pm5.16}$ |
| | EdgePreGPPT | $66.37_{\pm3.59}$ | $50.16_{\pm9.23}$ | $57.37_{\pm4.85}$ | $65.33_{\pm3.65}$ | $21.33_{\pm1.04}$ | $60.05_{\pm13.16}$ | $45.37_{\pm19.12}$ | $51.36_{\pm4.26}$ |
| | EdgePreGprompt | $49.60_{\pm0.75}$ | $37.57_{\pm10.07}$ | $46.92_{\pm11.86}$ | $40.67_{\pm14.47}$ | $21.96_{\pm2.12}$ | $67.88_{\pm17.34}$ | $48.02_{\pm12.75}$ | $60.02_{\pm3.24}$ |
| | GCL | $59.58_{\pm9.19}$ | $55.21_{\pm1.21}$ | $58.25_{\pm3.13}$ | $70.40_{\pm4.10}$ | $20.92_{\pm2.40}$ | $64.50_{\pm19.35}$ | $56.11_{\pm21.68}$ | $57.05_{\pm3.15}$ |
| | SimGRACE | $60.35_{\pm8.28}$ | $54.05_{\pm4.58}$ | $56.85_{\pm10.85}$ | $68.67_{\pm5.98}$ | $20.96_{\pm1.23}$ | $60.80_{\pm13.90}$ | $60.80_{\pm8.71}$ | $56.71_{\pm4.10}$ |
| ALL-in-one | DGI | $60.47_{\pm7.38}$ | $57.59_{\pm5.73}$ | $71.37_{\pm4.89}$ | $84.00_{\pm4.32}$ | $26.46_{\pm2.24}$ | $48.15_{\pm4.32}$ | $61.73_{\pm8.94}$ | $54.59_{\pm2.82}$ |
| | GraphMAE | $59.37_{\pm9.45}$ | $36.86_{\pm13.96}$ | $68.72_{\pm3.98}$ | $82.67_{\pm2.89}$ | $24.00_{\pm0.87}$ | $48.79_{\pm7.42}$ | $57.10_{\pm17.82}$ | $63.44_{\pm1.35}$ |
| | EdgePreGPPT | $63.62_{\pm2.30}$ | $57.86_{\pm5.88}$ | $70.56_{\pm2.54}$ | $84.67_{\pm0.73}$ | $23.92_{\pm2.14}$ | $52.17_{\pm5.59}$ | $62.78_{\pm10.18}$ | $53.93_{\pm4.10}$ |
| | EdgePreGprompt | $60.18_{\pm7.66}$ | $57.63_{\pm7.77}$ | $63.33_{\pm2.98}$ | $80.53_{\pm1.54}$ | $25.00_{\pm1.56}$ | $51.05_{\pm7.26}$ | $49.69_{\pm13.79}$ | $62.78_{\pm0.65}$ |
| | GCL | $61.80_{\pm4.92}$ | $57.04_{\pm4.46}$ | $69.69_{\pm6.19}$ | $75.87_{\pm0.98}$ | $26.71_{\pm2.17}$ | $56.41_{\pm6.50}$ | $51.67_{\pm9.26}$ | $60.62_{\pm3.84}$ |
| | SimGRACE | $63.05_{\pm3.01}$ | $54.01_{\pm0.83}$ | $68.72_{\pm4.97}$ | $75.47_{\pm3.08}$ | $26.67_{\pm0.99}$ | $62.95_{\pm8.57}$ | $53.64_{\pm5.97}$ | $59.32_{\pm0.34}$ |
| Gprompt | DGI | $53.05_{\pm10.49}$ | $60.62_{\pm2.31}$ | $61.30_{\pm3.46}$ | $62.93_{\pm15.66}$ | $21.38_{\pm1.97}$ | $46.54_{\pm7.06}$ | $59.38_{\pm14.43}$ | $54.12_{\pm5.51}$ |
| | GraphMAE | $55.15_{\pm7.10}$ | $32.55_{\pm0.12}$ | $58.00_{\pm7.16}$ | $65.87_{\pm8.50}$ | $19.87_{\pm2.24}$ | $52.23_{\pm5.40}$ | $53.77_{\pm6.74}$ | $50.68_{\pm5.33}$ |
| | EdgePreGPPT | $66.70_{\pm3.87}$ | $57.23_{\pm2.43}$ | $62.94_{\pm1.38}$ | $63.87_{\pm12.45}$ | $17.79_{\pm1.27}$ | $53.35_{\pm4.42}$ | $51.98_{\pm8.17}$ | $52.46_{\pm4.48}$ |
| | EdgePreGprompt | $52.75_{\pm10.26}$ | $57.71_{\pm1.56}$ | $55.55_{\pm6.18}$ | $73.07_{\pm2.13}$ | $20.79_{\pm2.66}$ | $52.12_{\pm7.21}$ | $54.32_{\pm11.18}$ | $56.24_{\pm3.69}$ |
| | GCL | $61.38_{\pm10.05}$ | $60.71_{\pm4.54}$ | $60.54_{\pm2.22}$ | $64.40_{\pm8.95}$ | $20.33_{\pm2.27}$ | $53.24_{\pm10.24}$ | $56.73_{\pm7.80}$ | $58.28_{\pm2.18}$ |
| | SimGRACE | $54.85_{\pm10.71}$ | $60.76_{\pm5.08}$ | $61.62_{\pm2.86}$ | $63.47_{\pm7.90}$ | $21.46_{\pm2.27}$ | $53.35_{\pm7.75}$ | $51.36_{\pm9.14}$ | $54.33_{\pm4.04}$ |
| GPF | DGI | $59.90_{\pm9.95}$ | $59.65_{\pm6.25}$ | $60.99_{\pm4.00}$ | $72.93_{\pm4.14}$ | $24.75_{\pm0.98}$ | $52.98_{\pm15.39}$ | $61.05_{\pm11.51}$ | $58.70_{\pm1.84}$ |
| | GraphMAE | $66.02_{\pm3.96}$ | $29.68_{\pm13.57}$ | $63.37_{\pm4.37}$ | $70.13_{\pm2.58}$ | $23.92_{\pm2.16}$ | $55.87_{\pm15.72}$ | $54.63_{\pm9.43}$ | $60.57_{\pm3.74}$ |
| | EdgePreGPPT | $67.80_{\pm5.58}$ | $58.07_{\pm6.03}$ | $63.28_{\pm4.33}$ | $68.40_{\pm3.44}$ | $24.13_{\pm1.56}$ | $66.27_{\pm14.57}$ | $32.53_{\pm10.35}$ | $55.73_{\pm3.64}$ |
| | EdgePreGprompt | $62.62_{\pm4.73}$ | $56.62_{\pm5.44}$ | $62.34_{\pm3.36}$ | $74.00_{\pm3.65}$ | $24.29_{\pm1.44}$ | $64.77_{\pm7.22}$ | $50.99_{\pm16.41}$ | $61.06_{\pm2.63}$ |
| | GCL | $60.17_{\pm9.46}$ | $56.65_{\pm2.44}$ | $62.34_{\pm4.37}$ | $70.67_{\pm3.89}$ | $27.00_{\pm0.78}$ | $58.28_{\pm9.07}$ | $53.70_{\pm10.84}$ | $59.07_{\pm0.65}$ |
| | SimGRACE | $62.33_{\pm4.53}$ | $55.67_{\pm7.42}$ | $60.74_{\pm3.46}$ | $72.27_{\pm4.51}$ | $26.21_{\pm0.96}$ | $29.81_{\pm15.53}$ | $27.10_{\pm4.50}$ | $43.52_{\pm5.74}$ |
| GPF-plus | DGI | $63.50_{\pm5.68}$ | $54.91_{\pm7.08}$ | $63.51_{\pm2.89}$ | $71.87_{\pm5.55}$ | $25.67_{\pm1.75}$ | $60.16_{\pm3.91}$ | $62.84_{\pm9.18}$ | $62.76_{\pm4.22}$ |
| | GraphMAE | $63.72_{\pm5.43}$ | $37.23_{\pm18.02}$ | $63.15_{\pm4.75}$ | $72.67_{\pm1.89}$ | $24.79_{\pm1.80}$ | $67.35_{\pm13.04}$ | $49.51_{\pm9.62}$ | $59.21_{\pm6.53}$ |
| | EdgePreGPPT | $68.13_{\pm3.31}$ | $60.68_{\pm4.67}$ | $61.89_{\pm4.59}$ | $70.93_{\pm3.00}$ | $23.04_{\pm1.13}$ | $68.20_{\pm12.65}$ | $24.01_{\pm5.97}$ | $64.46_{\pm3.57}$ |
| | EdgePreGprompt | $63.85_{\pm5.88}$ | $57.76_{\pm6.85}$ | $58.83_{\pm2.28}$ | $72.67_{\pm4.04}$ | $26.87_{\pm1.89}$ | $72.87_{\pm10.17}$ | $22.35_{\pm2.72}$ | $64.80_{\pm3.45}$ |
| | GCL | $60.18_{\pm10.28}$ | $57.43_{\pm3.27}$ | $63.08_{\pm4.23}$ | $73.87_{\pm3.51}$ | $25.79_{\pm1.76}$ | $60.70_{\pm11.73}$ | $57.59_{\pm5.89}$ | $52.78_{\pm10.34}$ |
| | SimGRACE | $60.43_{\pm9.87}$ | $58.99_{\pm4.32}$ | $63.10_{\pm4.06}$ | $72.80_{\pm3.30}$ | $24.58_{\pm1.68}$ | $61.77_{\pm16.67}$ | $71.54_{\pm14.81}$ | $61.97_{\pm3.07}$ |

Table 27: Graph classification performance F1-score (5-shot)

| Training schemes | Methods | IMDB-B | COLLAB | PROTEINS | MUTAG | ENZYMES | COX2 | BZR | DD |
|---|---|---|---|---|---|---|---|---|---|
| self-supervise | GCN | $61.53_{\pm3.82}$ | $46.72_{\pm7.81}$ | $60.12_{\pm2.97}$ | $68.77_{\pm4.51}$ | $21.36_{\pm3.02}$ | $50.40_{\pm4.10}$ | $49.85_{\pm2.02}$ | $61.16_{\pm2.22}$ |
| Pretrain + Fine-tuning | DGI | $37.27_{\pm7.49}$ | $60.77_{\pm1.97}$ | $59.48_{\pm2.21}$ | $69.60_{\pm5.01}$ | $23.93_{\pm3.09}$ | $46.29_{\pm3.39}$ | $21.92_{\pm9.07}$ | $53.50_{\pm13.04}$ |
| | GraphMAE | $48.52_{\pm12.68}$ | $47.93_{\pm10.31}$ | $58.76_{\pm5.50}$ | $61.19_{\pm5.53}$ | $21.10_{\pm2.81}$ | $46.30_{\pm3.29}$ | $34.64_{\pm16.14}$ | $60.62_{\pm2.52}$ |
| | EdgePreGPPT | $59.95_{\pm10.96}$ | $55.01_{\pm7.38}$ | $58.97_{\pm1.92}$ | $66.40_{\pm3.88}$ | $23.13_{\pm4.27}$ | $46.90_{\pm4.51}$ | $49.13_{\pm5.88}$ | $60.32_{\pm3.32}$ |
| | EdgePreGprompt | $61.87_{\pm3.87}$ | $58.08_{\pm3.74}$ | $57.72_{\pm3.83}$ | $67.43_{\pm5.46}$ | $24.23_{\pm3.11}$ | $45.22_{\pm2.84}$ | $29.92_{\pm15.69}$ | $55.91_{\pm9.68}$ |
| | GCL | $64.48_{\pm3.90}$ | $46.25_{\pm8.10}$ | $60.16_{\pm1.94}$ | $72.18_{\pm2.14}$ | $20.23_{\pm2.20}$ | $50.69_{\pm2.30}$ | $44.56_{\pm0.94}$ | $61.72_{\pm1.92}$ |
| | SimGRACE | $58.80_{\pm4.71}$ | $46.94_{\pm6.45}$ | $60.31_{\pm2.61}$ | $67.55_{\pm3.52}$ | $19.51_{\pm3.69}$ | $50.79_{\pm10.90}$ | $42.64_{\pm2.95}$ | $54.10_{\pm1.53}$ |
| GPPTPrompt | DGI | $33.29_{\pm0.37}$ | $44.26_{\pm14.53}$ | $51.17_{\pm11.32}$ | $62.76_{\pm9.57}$ | $15.91_{\pm3.76}$ | $31.80_{\pm11.62}$ | $40.57_{\pm12.66}$ | $49.10_{\pm8.69}$ |
| | GraphMAE | $33.15_{\pm0.33}$ | $16.01_{\pm6.20}$ | $56.63_{\pm5.43}$ | $61.64_{\pm4.83}$ | $18.70_{\pm2.65}$ | $36.00_{\pm12.12}$ | $45.03_{\pm4.50}$ | $48.97_{\pm2.70}$ |
| | EdgePreGPPT | $65.92_{\pm3.82}$ | $44.90_{\pm15.76}$ | $39.47_{\pm4.13}$ | $60.33_{\pm5.13}$ | $17.40_{\pm1.52}$ | $45.39_{\pm5.02}$ | $38.16_{\pm10.87}$ | $50.83_{\pm4.76}$ |
| | EdgePreGprompt | $33.15_{\pm0.33}$ | $19.75_{\pm6.76}$ | $44.10_{\pm11.48}$ | $33.67_{\pm17.23}$ | $19.46_{\pm2.35}$ | $42.06_{\pm4.74}$ | $41.95_{\pm7.64}$ | $55.63_{\pm2.35}$ |
| | GCL | $55.16_{\pm14.72}$ | $55.24_{\pm1.26}$ | $40.33_{\pm5.84}$ | $65.16_{\pm6.09}$ | $19.05_{\pm1.73}$ | $46.02_{\pm10.48}$ | $39.32_{\pm8.94}$ | $54.10_{\pm2.94}$ |
| | SimGRACE | $59.66_{\pm9.14}$ | $54.05_{\pm3.81}$ | $54.48_{\pm10.12}$ | $62.48_{\pm5.78}$ | $19.62_{\pm1.57}$ | $49.41_{\pm6.89}$ | $50.14_{\pm3.71}$ | $54.01_{\pm3.13}$ |
| ALL-in-one | DGI | $59.53_{\pm8.82}$ | $57.32_{\pm5.81}$ | $70.60_{\pm4.58}$ | $82.31_{\pm2.87}$ | $17.66_{\pm2.59}$ | $44.71_{\pm2.85}$ | $52.91_{\pm7.30}$ | $52.63_{\pm1.53}$ |
| | GraphMAE | $58.72_{\pm10.43}$ | $17.43_{\pm5.21}$ | $66.84_{\pm3.16}$ | $80.56_{\pm3.23}$ | $15.28_{\pm0.64}$ | $45.02_{\pm5.39}$ | $50.25_{\pm14.39}$ | $57.48_{\pm1.16}$ |
| | EdgePreGPPT | $63.43_{\pm2.34}$ | $57.44_{\pm6.03}$ | $69.85_{\pm2.07}$ | $83.59_{\pm1.08}$ | $18.20_{\pm5.07}$ | $47.22_{\pm3.60}$ | $46.61_{\pm3.47}$ | $53.27_{\pm3.89}$ |
| | EdgePreGprompt | $58.91_{\pm9.07}$ | $56.52_{\pm8.78}$ | $61.72_{\pm1.75}$ | $78.93_{\pm1.51}$ | $19.95_{\pm1.44}$ | $48.65_{\pm5.08}$ | $44.03_{\pm8.76}$ | $59.90_{\pm1.32}$ |
| | GCL | $61.47_{\pm5.33}$ | $56.68_{\pm4.02}$ | $68.30_{\pm5.95}$ | $73.50_{\pm0.83}$ | $22.30_{\pm0.99}$ | $47.14_{\pm2.57}$ | $45.65_{\pm4.16}$ | $53.88_{\pm4.86}$ |
| | SimGRACE | $62.85_{\pm2.99}$ | $55.22_{\pm0.57}$ | $67.22_{\pm4.47}$ | $73.69_{\pm2.49}$ | $21.06_{\pm0.77}$ | $51.71_{\pm4.63}$ | $48.74_{\pm3.72}$ | $40.65_{\pm6.82}$ |
| Gprompt | DGI | $50.47_{\pm12.00}$ | $60.36_{\pm1.92}$ | $58.82_{\pm4.98}$ | $59.86_{\pm14.59}$ | $19.68_{\pm2.10}$ | $43.07_{\pm4.95}$ | $52.76_{\pm11.53}$ | $52.64_{\pm4.28}$ |
| | GraphMAE | $51.26_{\pm10.62}$ | $16.37_{\pm0.04}$ | $56.08_{\pm6.65}$ | $61.49_{\pm7.64}$ | $18.48_{\pm2.23}$ | $46.44_{\pm3.42}$ | $48.92_{\pm4.67}$ | $49.81_{\pm4.14}$ |
| | EdgePreGPPT | $66.49_{\pm4.14}$ | $57.50_{\pm2.41}$ | $61.89_{\pm1.74}$ | $61.80_{\pm11.24}$ | $16.31_{\pm1.68}$ | $48.08_{\pm4.53}$ | $46.79_{\pm5.22}$ | $51.83_{\pm4.02}$ |
| | EdgePreGprompt | $50.18_{\pm11.72}$ | $58.02_{\pm1.57}$ | $53.81_{\pm6.27}$ | $70.82_{\pm1.39}$ | $18.42_{\pm3.25}$ | $46.63_{\pm4.27}$ | $47.81_{\pm6.75}$ | $55.47_{\pm3.78}$ |
| | GCL | $60.64_{\pm11.30}$ | $59.83_{\pm4.00}$ | $57.86_{\pm1.00}$ | $60.49_{\pm10.88}$ | $19.29_{\pm2.26}$ | $47.44_{\pm7.08}$ | $50.30_{\pm4.88}$ | $57.01_{\pm2.25}$ |
| | SimGRACE | $53.16_{\pm12.04}$ | $60.09_{\pm4.57}$ | $59.49_{\pm2.71}$ | $59.62_{\pm6.15}$ | $20.75_{\pm2.19}$ | $49.12_{\pm4.71}$ | $47.73_{\pm7.10}$ | $52.97_{\pm4.17}$ |
| GPF | DGI | $58.67_{\pm10.64}$ | $59.18_{\pm6.42}$ | $58.89_{\pm2.28}$ | $67.69_{\pm4.30}$ | $18.41_{\pm2.04}$ | $43.64_{\pm5.83}$ | $56.38_{\pm8.64}$ | $43.94_{\pm8.16}$ |
| | GraphMAE | $65.59_{\pm4.08}$ | $14.71_{\pm5.24}$ | $60.67_{\pm2.38}$ | $65.88_{\pm1.64}$ | $19.47_{\pm2.56}$ | $45.45_{\pm8.77}$ | $49.37_{\pm6.61}$ | $57.04_{\pm2.26}$ |
| | EdgePreGPPT | $67.73_{\pm5.58}$ | $57.75_{\pm6.17}$ | $61.14_{\pm4.21}$ | $63.56_{\pm4.73}$ | $19.89_{\pm2.46}$ | $43.97_{\pm2.80}$ | $30.06_{\pm8.48}$ | $55.24_{\pm3.51}$ |
| | EdgePreGprompt | $61.77_{\pm4.51}$ | $56.12_{\pm5.23}$ | $60.54_{\pm2.00}$ | $72.05_{\pm3.02}$ | $19.31_{\pm2.59}$ | $43.97_{\pm3.91}$ | $39.07_{\pm4.52}$ | $58.14_{\pm3.07}$ |
| | GCL | $58.58_{\pm10.31}$ | $54.42_{\pm4.20}$ | $58.69_{\pm2.31}$ | $67.61_{\pm3.98}$ | $21.36_{\pm3.31}$ | $50.58_{\pm4.57}$ | $47.55_{\pm4.52}$ | $40.86_{\pm7.24}$ |
| | SimGRACE | $61.27_{\pm4.49}$ | $54.22_{\pm8.92}$ | $59.44_{\pm2.34}$ | $67.56_{\pm6.57}$ | $23.55_{\pm3.05}$ | $25.14_{\pm14.15}$ | $26.14_{\pm5.64}$ | $33.65_{\pm9.49}$ |
| GPF-plus | DGI | $63.03_{\pm5.96}$ | $53.58_{\pm8.50}$ | $59.69_{\pm2.41}$ | $70.16_{\pm4.83}$ | $21.63_{\pm0.48}$ | $52.70_{\pm2.44}$ | $57.31_{\pm6.19}$ | $57.95_{\pm3.57}$ |
| | GraphMAE | $62.85_{\pm5.85}$ | $17.18_{\pm6.87}$ | $60.52_{\pm2.74}$ | $67.74_{\pm2.44}$ | $21.75_{\pm1.83}$ | $45.30_{\pm2.36}$ | $45.36_{\pm7.65}$ | $56.39_{\pm7.29}$ |
| | EdgePreGPPT | $67.66_{\pm3.65}$ | $60.45_{\pm4.30}$ | $58.78_{\pm2.63}$ | $68.26_{\pm2.34}$ | $20.46_{\pm1.22}$ | $45.80_{\pm4.06}$ | $21.07_{\pm7.39}$ | $60.10_{\pm2.62}$ |
| | EdgePreGprompt | $63.38_{\pm6.30}$ | $56.87_{\pm7.08}$ | $55.28_{\pm2.90}$ | $69.63_{\pm3.06}$ | $23.58_{\pm3.05}$ | $44.79_{\pm1.97}$ | $19.35_{\pm3.96}$ | $60.64_{\pm2.13}$ |
| | GCL | $58.81_{\pm10.82}$ | $57.11_{\pm3.43}$ | $60.02_{\pm2.04}$ | $69.79_{\pm2.97}$ | $22.03_{\pm2.03}$ | $47.74_{\pm4.03}$ | $53.04_{\pm4.37}$ | $47.52_{\pm13.56}$ |
| | SimGRACE | $59.10_{\pm10.57}$ | $58.56_{\pm4.61}$ | $60.30_{\pm2.04}$ | $69.16_{\pm2.98}$ | $22.60_{\pm2.36}$ | $51.12_{\pm11.99}$ | $43.15_{\pm1.95}$ | $53.98_{\pm1.48}$ |

Table 28: Graph classification performance AUROC (5-shot)

| Training schemes | Methods | IMDB-B | COLLAB | PROTEINS | MUTAG | ENZYMES | COX2 | BZR | DD |
|---|---|---|---|---|---|---|---|---|---|
| self-supervise | GCN | $66.66_{\pm4.29}$ | $65.98_{\pm11.45}$ | $60.55_{\pm1.27}$ | $68.80_{\pm5.31}$ | $59.06_{\pm0.90}$ | $53.16_{\pm1.88}$ | $63.16_{\pm1.91}$ | $64.75_{\pm3.62}$ |
| Pretrain + Fine-tuning | DGI | $50.02_{\pm0.04}$ | $81.66_{\pm0.61}$ | $60.91_{\pm1.77}$ | $68.55_{\pm5.83}$ | $58.93_{\pm1.27}$ | $51.76_{\pm2.64}$ | $52.81_{\pm2.18}$ | $63.61_{\pm4.41}$ |
| | GraphMAE | $57.68_{\pm8.77}$ | $69.01_{\pm10.12}$ | $59.92_{\pm2.11}$ | $74.33_{\pm4.22}$ | $58.39_{\pm0.82}$ | $51.25_{\pm2.21}$ | $57.13_{\pm7.05}$ | $64.45_{\pm3.57}$ |
| | EdgePreGPPT | $61.63_{\pm14.66}$ | $76.71_{\pm9.50}$ | $60.88_{\pm1.39}$ | $72.84_{\pm3.58}$ | $59.64_{\pm0.79}$ | $53.48_{\pm3.98}$ | $60.58_{\pm6.74}$ | $63.82_{\pm3.08}$ |
| | EdgePreGprompt | $66.78_{\pm4.49}$ | $81.42_{\pm0.65}$ | $56.93_{\pm4.04}$ | $72.62_{\pm1.98}$ | $59.65_{\pm1.08}$ | $51.34_{\pm2.69}$ | $50.75_{\pm1.31}$ | $62.82_{\pm3.74}$ |
| | GCL | $69.68_{\pm5.15}$ | $76.71_{\pm2.70}$ | $61.80_{\pm2.09}$ | $70.63_{\pm4.24}$ | $57.49_{\pm0.69}$ | $55.91_{\pm4.67}$ | $50.86_{\pm1.72}$ | $64.37_{\pm3.44}$ |
| | SimGRACE | $64.69_{\pm7.42}$ | $72.81_{\pm3.58}$ | $61.49_{\pm2.04}$ | $71.48_{\pm3.93}$ | $55.98_{\pm2.05}$ | $61.95_{\pm2.47}$ | $48.92_{\pm2.16}$ | $57.39_{\pm2.98}$ |
| GPPTPrompt | DGI | $50.48_{\pm0.70}$ | $70.62_{\pm7.22}$ | $55.92_{\pm14.32}$ | $79.97_{\pm4.66}$ | $52.20_{\pm2.26}$ | $51.22_{\pm2.96}$ | $47.75_{\pm9.82}$ | $53.81_{\pm9.82}$ |
| | GraphMAE | $50.63_{\pm0.56}$ | $50.56_{\pm3.78}$ | $60.56_{\pm6.96}$ | $77.76_{\pm5.00}$ | $53.63_{\pm1.06}$ | $50.60_{\pm6.83}$ | $53.55_{\pm6.08}$ | $58.60_{\pm5.56}$ |
| | EdgePreGPPT | $69.02_{\pm4.94}$ | $68.78_{\pm9.45}$ | $68.50_{\pm9.36}$ | $72.44_{\pm6.57}$ | $53.43_{\pm1.03}$ | $53.02_{\pm8.32}$ | $47.93_{\pm5.96}$ | $51.76_{\pm5.53}$ |
| | EdgePreGprompt | $50.63_{\pm0.56}$ | $56.85_{\pm1.07}$ | $45.95_{\pm18.22}$ | $24.75_{\pm25.84}$ | $53.66_{\pm1.73}$ | $52.56_{\pm3.29}$ | $49.31_{\pm5.04}$ | $61.01_{\pm3.67}$ |
| | GCL | $59.75_{\pm10.88}$ | $76.73_{\pm1.15}$ | $69.17_{\pm8.03}$ | $77.80_{\pm5.27}$ | $53.37_{\pm1.20}$ | $56.22_{\pm8.12}$ | $50.45_{\pm6.37}$ | $57.89_{\pm3.91}$ |
| | SimGRACE | $61.35_{\pm10.03}$ | $70.93_{\pm4.01}$ | $59.76_{\pm13.58}$ | $77.27_{\pm7.67}$ | $53.63_{\pm1.06}$ | $57.78_{\pm6.95}$ | $59.49_{\pm6.13}$ | $57.49_{\pm4.92}$ |
| ALL-in-one | DGI | $61.66_{\pm12.01}$ | $80.99_{\pm1.96}$ | $79.39_{\pm1.76}$ | $90.73_{\pm0.46}$ | $58.71_{\pm0.88}$ | $49.33_{\pm1.56}$ | $52.97_{\pm11.15}$ | $52.92_{\pm1.58}$ |
| | GraphMAE | $60.44_{\pm13.36}$ | $50.00_{\pm0.00}$ | $73.82_{\pm1.89}$ | $89.59_{\pm1.17}$ | $53.75_{\pm1.13}$ | $54.24_{\pm10.01}$ | $55.73_{\pm11.76}$ | $64.55_{\pm1.63}$ |
| | EdgePreGPPT | $69.53_{\pm3.98}$ | $81.50_{\pm1.45}$ | $77.60_{\pm0.81}$ | $91.73_{\pm0.49}$ | $54.18_{\pm3.09}$ | $52.22_{\pm5.14}$ | $39.69_{\pm2.87}$ | $58.41_{\pm0.60}$ |
| | EdgePreGprompt | $61.78_{\pm11.31}$ | $81.93_{\pm1.28}$ | $66.26_{\pm1.84}$ | $87.58_{\pm0.76}$ | $55.09_{\pm2.15}$ | $61.86_{\pm2.18}$ | $50.64_{\pm4.57}$ | $64.95_{\pm0.92}$ |
| | GCL | $65.55_{\pm5.57}$ | $77.86_{\pm4.19}$ | $73.65_{\pm4.67}$ | $85.46_{\pm1.40}$ | $59.50_{\pm1.10}$ | $49.77_{\pm1.51}$ | $52.63_{\pm2.91}$ | $59.95_{\pm4.37}$ |
| | SimGRACE | $66.52_{\pm3.18}$ | $75.84_{\pm0.40}$ | $77.13_{\pm0.44}$ | $84.04_{\pm2.22}$ | $55.76_{\pm0.82}$ | $62.42_{\pm1.89}$ | $54.60_{\pm2.87}$ | $55.43_{\pm0.66}$ |
| Gprompt | DGI | $53.91_{\pm13.03}$ | $70.52_{\pm5.99}$ | $57.55_{\pm7.36}$ | $64.03_{\pm19.95}$ | $53.87_{\pm2.02}$ | $50.06_{\pm2.93}$ | $56.36_{\pm6.92}$ | $54.21_{\pm4.23}$ |
| | GraphMAE | $55.46_{\pm7.98}$ | $44.83_{\pm0.15}$ | $54.75_{\pm7.17}$ | $63.37_{\pm9.48}$ | $52.82_{\pm1.11}$ | $49.40_{\pm6.97}$ | $54.99_{\pm5.88}$ | $51.72_{\pm5.54}$ |
| | EdgePreGPPT | $70.26_{\pm4.44}$ | $73.55_{\pm4.01}$ | $67.07_{\pm0.80}$ | $66.04_{\pm10.33}$ | $53.32_{\pm2.77}$ | $52.47_{\pm6.25}$ | $52.16_{\pm2.83}$ | $53.26_{\pm4.39}$ |
| | EdgePreGprompt | $56.66_{\pm8.48}$ | $75.59_{\pm3.42}$ | $56.24_{\pm7.10}$ | $74.43_{\pm4.69}$ | $54.57_{\pm2.69}$ | $50.60_{\pm3.01}$ | $53.90_{\pm6.28}$ | $56.36_{\pm5.42}$ |
| | GCL | $52.62_{\pm12.32}$ | $75.45_{\pm3.52}$ | $61.03_{\pm5.77}$ | $64.43_{\pm10.15}$ | $54.04_{\pm2.81}$ | $50.64_{\pm7.63}$ | $53.71_{\pm2.16}$ | $56.22_{\pm3.97}$ |
| | SimGRACE | $57.24_{\pm12.21}$ | $73.41_{\pm3.61}$ | $60.58_{\pm5.34}$ | $64.67_{\pm8.28}$ | $55.02_{\pm1.57}$ | $56.79_{\pm5.66}$ | $56.83_{\pm6.49}$ | $53.99_{\pm6.27}$ |
| GPF | DGI | $61.28_{\pm14.33}$ | $80.91_{\pm1.35}$ | $59.36_{\pm1.27}$ | $68.29_{\pm4.20}$ | $57.38_{\pm0.69}$ | $55.65_{\pm2.89}$ | $69.56_{\pm8.43}$ | $52.10_{\pm1.18}$ |
| | GraphMAE | $71.36_{\pm4.86}$ | $50.00_{\pm0.00}$ | $60.79_{\pm1.35}$ | $73.75_{\pm6.18}$ | $57.51_{\pm1.83}$ | $55.80_{\pm3.40}$ | $57.87_{\pm6.08}$ | $59.78_{\pm6.01}$ |
| | EdgePreGPPT | $72.09_{\pm6.35}$ | $81.60_{\pm1.83}$ | $63.33_{\pm6.33}$ | $71.14_{\pm4.08}$ | $57.09_{\pm1.47}$ | $45.44_{\pm3.51}$ | $40.30_{\pm2.41}$ | $57.56_{\pm1.55}$ |
| | EdgePreGprompt | $66.56_{\pm4.93}$ | $82.11_{\pm0.58}$ | $59.32_{\pm1.46}$ | $78.72_{\pm2.44}$ | $56.88_{\pm0.88}$ | $57.11_{\pm3.21}$ | $38.86_{\pm3.08}$ | $59.75_{\pm5.73}$ |
| | GCL | $61.77_{\pm14.84}$ | $76.32_{\pm2.61}$ | $58.70_{\pm2.93}$ | $75.54_{\pm4.82}$ | $57.23_{\pm1.32}$ | $50.87_{\pm1.24}$ | $54.74_{\pm5.41}$ | $51.02_{\pm2.03}$ |
| | SimGRACE | $66.69_{\pm4.51}$ | $78.63_{\pm3.50}$ | $60.84_{\pm0.83}$ | $73.60_{\pm4.98}$ | $57.64_{\pm0.64}$ | $55.74_{\pm2.01}$ | $44.52_{\pm2.78}$ | $48.14_{\pm1.17}$ |
| GPF-plus | DGI | $67.76_{\pm7.65}$ | $80.06_{\pm1.94}$ | $60.56_{\pm1.11}$ | $75.93_{\pm5.59}$ | $57.99_{\pm1.37}$ | $56.06_{\pm2.35}$ | $67.92_{\pm4.56}$ | $62.75_{\pm3.59}$ |
| | GraphMAE | $68.56_{\pm7.52}$ | $50.00_{\pm0.00}$ | $60.88_{\pm1.41}$ | $74.54_{\pm6.62}$ | $56.78_{\pm1.51}$ | $48.93_{\pm2.41}$ | $54.11_{\pm8.51}$ | $62.56_{\pm3.54}$ |
| | EdgePreGPPT | $71.71_{\pm3.89}$ | $82.51_{\pm0.50}$ | $59.30_{\pm1.96}$ | $77.03_{\pm2.00}$ | $56.17_{\pm0.68}$ | $49.53_{\pm4.57}$ | $49.78_{\pm0.43}$ | $63.49_{\pm4.00}$ |
| | EdgePreGprompt | $67.36_{\pm7.32}$ | $81.68_{\pm0.93}$ | $54.63_{\pm1.53}$ | $75.31_{\pm2.84}$ | $60.04_{\pm0.95}$ | $50.95_{\pm1.90}$ | $48.60_{\pm2.80}$ | $64.20_{\pm3.47}$ |
| | GCL | $61.80_{\pm14.82}$ | $80.57_{\pm0.98}$ | $61.23_{\pm1.64}$ | $69.47_{\pm3.56}$ | $57.95_{\pm1.27}$ | $51.87_{\pm1.80}$ | $64.97_{\pm3.46}$ | $61.94_{\pm4.01}$ |
| | SimGRACE | $61.85_{\pm14.76}$ | $77.71_{\pm3.28}$ | $61.07_{\pm1.57}$ | $70.13_{\pm2.87}$ | $57.30_{\pm0.70}$ | $63.15_{\pm2.98}$ | $49.11_{\pm1.78}$ | $57.21_{\pm2.94}$ |

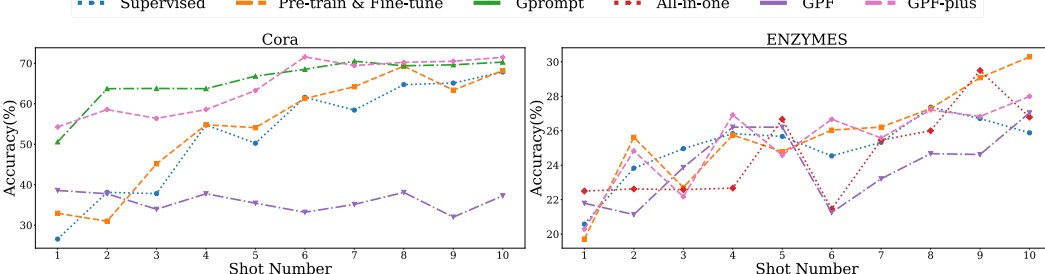

Figure 5: The curve of the shot number from 1 to 10 and the accuracy (%) on Cora and ENZYMES.

Table 29: Node classification performance Accuracy (%) across different graph models on Wisconsin (1-shot setting). The **Accuracy** of supervised methods: **GAT** is $34.51_{\pm 18.02}$, **GraphSAGE** is $25.37_{\pm 5.61}$, **GT** is $20.91_{\pm 7.07}$

| Pre-train & Fine-tune | | | | | | |
|---|---|---|---|---|---|---|
| Model | DGI | GraphMAE | EdgePreGPPT | EdgePreGprompt | GCL | SimGRACE |
| GAT | $16.00_{\pm 6.24}$ | $37.60_{\pm 10.69}$ | $20.00_{\pm 3.82}$ | $33.37_{\pm 4.76}$ | $18.86_{\pm 1.88}$ | $28.00_{\pm 9.40}$ |
| GraphSAGE | $40.69_{\pm 9.46}$ | $43.77_{\pm 12.43}$ | $26.06_{\pm 5.38}$ | $29.94_{\pm 3.75}$ | $36.57_{\pm 4.88}$ | $9.37_{\pm 2.72}$ |
| GT | $25.71_{\pm 3.07}$ | $39.77_{\pm 8.42}$ | $23.20_{\pm 2.65}$ | $28.23_{\pm 6.64}$ | $11.77_{\pm 1.06}$ | $14.51_{\pm 5.08}$ |

| GPPT | | | | | | |
|---|---|---|---|---|---|---|
| Model | DGI | GraphMAE | EdgePreGPPT | EdgePreGprompt | GCL | SimGRACE |
| GAT | $22.17_{\pm 6.13}$ | $33.94_{\pm 7.76}$ | $23.43_{\pm 4.46}$ | $37.94_{\pm 7.11}$ | $26.86_{\pm 6.12}$ | $29.83_{\pm 8.04}$ |
| GraphSAGE | $26.51_{\pm 8.00}$ | $30.51_{\pm 5.40}$ | $21.49_{\pm 5.17}$ | $24.23_{\pm 6.55}$ | $20.91_{\pm 7.11}$ | $25.37_{\pm 7.22}$ |
| GT | $27.20_{\pm 10.48}$ | $29.83_{\pm 5.80}$ | $28.00_{\pm 6.01}$ | $23.31_{\pm 3.01}$ | $27.66_{\pm 0.69}$ | $25.03_{\pm 5.43}$ |

| All-in-one | | | | | | |
|---|---|---|---|---|---|---|
| Model | DGI | GraphMAE | EdgePreGPPT | EdgePreGprompt | GCL | SimGRACE |
| GAT | $69.44_{\pm 5.19}$ | $36.25_{\pm 10.63}$ | $91.25_{\pm 4.33}$ | $92.65_{\pm 3.75}$ | $42.85_{\pm 9.16}$ | $36.61_{\pm 14.86}$ |
| GraphSAGE | $94.73_{\pm 5.86}$ | $97.55_{\pm 0.78}$ | $98.60_{\pm 0.87}$ | $100.00_{\pm 0.00}$ | $83.01_{\pm 18.37}$ | $85.12_{\pm 16.29}$ |
| GT | $68.59_{\pm 11.78}$ | $98.48_{\pm 0.70}$ | $96.74_{\pm 1.01}$ | $100.00_{\pm 0.00}$ | $70.23_{\pm 8.80}$ | $45.70_{\pm 11.83}$ |

| Gprompt | | | | | | |
|---|---|---|---|---|---|---|
| Model | DGI | GraphMAE | EdgePreGPPT | EdgePreGprompt | GCL | SimGRACE |
| GAT | $58.25_{\pm 13.83}$ | $67.77_{\pm 15.91}$ | $94.17_{\pm 2.26}$ | $84.28_{\pm 3.63}$ | $80.11_{\pm 16.65}$ | $57.18_{\pm 12.60}$ |
| GraphSAGE | $66.48_{\pm 12.88}$ | $83.49_{\pm 15.93}$ | $87.52_{\pm 3.79}$ | $82.16_{\pm 2.64}$ | $65.50_{\pm 6.48}$ | $72.61_{\pm 5.97}$ |
| GT | $56.03_{\pm 7.33}$ | $73.50_{\pm 9.72}$ | $76.97_{\pm 13.39}$ | $80.07_{\pm 2.84}$ | $59.31_{\pm 10.17}$ | $69.30_{\pm 10.57}$ |

| GPF | | | | | | |
|---|---|---|---|---|---|---|
| Model | DGI | GraphMAE | EdgePreGPPT | EdgePreGprompt | GCL | SimGRACE |
| GAT | $63.64_{\pm 3.91}$ | $78.59_{\pm 18.57}$ | $97.90_{\pm 0.79}$ | $91.84_{\pm 5.19}$ | $74.93_{\pm 4.49}$ | $65.67_{\pm 15.07}$ |
| GraphSAGE | $68.12_{\pm 13.96}$ | $67.66_{\pm 13.37}$ | $74.06_{\pm 14.59}$ | $72.45_{\pm 10.14}$ | $59.69_{\pm 21.37}$ | $78.37_{\pm 14.84}$ |
| GT | $39.85_{\pm 4.83}$ | $71.26_{\pm 14.43}$ | $72.67_{\pm 13.36}$ | $81.33_{\pm 3.41}$ | $78.19_{\pm 2.19}$ | $67.90_{\pm 10.53}$ |

| GPF-plus | | | | | | |
|---|---|---|---|---|---|---|
| Model | DGI | GraphMAE | EdgePreGPPT | EdgePreGprompt | GCL | SimGRACE |
| GAT | $92.62_{\pm 8.60}$ | $96.37_{\pm 4.44}$ | $99.29_{\pm 1.41}$ | $97.88_{\pm 4.24}$ | $86.12_{\pm 7.73}$ | $94.17_{\pm 2.63}$ |
| GraphSAGE | $77.85_{\pm 3.30}$ | $87.88_{\pm 0.90}$ | $99.30_{\pm 0.86}$ | $96.16_{\pm 5.09}$ | $85.93_{\pm 19.52}$ | $72.47_{\pm 8.33}$ |
| GT | $91.69_{\pm 6.84}$ | $98.60_{\pm 1.02}$ | $99.30_{\pm 0.86}$ | $96.13_{\pm 4.27}$ | $58.83_{\pm 18.53}$ | $89.04_{\pm 4.25}$ |

Table 30: Graph classification performance Accuracy (%) across different graph models on PRO-TEINS (1-shot setting). The **Accuracy** of supervised methods: **GAT** is $48.34_{\pm 9.96}$, **GraphSAGE** is $60.54_{\pm 2.95}$, **GT** is $61.44_{\pm 2.48}$

| Pre-train & Fine-tune | | | | | | |
|---|---|---|---|---|---|---|
| Model | DGI | GraphMAE | EdgePreGPPT | EdgePreGprompt | GCL | SimGRACE |
| GAT | $58.34_{\pm 6.52}$ | $61.06_{\pm 4.13}$ | $63.75_{\pm 3.71}$ | $54.09_{\pm 4.03}$ | $60.04_{\pm 3.06}$ | $58.65_{\pm 6.71}$ |
| GraphSAGE | $60.70_{\pm 4.08}$ | $60.56_{\pm 5.12}$ | $61.60_{\pm 1.78}$ | $63.21_{\pm 1.80}$ | $61.80_{\pm 3.77}$ | $58.56_{\pm 1.84}$ |
| GT | $53.87_{\pm 4.81}$ | $60.00_{\pm 3.99}$ | $64.92_{\pm 3.19}$ | $56.58_{\pm 3.28}$ | $62.88_{\pm 1.82}$ | $60.00_{\pm 1.60}$ |

| GPPT | | | | | | |
|---|---|---|---|---|---|---|
| Model | DGI | GraphMAE | EdgePreGPPT | EdgePreGprompt | GCL | SimGRACE |
| GAT | $57.71_{\pm 8.98}$ | $57.80_{\pm 10.55}$ | $58.04_{\pm 9.92}$ | $54.97_{\pm 7.45}$ | $52.29_{\pm 7.83}$ | $55.15_{\pm 9.84}$ |
| GraphSAGE | $56.56_{\pm 6.73}$ | $57.73_{\pm 7.95}$ | $58.63_{\pm 11.78}$ | $56.94_{\pm 5.67}$ | $58.00_{\pm 7.80}$ | $54.74_{\pm 6.59}$ |
| GT | $53.08_{\pm 7.56}$ | $57.35_{\pm 8.58}$ | $60.27_{\pm 3.92}$ | $55.51_{\pm 7.68}$ | $56.18_{\pm 5.79}$ | $55.87_{\pm 7.69}$ |

| All-in-one | | | | | | |
|---|---|---|---|---|---|---|
| Model | DGI | GraphMAE | EdgePreGPPT | EdgePreGprompt | GCL | SimGRACE |
| GAT | $60.04_{\pm 3.84}$ | $60.00_{\pm 6.04}$ | $62.11_{\pm 2.85}$ | $63.21_{\pm 2.22}$ | $58.36_{\pm 4.93}$ | $59.37_{\pm 5.59}$ |
| GraphSAGE | $59.53_{\pm 4.94}$ | $60.70_{\pm 4.89}$ | $63.12_{\pm 1.59}$ | $59.98_{\pm 8.46}$ | $62.22_{\pm 3.81}$ | $62.04_{\pm 2.07}$ |
| GT | $57.39_{\pm 3.66}$ | $58.92_{\pm 6.61}$ | $62.61_{\pm 4.08}$ | $60.20_{\pm 7.55}$ | $62.81_{\pm 1.63}$ | $50.52_{\pm 6.17}$ |

| Gprompt | | | | | | |
|---|---|---|---|---|---|---|
| Model | DGI | GraphMAE | EdgePreGPPT | EdgePreGprompt | GCL | SimGRACE |
| GAT | $61.08_{\pm 6.19}$ | $63.03_{\pm 2.61}$ | $64.47_{\pm 4.30}$ | $61.48_{\pm 3.34}$ | $59.12_{\pm 6.84}$ | $58.13_{\pm 7.27}$ |
| GraphSAGE | $61.35_{\pm 2.21}$ | $59.48_{\pm 9.19}$ | $60.92_{\pm 3.16}$ | $63.30_{\pm 1.43}$ | $55.26_{\pm 2.61}$ | $63.21_{\pm 2.66}$ |
| GT | $56.65_{\pm 5.81}$ | $60.99_{\pm 1.62}$ | $61.87_{\pm 5.60}$ | $55.33_{\pm 3.69}$ | $54.81_{\pm 7.62}$ | $58.97_{\pm 1.16}$ |

| GPF | | | | | | |
|---|---|---|---|---|---|---|
| Model | DGI | GraphMAE | EdgePreGPPT | EdgePreGprompt | GCL | SimGRACE |
| GAT | $66.11_{\pm 5.18}$ | $65.62_{\pm 5.38}$ | $56.38_{\pm 8.64}$ | $65.55_{\pm 6.65}$ | $59.71_{\pm 7.66}$ | $67.42_{\pm 6.26}$ |
| GraphSAGE | $64.20_{\pm 7.63}$ | $67.87_{\pm 4.32}$ | $58.18_{\pm 9.06}$ | $64.49_{\pm 6.80}$ | $60.25_{\pm 2.91}$ | $62.94_{\pm 2.29}$ |
| GT | $65.80_{\pm 7.42}$ | $60.16_{\pm 5.81}$ | $64.54_{\pm 7.18}$ | $61.21_{\pm 2.91}$ | $58.74_{\pm 5.51}$ | $59.57_{\pm 2.93}$ |

| GPF-plus | | | | | | |
|---|---|---|---|---|---|---|
| Model | DGI | GraphMAE | EdgePreGPPT | EdgePreGprompt | GCL | SimGRACE |
| GAT | $56.20_{\pm 12.87}$ | $57.35_{\pm 11.28}$ | $56.25_{\pm 8.61}$ | $53.24_{\pm 4.79}$ | $57.48_{\pm 11.74}$ | $57.48_{\pm 9.63}$ |
| GraphSAGE | $56.22_{\pm 9.08}$ | $57.55_{\pm 10.56}$ | $56.31_{\pm 9.26}$ | $57.71_{\pm 9.60}$ | $53.89_{\pm 9.47}$ | $55.89_{\pm 4.30}$ |
| GT | $53.39_{\pm 5.23}$ | $57.37_{\pm 10.95}$ | $57.39_{\pm 11.88}$ | $52.61_{\pm 5.30}$ | $57.62_{\pm 12.27}$ | $56.16_{\pm 5.07}$ |

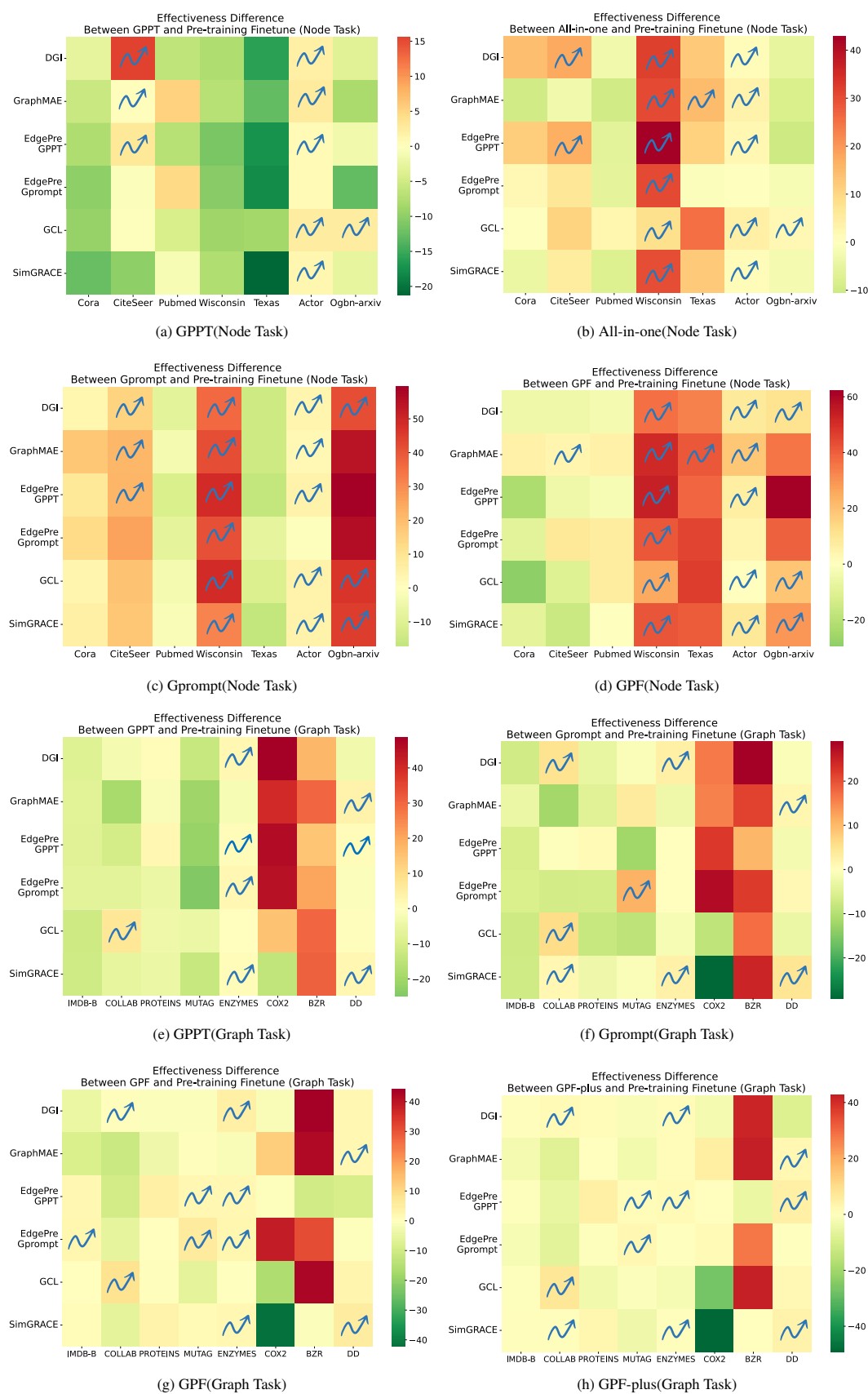

Figure 6: Heat maps of graph prompt methods on different node/graph-level datasets in 1-shot settings, excluding GPF-plus on the node task and All-in-one on the graph tasks.

