# OpenReview forum: "ProG: A Graph Prompt Learning Benchmark"
_NeurIPS.cc/2024/Datasets_and_Benchmarks_Track — NeurIPS 2024 Track Datasets and Benchmarks Poster_

### Official Review · Reviewer_AAD3 · 2024-06-14

**Rating:** 5
**Confidence:** 5
**Correctness:** Yes
**Clarity:** yes

**Review:**

strength:

1. ProG includes a wide range of pre-training methods and graph prompt techniques, evaluated on fifteen diverse datasets, providing a thorough assessment of graph prompt learning methods across various domains and scenarios.

2. The benchmark introduces a unified framework that categorizes graph prompts into two main approaches, enhancing the understanding and comparison of different graph prompt techniques and making the benchmark compatible with most existing methods.

3. ProG is publicly available as an open-source library, promoting transparency, reproducibility, and ease of use for researchers. The library streamlines the execution of various graph prompt models, facilitating objective evaluations and broader exploration in the field.


Weakness:

There is unknown usage of the graph prompt. The major difference between the NLP domain and the graph domain is that the NLP domain is majorly to customer while graph is more about to business. We do not need a user prompt to interact with users.  Moreover, different from the LLM with one model serves for all, one particular graph generally does not have multiple downstream tasks. For the current benchmark dataset utilized in the paper, there is no with multiple downstream tasks. Moreover, LLM is large and quit expensive to fine-tune. In contrast, graph model is small and dataset specific. Current pipeline requires to pre-train on each dataset, and fine-tune on each dataset.

It is also unclear about the usage about the zero/few shot evaluation. The current graph learning is still end-to-end training. Specifically, the node classification adapts the semi-supervised training requires only a few labeled data to achieve good performance (less than 300 nodes). However, the current prompt only focuses on the poor performance of a small graph SSL model.


---After rebuttal--

The authors solved none of my concerns during the rebuttal. Instead of bringing new insights to convince me, the authors just repeat their original contribution again and again. As other reviewers strongly champion for the paper. I will not argue for rejection.

**Strengths:**

1. ProG includes a wide range of pre-training methods and graph prompt techniques, evaluated on fifteen diverse datasets, providing a thorough assessment of graph prompt learning methods across various domains and scenarios.

2. The benchmark introduces a unified framework that categorizes graph prompts into two main approaches, enhancing the understanding and comparison of different graph prompt techniques and making the benchmark compatible with most existing methods.

3. ProG is publicly available as an open-source library, promoting transparency, reproducibility, and ease of use for researchers. The library streamlines the execution of various graph prompt models, facilitating objective evaluations and broader exploration in the field.

**Additional Feedback:**

No

**Documentation:**

Yes

**Opportunities For Improvement:**

There is unknown usage of the graph prompt. The major difference between the NLP domain and the graph domain is that the NLP domain is majorly to customer while graph is more about to business. We do not need a user prompt to interact with users.  Moreover, different from the LLM with one model serves for all, one particular graph generally does not have multiple downstream tasks. For the current benchmark dataset utilized in the paper, there is no with multiple downstream tasks. It remains concern why we require the learning model pre-training on the citation network to transfer to the molecular graph. Moreover, LLM is large and quit expensive to fine-tune. In contrast, graph model is small and dataset specific. Current pipeline requires to pre-train on each dataset, and fine-tune on each dataset.

It is also unclear about the usage about the zero/few shot evaluation. The current graph learning is still end-to-end training. Specifically, the node classification adapts the semi-supervised training requires only a few labeled data to achieve good performance (less than 300 nodes). However, the current prompt only focuses on the poor performance of a small graph SSL model.

**Relation To Prior Work:**

Yes

**Summary And Contributions:**

This paper introduces ProG, a comprehensive benchmark designed for evaluating graph prompt learning methods. ProG integrates six pre-training methods and five state-of-the-art graph prompt techniques, evaluated across fifteen diverse datasets. The benchmark aims to address critical challenges in graph prompt learning, such as unifying diverse models, evaluating the quality of graph prompts, and improving their usability for practical comparisons and selections.

---

> ### Author Rebuttal · Authors · 2024-08-14
>
> ### Further Clarification of Graph Prompt and Its Applicability
> We appreciate your insights and would like to take this opportunity to further clarify the concept and usage of graph prompts. For the **importance and usage** of graph prompts, we agree with you that graphs are more often used in business end and less in customer end. However, there are diverse applications in business end ranging from recommendation systems[5,6,7], protein prediction[8,9], social networks[10,11,15], dynamic graphs[12,13] to molecular property prediction [4,16], designing a universal graph prompt capable of handling **complex graph data from different domains** presents a significant challenge, especially when compared to the relatively straightforward task of dealing with uniformly formatted text data. Currently, graph prompts are recognized as effective tools for knowledge transfer, as evidenced by the community's acknowledgment of papers like **`All-in-One`** [2] proposed at KDD 2023 (**the Best Paper award**) and **`GPF-plus`**[4] proposed at **Neurips 2023**. These prompts have been widely studied in various graph-related applications.
>
> The concept of NLP prompts emerged following the advent of LLMs. In contrast, the research on graph prompts and graph foundation models has been conducted simultaneously, aiming to achieve the *all-in-one and one-for-all* paradigm in LLMs. This paradigm involves pre-training a foundation model using various contexts and then generalizing and applying the learned knowledge to a wide spectrum of downstream tasks within the graph domain. However, replicating this success in the graph domain is challenging, particularly when addressing cross-domain, cross-task, and cross-modal issues. Although the research on graph foundation models is still in its early stages, many efforts are already dedicated to achieving a true graph foundation model [1]. `All-in-One` [2] is the first paper to unify various graph tasks as the graph classification task via induced graph, and the proposed graph prompt is validated effectiveness for making pre-trained graph models adapt to multiple downstream tasks. `GCOPE` [3], a cross-domain graph pre-training framework proposed in KDD 2024, can pre-train a graph model on various graph datasets from different domains and the pre-trained knowledge can be transferred to various downstream datasets. Moreover, **extensive research is being conducted to make a true graph foundation model a reality, ensuring it is grounded in practicality rather than just a theoretical concept [1,17], which involves addressing previously mentioned issues such as cross-domain, cross-task, and cross-modal challenges.**
>
> Besides, when developing a graph foundation model, another critical challenge is the issue of negative transfer when transferring the pre-trained knowledge, particularly in few/zero-shot downstream scenarios. Honestly, we acknowledge that in scenarios where a relatively large number of labeled samples are available (e.g., 300 nodes as you mentioned), traditional semi-supervised learning (SSL) models may indeed perform better than `pre-train & prompting` models. In fact, we have included 5-shot experiments in Appendix E.2, demonstrating the diminishing effect of prompts as the number of labeled samples increases. However, few-shot learning performance is crucial for evaluating a graph foundation model. First, it indicates whether the model possesses strong knowledge transfer and reasoning capabilities. Second, few-shot learning has long been an important topic in graph communities, where graph prompt learning has been regarded as an advanced technique [18,19,20,21]. To better understand the importance of few-shot scenarios, we would like to take an example in network fault diagnosis, where the occurrence of faults is exceptionally rare. The scarcity of fault data makes it challenging to train models that can accurately diagnose faults when they do occur.
>
> Overall, in this paper, we focus on benchmarking various graph prompt and pre-training methods to investigate the prompting ability to various pre-trained models. Additionally, we are glad to offer 'ProG,' an easy-to-use open-source library for conducting various graph prompt models. This project has made a significant contribution to the graph community, and it has already garnered 472 stars on GitHub.

---

> > ### Author Rebuttal · Authors · 2024-08-16
> >
> > Dear Reviewer AAD3,
> >
> > Thank you for revisiting our submission. we understand that there might have been some misunderstanding regarding the contributions and novelty of our paper and the score was changed from 4 to 2 after our rebuttal. We will try our best to eliminate the misunderstandings via the following responses and sincerely hope that you raise any further questions if you still require clarification on our answers.
> >
> > Best regards,
> > Authors

---

> ### Author Rebuttal · Authors · 2024-08-14
>
> [1] Shi, Chuan, et al. "Graph foundation model." Frontiers of Computer Science 18.6 (2024): 186355
>
> [2] All in one and one for all: A simple yet effective method towards cross-domain graph pretraining. KDD 2024
>
> [3] All in One: Multi-task Prompting for Graph Neural Networks. KDD 2023 (Best Paper Award)
>
> [4] Universal prompt tuning for graph neural networks. NeurIPS 2023
>
> [5] GraphPro: Graph Pre-training and Prompt Learning for Recommendation. WWW 2024
>
> [6] Motif-Based Prompt Learning for Universal Cross-Domain Recommendation. WSDM 2024
>
> [7] An Empirical Study Towards Prompt-Tuning for Graph Contrastive Pre-Training in Recommendations. NeurIPS 2023
>
> [8] DDIPrompt: Drug-Drug Interaction Event Prediction based on Graph Prompt Learning. CIKM 2024
>
> [9] Protein Multimer Structure Prediction via PPI-guided Prompt Learning. ICLR 2024
>
> [10] Prompt-and-Align: Prompt-Based Social Alignment for Few-Shot Fake News Detection. CIKM 2023
>
> [11] Voucher Abuse Detection with Prompt-based Fine-tuning on Graph Neural Networks. CIKM 2023
>
> [12] Hu, Junfeng, et al. "Prompt-Enhanced Spatio-Temporal Graph Transfer Learning." arXiv preprint arXiv:2405.12452 (2024)
>
> [13] Xingtong Yu, et al. "DyGPrompt: Learning Feature and Time Prompts on Dynamic Graphs." arXiv preprint arXiv:2405.13937(2024)
>
> [14] Prompt Learning on Temporal Interaction Graphs
>
> [15] HGPrompt: Bridging Homogeneous and Heterogeneous Graphs for Few-Shot Prompt Learning. AAAI 2024
>
> [16] Inductive Graph Alignment Prompt: Bridging the Gap between Graph Pre-training and Inductive Fine-tuning From Spectral Perspective. WWW 2024
>
> [17] Zhao, Ziwen, et al. "A Survey on Self-Supervised Pre-Training of Graph Foundation Models: A Knowledge-Based Perspective." arXiv preprint arXiv:2403.16137 (2024)
>
> [18] Yu, Xingtong, et al. "Few-shot learning on graphs: from meta-learning to pre-training and prompting." arXiv preprint arXiv:2402.01440 (2024)
>
> [19] Ju, Wei, et al. "A survey of data-efficient graph learning." arXiv preprint arXiv:2402.00447 (2024)
>
> [20] Virtual node tuning for few-shot node classification. KDD 2023
>
> [21] Graph Few-shot Class-incremental Learning. WSDM 2022

---

> > ### Author Response · Authors · 2024-08-26
> > **Friendly reminder for rebuttal**
> >
> > Dear Reviewer AAD3,
> >
> > This is a friendly reminder that we would very much value your input on our rebuttal. Your comments have been extremely helpful in improving our paper.
> >
> > With only **several days remaining** before the conclusion of the discussion phase, we wish to extend a respectful request for your feedback about our rebuttal. We would be happy to provide more details and clarifications.
> >
> > Thanks for your attention and best regards.
> >
> >
> > Best regards,
> >
> > Authors

---

> ### Author Response · Authors · 2024-08-28
> **Follow-up Reminder for Rebuttal**
>
> Dear Reviewer AAD3,
>
> This is a friendly reminder that we greatly value your input on our rebuttal. Your comments have been extremely helpful in improving our paper.
>
> With only 3 days remaining before the conclusion of the discussion phase, we wish to extend a respectful request for your feedback about our rebuttal. We would be happy to provide more details and clarifications. Thanks a lot.
>
> Best regards,
>
> Authors

---

> ### Author Response · Authors · 2024-08-29
> **Follow-up Reminder for Rebuttal**
>
> Dear Reviewer AAD3,
>
> As the deadline for the NeurIPS 2024 rebuttal period is approaching, we look forward to hearing your feedback and hope that our responses have adequately addressed your previous concerns. We are eager to address any remaining concerns that you may still have.
>
> Thanks,
>
> Authors

---

> ### Author Response · Authors · 2024-08-30
> **Friendly reminder for rebuttal**
>
> Dear Reviewer AAD3,
>
> With only 1 days remaining before the conclusion of the discussion phase, we look forward to hearing your feedback and hope that our responses have adequately addressed your previous concerns. We are eager to address any remaining concerns that you may still have.
>
> Thanks,
>
> Authors

---

> > ### Comment · Reviewer_AAD3 · 2024-08-30
> >
> > After rebuttal, the authors do not solve my concerns 1. all the methods performance are lower than a semi-supervised learning setting, which may only need hundred cases in one graphs.  Therefore, I do not understand why we require few-shot learning capability with sophisticated design but no performance gain and less efficiency 2. The authors also mention that the graph prompt indicates the reasoning ability on graph. However, there is no graph reasoning tasks are conducted 3. The authors mention that designing a universal graph prompt capable of handling complex graph data from different domains presents a significant challenge.  I do not know why we want to build such a model to handle different domains without any performance gain over the semi-supervised learning model.

---

> > > ### Author Rebuttal · Authors · 2024-09-01
> > >
> > > We hope the following clarification can address your concerns:
> > >
> > > > 1.All the methods performance are lower than a semi-supervised learning setting, which may only need hundred cases in one graphs. Therefore, I do not understand why we require few-shot learning capability with sophisticated design but no performance gain and less efficiency.
> > >
> > >
> > > 1. Firstly, In table 2 and table 3 of our article, our results indicate that **all the graph prompt approaches outperform traditional supervised method**, which is the semi-supervised method like GCN you mentioned. As you mentioned that our method is not as good as semi-supervised learning, we suspect it might be because you compared our method with semi-supervised models in the SSL setting, such as a 20-shot, whereas our setting is 1-shot. Therefore, it's quite normal that the absolute performance is not as good as SSL. However, what we should compare is the relative performance under the same setting.
> > >
> > > 2. We would like to clarify the differences between our few-shot and the traditional semi-supervised learning setting, using Cora as an example, under 1-shot learning scenario, we only have 1 sample per class [1]. This is in contrast to the traditional semi-supervised setting used in models like vanilla GNNs, which typically involve 140 labeled samples (20 samples per class across 7 classes).
> > >
> > > 3. The semi-supervised learning setting assumes that hundreds of labeled samples are available. However, in many real-world applications, **label annotation** is not only **costly** but also **time-consuming**. For example, in scenarios like anti-money laundering, labeling a node as an anomaly often requires a court ruling, making it extremely difficult to accumulate hundreds of labeled cases quickly. This highlights the importance of few-shot learning, where we must operate with very limited labeled data. Our key point is that in many cases, collecting such hundreds of labeled examples is **impractical** or **impossible**. In these situations, few-shot learning becomes not only **relevant** but **essential**[2][3].
> > >
> > >
> > >
> > > > 2.The authors also mention that the graph prompt indicates the reasoning ability on graph. However, there is no graph reasoning tasks are conducted.
> > >
> > > For your second concern, graph reasoning is indeed a highly promising area for future research, especially in graph-based tasks. Rather than exploring future research directions, our proposed benchmark is designed to fairly **compare the performance of various graph prompt methods** based on existing research (i.e., node & graph tasks). This can help understand the real-world performance of graph prompt methods in various scenarios (e.g., different pre-training methods), providing better insights.
> > >
> > > Reasoning is indeed a critical capability that can be explored further. While our current work does not include explicit reasoning experiments, we believe that reasoning tasks are not only aligned with the objectives of few-shot learning but also represent a natural progression in the development of graph prompt. For example, consider the task of **multi-hop relation prediction** in knowledge graphs. In this scenario, the goal is to predict the relationship between two entities via multi-hop reasoning, such as determining whether one person is the boss of another or if they are colleagues. This task inherently involves reasoning, as the model must infer the correct relationship from the patterns and connections present in the graph. Graph prompt can enhance this reasoning process by inserting specific subgraph tokens which are designed to capture and highlight relevant patterns or structures within the graph that are crucial for making more accurate predictions.
> > >
> > > Reasoning is a significant and valuable direction for graph prompt. While we did not explore it in this paper, we see it as a crucial area for future work. And we look forward to further exploring and demonstrating the reasoning capabilities of graph prompt in subsequent research.
> > >
> > >
> > > > 3.The authors mention that designing a universal graph prompt capable of handling complex graph data from different domains presents a significant challenge. I do not know why we want to build such a model to handle different domains without any performance gain over the semi-supervised learning model.
> > >
> > >
> > > Firstly, we would like to emphasize the importance of transfer learning across domains, particularly when sample labels are **scarce**. Consider Company A, a large financial institution with extensive customer data, which has developed a highly accurate fraud detection model using graph-based models. In contrast, Company B, a smaller micro-lending company, lacks the rich dataset of Company A, making it difficult to build an equally reliable model.
> > >
> > > Despite limited data, Company B could benefit from Company A's model since both operate in the financial sector with similar transaction types. However, due to **legal** and **privacy** constraints, direct data sharing is not feasible. In this case, a cross-domain model becomes an efficient solution, demonstrating the commercial value of domain transfer.
> > >
> > > Finally, if the graph prompt is not universally effective across different domains, it could compromise the success of transferring the pre-trained model. Thus, developing a universal graph prompt method is crucial for ensuring consistent effectiveness in various domains.
> > >
> > > [1] Graphprompt: Unifying pre-training and downstream tasks for graph neural networks. WWW 2023
> > >
> > > [2] Universal prompt tuning for graph neural networks. NeurIPS 2023
> > >
> > > [3] Towards Graph Prompt Learning: A Survey and Beyond. arXiv 2024

---

### Official Review · Reviewer_fofj · 2024-07-22
**Review for Submission 1790**

**Rating:** 8
**Confidence:** 5
**Correctness:** The benchmark is constructed appropri…
**Clarity:** The paper is well-written.

**Review:**

**Pros**
1. The design is reasonable and comprehensive, including various pre-training methods, state-of-the-art graph prompt methods, and diverse datasets.
2. The experiments in the paper are sufficient to provide valuable guidance for future graph prompt learning research. For example, the combination of pre-training methods and prompt methods reveals that the node-level pre-training adapts to node-level downstream tasks and the graph-level pre-training adapts to graph-level when applying graph prompt methods.
3. From my perspective, the introduction of heat maps (e.g., Figure 3), which show the performance comparisons between "Pre-train & Fine-tune" methods and graph prompt methods on diverse datasets, is novel and necessary. They intuitively demonstrate that "Pre-train & Fine-tune" methods usually experience negative transfer, while the graph prompt learning method can alleviate it significantly.
4. This paper releases an easy-to-use open-source library that greatly accelerates the progress in the field of graph prompts.

**Cons**

1. Does the "ProG" in Table 7 and Table 8 mean "All-in-one" in the Appendix, it's different from the table in Section 4?
2. Is the Flickr dataset redundant in Table 7 and Table 9? I can't see the introduction of the dataset in the paper.
3. The meaning of the symbol '↓ (\%)' could be further clarified, though I can understand it.

**Strengths:**

See the previous content.

**Additional Feedback:**

Not applicable.

**Documentation:**

Not applicable.

**Ethics:**

Not applicable.

**Limitations:**

The authors provide a discussion of the limitations.

**Opportunities For Improvement:**

I suggest the author check and revise the submission to alleviate potential typos.

**Relation To Prior Work:**

Yes, the discussion of related work is adequate and prior work is cited when appropriate.

**Summary And Contributions:**

The paper introduces ProG, the first comprehensive benchmark for graph prompt learning. ProG includes various combinations of graph pre-training and graph prompt learning methods and evaluates them across diverse datasets to assess performance, flexibility, and efficiency. Additionally, ProG has been published as an easy-to-use open-source library that streamlines the execution of various graph prompt methods, facilitating objective evaluations.

**Contributions:**

1. The author develops the first comprehensive graph prompt learning benchmark named ProG, which systematically investigates the effectiveness, flexibility, and efficiency of different state-of-the-art graph prompt methods combined with various-level graph pre-training methods (i.e., node-level, edge-level, and graph-level) on diverse graph datasets under the few-shot settings.

2. The author demonstrates that graph prompt learning methods present significant improvements compared to traditional supervised methods and "pre-train & fine-tune" methods. In particular, graph prompt method learning has the powerful ability to reverse negative transfer effects observed in some datasets.

3. The author releases ProG, an easy-to-use open-source library to conduct various graph prompt models, to address the complexity and redundancy of existing graph prompt methods. This library can be installed directly via PyPI, allowing users to obtain graph prompt benchmarking results easily. Besides, the repository has been widely used and cloned on GitHub with more than 460 stars in such a short time.

4. The author proposes a unified framework that categorizes graph prompt methods into two types: prompts as graphs and prompts as tokens. This categorization helps in understanding and applying different graph prompt strategies effectively.

---

> ### Author Rebuttal · Authors · 2024-08-19
>
> Thanks for your support and detailed suggestions. We appreciate your careful reading of our paper and hope our response further addresses your concerns. Please find our detailed response below.
>
> >Cons 1: Does the "ProG" in Table 7 and Table 8 mean "All-in-one" in the Appendix, it's different from the table in Section 4?
>
> Yes, the "ProG" in Table 7 and Table 8 shall be "All-in-one". We have revised this difference.
>
> >Cons 2: Is the Flickr dataset redundant in Table 7 and Table 9? I can't see the introduction of the dataset in the paper.
>
> Yes, it is a redundant item. We have revised it in our paper.
>
> >Cons 3:: The meaning of the symbol '↓ (%)' could be further clarified, though I can understand it.
>
> Thank you for giving us such valuable comments! We would like to clarify the meaning of the symbol "↓(%)" in our paper. The concrete clarification is `↓(%) means the negative transfer rate, computed by dividing the number of ↓ by the number of datasets.`
>
> Overall, we noticed your interest in our library, we will add more prompt models to 'ProG' in the future, many thanks to you again for supporting our work.

---

> ### Author Response · Authors · 2024-08-29
> **Appreciation for review**
>
> Dear Reviewer fofj,
>
> Thank you very much for the dedicated review of our paper. We are fully aware of the commitment and time your review entails. Your efforts are deeply valued by us.
>
> With 2 days remaining before the conclusion of the discussion phase, we wish to extend a respectful request for your feedback about our responses. Your insights are of immense importance to us, and we eagerly anticipate your updated evaluation. We are grateful for your acknowledgment. Furthermore, please don't hesitate to reach out if you have any further inquiries or require additional clarifications. We are fully committed to providing additional responses.
>
> Best regards,
>
> Authors

---

> > ### Comment · Reviewer_fofj · 2024-08-29
> >
> > Thanks for your reply, I am glad to see a comprehensive benchmark and user-friendly library in the field of graph prompt, I'll keep my rating.

---

### Official Review · Reviewer_bbb5 · 2024-08-12
**Review for Submission1790**

**Rating:** 6
**Confidence:** 4
**Correctness:** Yes.
**Clarity:** Yes.

**Review:**

"Pre-train and Prompt" paradigm originally motivated from the Natural Language Process field is rather a new addition in the field of graph representation learning and I see a strong potential for it to enhance the field of Graph Neural Networks. To this effect, this benchmark has significance.  I appreciate the authors using datasets well known to the community, where, some Graph Prompt methods show transferability improvements when compared with classical Fine-tuning methods. An extensive evaluation on a diverse datasets and three different performance metrics have been reported which is a positive. The supplementary material provides a fair amount of detail on the methods provided in the benchmark and several more experiments for different levels of k-shot learning.
 To summarize the strengths,
1. Extensive evaluation, diverse datasets (includes homophilic and heterophilic datasets) and multiple metrics of interest are important factors for a benchmark's utility which this paper provides.
2. A publicly accessible code with clear documentation has been provided on GitHub and project website which improves its ease of use to the Graph Prompt research community.
3. Good state-of-the-art methods have been used.

Overall the paper is well written and clear.

Having said that there are still some important cons of this paper which in my opinion are critical for a benchmark papers utility.
1. Certain experimental details such as seed and number of runs that the metrics have been averaged over are missing. Such details for a benchmark paper are vital and lack of it impacts reproducibility negatively. I have looked at the paper and supplementary material and didn't find these details which I hope authors can provide or point to me in the paper in case I missed it.
2. Graph prompt methods seem to increase standard deviation by a lot. For example in Table 2, Gprompt has ~7.7% increase in performance, however almost a ~600% increase in variability. This can cloud the significance of the results and any discussion around increased variability is completely absent. This combined with my previous point may make it difficult to trust the claims of the paper which is critical for any benchmark paper as it attempts to set a standard for research community.

Some minor issues :
1. The training time analysis doesn't report or contract prompt based methods with fine-tuning based methods.
2. There are some grammatical and spelling errors that I hope authors would correct. For example,
                             -  Figure 3 "Heat Map" instead of "Head map"
                             -  Page 1 Introduction para 1 line 5  it should be "pretext with easily accessible data" instead of "access data"
                             -  Page 3 Preliminaries para 1 line 7, it doesn't read well "making them can be formulated..."
3. This maybe personal, but I would be careful using or claiming the phrase "Artificial general intelligence". This in my opinion may not be good research practice as there is no clear definition of what constitutes AGI yet.
4. Link prediction task is missing which limits the scope of the proposed benchmark especially for active research areas such as Knowledge Graphs.

**Strengths:**

1. Diverse well understood homophilic and heterophilic datasets have been used.
2. A good publicly accessible code and package has been provided with clear documentation.
3. Good state-of-the-art methods for Graph prompts such as All-In-One have been used.

**Additional Feedback:**

See review.

**Documentation:**

Yes.

**Ethics:**

No.

**Limitations:**

Negatives of graph prompt methods need to be discussed in greater detail. For example, when combining GraphMAE with All-in-One , there is 25% negative transfer for graph prompt method. Why? An insight and more discussion into when Graph prompt fails could be immensely helpful for research community.

**Opportunities For Improvement:**

Discussion about increase in standard deviation of Graph prompt methods when comparing with Fine-Tuning methods and more transparent experimental details is quite critical for a benchmark paper.  Upon providing these, I may consider revising my score.

**Relation To Prior Work:**

Yes.

**Summary And Contributions:**

This paper introduces ProG, a benchmark for graph prompt learning methods, which includes 6 pre-training methods and 5 graph prompt learning methods over 15 datasets. Graph prompt learning involves modifying downstream data by inserting additional small prompt graphs or tokens, and then reformulating the downstream task to match the pre-training task without changing the pre-trained GNN model. This is in contrast to the "Pre-train and Fine-Tune" paradigm which does model level retraining and can be quite expensive. The paper investigates performance of 5 state-of-the-art Graph Prompt methods with notable performance improvement in few-shot node classification and graph classification tasks. Moreover a good well documented library has been made publicly accessible to help further the research area in this field. Flexibility, efficiency and effectiveness analysis highlight interesting properties about Graph prompt such reversal of negative transfer.

---

> ### Author Rebuttal · Authors · 2024-08-17
>
> Thanks for your insightful comments and detailed advice. We hope our point-to-point responses can address your concerns and provide you with better clarification.
>
> >**Main Cons 1: More transparent experimental details**, such as seed and number of runs that the metrics have been averaged.
>
> We appreciate the reviewer's feedback regarding the need for more transparent experimental details. We understand the importance of providing complete information to ensure the reproducibility of our experiments.
>
> To address this concern, we have made the following clarification:
>
> 1. **Seed**: 42. For fairness and reproducibility, we assign all experiments a unified seed value (i.e., 42).
>
> 2. **Number of Runs**: 5. We sincerely apologize for the lack of clarity. In lines 200-202 of our paper, we wrote, 'xxx, we repeat the sampling process five times to construct k-shot tasks for both node and graph tasks. We then report the average and standard deviation over these five results.' **To clarify this**, we plan to include the seed and number of runs and **rephrase it** as follows: "xxx, using **a unified seed of 42**, we randomly sample five groups of k-shot classification datasets from the remaining data for each node or graph task. Then, **we run 5 experiments for each task**, with each evaluation corresponding to one of the five sampled datasets, and then report the average and standard deviation over these results."
>
> 3. **More Detailed Descriptions**: For more details, such as hyperparameter optimization and the search space, we respectfully invite the reviewers to refer to Appendix D. Additionally, we would like to clarify that during the random search process, the seed was also assigned 42.
>
> Overall, we will update the above clarification in our revision.
>
> >**Main Cons 2: Discussion about increase in standard deviation** of Graph prompt methods when comparing with Fine-Tuning methods. Graph prompt methods seem to increase standard deviation by a lot. For example in Table 2, Gprompt has ~7.7% increase in performance, however almost a ~600% increase in variability. This can cloud the significance of the results and any discussion around increased variability is completely absent.
>
> Thank you for your valuable and constructive suggestions! We greatly appreciate the time and effort you put into carefully reviewing our work, especially in highlighting the increase in standard deviation. This insight significantly enhances the comprehensiveness of our study. To address your concerns, we conducted the following discussions as follows:
>
> 1. **Analysis of Gprompt:** We found that Gprompt exhibited significant increase on the Texas dataset, primarily **due to its difficulty in effectively transferring pre-trained knowledge**, leading to negative transfer. Specifically, the performance of Gprompt on Texas was obviously lower than that of the supervised and fine-tuning methods. This negative transfer phenomenon was also observed in GPPT, resulting in a similar increase in standard deviation as seen with Gprompt. In general, compared with other graph prompt methods, Gprompt is more prone to experiencing an increase in standard deviation. Additionally, **the increase in standard deviation is consistent with findings in existing papers** [1].
> 2. **Analysis of graph prompts:** We observed that the instability in standard deviation for graph prompt methods is common, compared to fine-tuning, even with different pre-training strategies. This is mainly **due to the differing parameter spaces between fine-tuning and prompt methods**. While the smaller parameter space of prompt methods can lead to instability, when their average accuracy exceeds that of fine-tuning, their lower bound is typically higher, despite the larger standard deviation. An example of this can be seen in the performance of the GPF-plus method on node classification datasets.
>
> Kindly note that this paper aims to objectively reflect both the advantages and shortage of graph prompts. Thanks to the reviewer, we might be **the first to find that graph prompt methods indeed generally suffer from the issue of the increase in standard deviation**, which leads to performance instability. **We will point out this in our revision as an open problem**. and we believe that these insights can more effectively contribute to the growth of the community.
>
> [1] Graphprompt: Unifying pre-training and downstream tasks for graph neural networks. WWW 2023

---

> > ### Author Response · Authors · 2024-08-22
> > **Release of our rebuttal codes**
> >
> > Dear Reviewer bbb5,
> >
> > Thanks for your valuable comments again. Recently, we have polished and released the codes used for the additional experiments within our rebuttal. Here is the link: https://github.com/Barristen/ProG_rebuttal. We hope this release can guarantee the experimental analysis reliable. Moving forward, we will continue to optimize this code.
> >
> > Sincerely,
> >
> > Authors

---

> ### Author Rebuttal · Authors · 2024-08-17
>
> >**Minor Cons 1: The training time analysis** doesn't report or contract prompt based methods with fine-tuning based methods.
>
> | GCN  ($10^{-3}s$) | 2 layers | 10 layers | 50 layers | 100 layers |
> | ---------------------- | -------- | --------- | --------- | ---------- |
> | Fine-tune              | 1.78     | 3.81      | 10.98     | 22.07      |
> | All-in-one             | 2.15     | 3.91      | 10.56     | 20.89      |
> | Gprompt                | 1.89     | 3.25      | 10.01     | 20.09      |
> | GPF                    | 0.61     | 0.66      | 0.69      | 0.70       |
> | GPF-plus               | 0.68     | 0.72      | 0.73      | 0.74       |
> | GPPT                   | 2.26     | 3.99      | 10.93     | 21.63      |
>
> Thanks for your advice! To elaborate, we first compared the training time of the fine-tuning and graph prompt methods, following the settings in our paper. Since a key objective of the graph prompt technique is to reduce the adaptation time of graph foundation models for future downstream tasks, we also evaluated their training time with GNNs that have more layers. We observed that the training time for the All-in-one and Gprompt methods was slightly longer than that of fine-tuning when using a 2-layer GNN. This is because, with fewer GNN layers, the difference in the number of parameters requiring tuning between fine-tuning and graph prompt methods isn’t very pronounced. Moreover, these two graph prompt methods involve additional operations, such as similarity computation and element-wise multiplication, which require extra time. However, as the number of GNN layers increases, the advantages of these methods become more evident.
>
> In general, compared to the fine-tuning methods, graph prompts offer a natural advantage in terms of training efficiency. The results of training time are also shown in Table 1 of our submitted pdf.
>
> >**Minor Cons 2:** There are some **grammatical and spelling errors** that I hope authors would correct. For example, - Figure 3 "Heat Map" instead of "Head map" - Page 1 Introduction para 1 line 5 it should be "pretext with easily accessible data" instead of "access data" - Page 3 Preliminaries para 1 line 7, it doesn't read well "making them can be formulated..."
>
> Thank you for your valuable feedback! We have carefully checked for grammatical errors and made the necessary corrections, as you pointed out, the 'Head Map' will be replaced with 'Heat Map', 'making them can be formulated' will be chaged to 'making them be formulated'and we also found other minor typos like the "ProG" in Table 7 and Table 8 will be changed "All-in-One" in the Appendix.
>
>
> >**Minor Cons 3:** This maybe personal, but I would be **careful using or claiming the phrase "Artificial general intelligence"**. This in my opinion may not be good research practice as there is no clear definition of what constitutes AGI yet.
>
> Thank you for raising this key point! We would like to **replace** Artificial general intelligence (AGI) with **"Transfer learning (TL)"**.

---

> ### Author Rebuttal · Authors · 2024-08-17
>
> >**Minor Cons 4: Link prediction task** is missing which limits the scope of the proposed benchmark especially for active research areas such as Knowledge Graphs.
>
> Thank you for pointing out this task. Concretely, in our new experiments, we evaluated link prediction performance using various graph prompt methods with GraphCL pretext. We tested on the NELL (Knowledge Graph) and Cora (citation network) datasets. We used accuracy, micro-F1 score, and AUC-ROC as metrics in 1-shot, 3-shot, and 5-shot scenarios.
>
> On the NELL dataset, we observed that state-of-the-art prompt methods like GPF-plus significantly outperform supervised methods and pre-train & fine-tune methods by 15.91% and 11.59% on average in ACC, respectively, in the 1-shot setting. Notably, GPF-plus nearly consistently outperformed other methods, including supervised, pre-train & fine-tune, and prompt methods, on both NELL and Cora, except in the 3-shot and 5-shot settings on Cora. While All-in-One performed well on Cora, its performance was poor on NELL, likely due to the large scale and high-dimensional node features of the NELL dataset, which made it challenging for All-in-One to capture this information effectively. Additionally, it is worth noting that the fine-tuning methods exhibited negative transfer on the 1-shot setting of the NELL dataset and the 1, 3, and 5-shot settings on the Cora dataset, whereas the graph prompt methods managed to overcome this issue.
>
> Overall, graph prompts have generally shown significant advantages in link prediction tasks compared to supervised and fine-tuning methods. And below is the detailed results:
>
> **Link prediction performance of different methods using ACC, micro-F1 score, and AUROC (1-shot for NELL), the best results are in bold.**
>
> | Model       | Acc(%)         | F1-score(%)    | AUROC(%)        |
> | ----------- | -------------- | -------------- | --------------- |
> | Supervised | 79.33±10.77    | 77.48±12.70    | 85.83±9.85      |
> | Fine-tune   | 83.65±11.22    | 78.56±11.93    | 85.56±10.56     |
> | All-in-one  | 74.03±11.08    | 70.99±13.78    | 97.92±0.43      |
> | Gprompt     | 85.48±6.63     | 85.06±7.42     | 88.99±3.27      |
> | GPF         | **96.12±3.77** | **96.10±3.79** | 99.73±0.47      |
> | GPF-plus    | 95.24±6.45     | 95.12±6.68     | **100.00±0.00** |
>
>
> **Link prediction performance of different methods using ACC, micro-F1 score, and AUROC (3-shot for NELL), the best results are in bold.**
>
> | Model       | Acc(%)         | F1-score(%)    | AUROC(%)        |
> | ----------- | -------------- | -------------- | --------------- |
> | Supervised | 71.77±6.52     | 68.89±8.08     | 78.89±5.81      |
> | Fine-tune   | 73.58±5.42     | 69.89±6.58     | 80.29±6.91      |
> | All-in-one  | 76.71±11.00    | 74.15±14.30    | 98.10±0.60      |
> | Gprompt     | 90.97±1.90     | 90.95±1.90     | 95.25±2.40      |
> | GPF         | **97.00±2.12** | **96.99±2.13** | 99.90±0.12      |
> | GPF-plus    | 94.91±3.73     | 94.87±3.80     | **100.00±0.00** |
>
> **Link prediction performance of different methods using ACC, micro-F1 score, and AUROC (5-shot for NELL), the best results are in bold.**
>
> | Model       | Acc(%)         | F1-score(%)    | AUROC(%)        |
> | ----------- | -------------- | -------------- | --------------- |
> | Supervised | 73.13±10.81    | 70.01±12.88    | 79.49±9.43      |
> | Fine-tune   | 71.58±9.51     | 68.01±10.65    | 78.12±8.13      |
> | All-in-one  | 74.29±9.43     | 71.64±11.44    | 97.83±0.55      |
> | Gprompt     | 92.54±3.18     | 92.50±3.22     | 96.37±1.20      |
> | GPF         | 92.67±7.51     | 92.43±7.97     | 99.98±0.03      |
> | GPF-plus    | **96.08±5.48** | **96.03±5.58** | **100.00±0.00** |
>
>
> **Link prediction performance of different methods using ACC, micro-F1 score, and AUROC (1-shot for Cora), the best results are in bold.**
>
> | Model       | Acc(%)         | F1-score(%)    | AUROC(%)       |
> | ----------- | -------------- | -------------- | -------------- |
> | Supervised | 53.84±2.15     | 53.23±5.13     | 59.98±6.60     |
> | Fine-tune   | 52.85±1.68     | 51.53±4.18     | 57.88±4.68     |
> | All-in-one  | 59.72±6.07     | 51.13±10.03    | 53.92±2.96     |
> | Gprompt     | 54.65±4.82     | 54.08±5.02     | 56.09±6.02     |
> | GPF         | 56.02±3.86     | 51.53±7.21     | **60.74±2.26** |
> | GPF-plus    | **62.04±3.70** | **55.56±5.83** | 60.22±5.09     |
>
> **Link prediction performance of different methods using ACC, micro-F1 score, and AUROC (3-shot for Cora), the best results are in bold.**
>
> | Model       | Acc(%)         | F1-score(%)    | AUROC(%)       |
> | ----------- | -------------- | -------------- | -------------- |
> | Supervised | 55.52±2.44     | 55.38±2.75     | 60.98±8.53     |
> | Fine-tune   | 52.68±5.66     | 52.22±5.44     | 53.98±8.01     |
> | All-in-one  | **66.01±2.43** | **61.48±3.42** | 53.72±3.45     |
> | Gprompt     | 52.50±5.91     | 51.97±5.58     | 54.11±8.15     |
> | GPF         | 53.24±5.59     | 52.26±4.98     | 54.09±8.31     |
> | GPF-plus    | 60.44±4.76     | 59.49±5.19     | **65.31±5.80** |
>
> **Link prediction performance of different methods using ACC, micro-F1 score, and AUROC (5-shot for Cora), the best results are in bold.**
>
> | Model       | Acc(%)         | F1-score(%)    | AUROC (%)      |
> | ----------- | -------------- | -------------- | -------------- |
> | Supervised | 61.64±4.88     | 60.84±5.59     | 66.96±6.01     |
> | Fine-tune   | 56.94±3.25     | 56.68±3.20     | 58.29±4.63     |
> | All-in-one  | **66.54±0.93** | 62.36±1.37     | 57.81±3.25     |
> | Gprompt     | 54.84±2.67     | 54.73±2.57     | 56.08±3.83     |
> | GPF         | 56.62±3.44     | 55.39±3.74     | 58.94±5.71     |
> | GPF-plus    | 65.66±2.36     | **63.86±4.41** | **76.61±3.18** |
>
> Additionally, the above experimental results of link prediction tasks are also shown in Table 2, Table 3, and Table 4 in our submitted pdf.

---

> ### Author Rebuttal · Authors · 2024-08-17
>
> >**Limitation 1: Negatives of graph prompt methods** need to be discussed in greater detail. For example, when combining GraphMAE with All-in-One , there is 25% negative transfer for graph prompt method. Why? An insight and more discussion into when Graph prompt fails could be immensely helpful for research community.
>
> Thank you very much for your insightful comments. First, we can recap the instability of the graph prompt method mentioned earlier (the increase in standard deviation). When this instability is combined with inconsistent downstream pretraining strategies, it indeed could potentially impact the transfer effectiveness of the graph prompt method.

---

> ### Author Response · Authors · 2024-08-23
> **Friendly reminder for rebuttal**
>
> Dear Reviewer bbb5,
>
> This is a friendly reminder that we would very much value your input on our rebuttal. Your comments have been extremely helpful in improving our paper.
>
> With only one week remaining before the conclusion of the discussion phase, we wish to extend a respectful request for your feedback about our rebuttal.
>
> We highly value your insights and are eagerly awaiting your revised evaluation. Should you find our rebuttal to be informative and beneficial, we would deeply appreciate your recognition. Additionally, if you have any further questions or need additional clarification, please feel free to contact us. We are fully dedicated to engaging and providing support during this important discussion phase.
>
> Thanks for your time again.
>
> Best regards,
>
> Authors

---

> ### Comment · Reviewer_bbb5 · 2024-08-23
> **Rebuttal**
>
> I have revised my score to a 6 after author rebuttals. They have addressed most of my concerns.

---

> > ### Author Response · Authors · 2024-08-23
> > **Official Comment by Authors**
> >
> > Dear Reviewer bbb5,
> >
> > We greatly appreciate your positive feedback on our rebuttal and are readily available to address any questions you may have.
> >
> > Thank you for your time and consideration.
> >
> > Best regards,
> >
> > Authors

---

### Official Review · Reviewer_v69Z · 2024-08-13
**Review for "ProG: A Graph Prompt Learning Benchmark"**

**Rating:** 7
**Confidence:** 2

**Review:**

This paper proposes a comprehensive benchmark for graph prompt learning. The proposed benchmark integrates 6 pre-training methods and 5 state-of-the-art graph prompt techniques, evaluated across 15 diverse datasets. Such benchmark is unified into an open-sourced library named ProG for easy experiments and benchmarking.

Pros of this paper are:
1. This paper addresses several challenges in emerging method graph prompt learning, such as no unified framework to incorporate diverse graph prompt models, and systematically evaluate the quality of graph prompts methods.
2. This framework is summarized in a GitHub code repository for researchers to easily compare various graph prompt techniques.
3. This paper demonstrates effectiveness (quality performance on various datasets), flexibility, and efficiency (time and space cost of various graph prompting methods) for multiple SoTA graph prompting methods.

Cons of this paper.
1. It lacks justification on why classifying the graph prompts as three basic components: prompt tokens, token structure, and insert patterns. No existing literatures are cited to show rationale.
2. It is unclear on why we need"flexility analysis" from line 117 - 118. More descriptions are needed and literature references need to be included.

**Strengths:**

1. This paper addresses several challenges in emerging method graph prompt learning, such as no unified framework to incorporate diverse graph prompt models, and systematically evaluate the quality of graph prompts.
2. This framework is summarized in a GitHub code repository for researchers to easily compare various graph prompt technique.
3. This paper demonstrates effectiveness (quality performance on various datasets), flexibility, and efficiency (time and space cost of various graph prompting methods) for multiple SoTA graph prompting methods.

**Additional Feedback:**

None

**Clarity:**

Yes this paper is well written with good structure and clear objective. In addition, this paper illustrates the objective with extensive experimental results.

**Correctness:**

The paper proposes a benchmark. The evaluation methods and experiment design including how to split the data, hyperparameter optimization, and selection of baseline methods are designed appropriately. Experiment results are comprehensive to make conclusions.

**Documentation:**

This paper is not about proposing a new dataset.

**Ethics:**

None.

**Limitations:**

Authors have mentioned the limitations of their works in Section 6.

**Opportunities For Improvement:**

1. It lacks justification on why classifying the graph prompts as three basic components: prompt tokens, token structure, and insert patterns. No existing literatures are cited to show rationale.
2. It is unclear on why we need"flexility analysis" from line 117 - 118. More descriptions are needed and literature references need to be included.

**Relation To Prior Work:**

This work claims to be the first benchmark for graph prompt learning. Thus there is no previous contribution being discussed.

**Summary And Contributions:**

This paper proposes a comprehensive benchmark for graph prompt learning. The proposed benchmark integrates 6 pre-training methods and 5 state-of-the-art graph prompt techniques, evaluated across 15 diverse datasets. Such benchmark is unified into an open-sourced library named ProG for easy experiments and benchmarking. Specifically, this paper addresses several challenges in emerging method graph prompt learning, such as no unified framework to incorporate diverse graph prompt models, and systematically evaluate the quality of graph prompts methods.

---

> ### Author Rebuttal · Authors · 2024-08-19
>
> Thanks for your positive and constructive feedback. We sincerely appreciate your recognition of our contributions to the field of graph prompt learning. We would like to address your concerns as follows:
>
> >Cons 1: It lacks justification on why classifying the graph prompts as three basic components: prompt tokens, token structure, and insert patterns. No existing literatures are cited to show rationale.
>
> Thanks for your valuable suggestions! We are sorry for lacking the corresponding literatures. We have involved **[1], which introduces the three components in Section 2.2 (2)i and Section 5.1**, as the literature to show the rationale.
>
> Concretely, the nature of graph prompts is to manipulate downstream graph data to align them to upstream graphs, guaranteeing the pre-trained knowledge transfer. The introduced **three types of components are the smallest manipulation units of a graph prompt module**. To better help you understand what are prompt tokens, token structure, and insert patterns, we would like to make the following further clarifications:
>
> 1. **Prompt tokens**: typically refer to the additional (learnable) vectors or features that are appended to the original graph data (e.g., added to node features or representing specific information relevant to the downstream task). These tokens are the key unit that determines the basic manipulations.
>
> 2. **Token structure**: describes how multiple prompt tokens are organized and interrelated. It aims to introduce additional relational patterns to the original graphs based on the prompt tokens.
>
> 3. **Insert patterns**: define how the token structure should be integrated into the original graph. This aims to determine which parts of the graph require the inclusion of additional relational patterns from the token structures.
>
> These three components work together organically to lead to different graph prompts, which can be broadly classified into two types: prompt as tokens (focusing more on the design of the prompt tokens themselves) and prompt as graphs (focusing more on the organic integration of the three components).
>
> A schematic graph for visualizing the three components is included in the uploaded pdf file.
>
> >Cons 2: It is unclear on why we need"flexility analysis" from line 117 - 118. More descriptions are needed and literature references need to be included.
>
> Thank you for your constructive comments. Specifically, flexibility analysis is to assess how powerful graph prompts are. One way to do this is by using graph prompts to restore a set of perturbed graphs (e.g., “changing node features”, “adding or removing edges,” etc.) via a series of manipulations. The degree to which these graphs are restored reflects the flexibility of the different graph prompts. This restoration degree is measured by the error bound defined on page 3, lines 103-105, of the paper. In detail, the measurement of graph prompt flexibility is computed by the error bound between the original graph and the restored graph w.r.t. their representations from the pre-trained graph model. More details mentioned above can be found in [2] (Sections 3.3 & 3.4) and [3] (Sections 3.5.2 & 4.6). Our flexibility analysis experiments were conducted following [2] and [3] to ensure their validity. We will include the above clarification and references in the flexibility analysis section of the paper.
>
> In general, we would like to thank the reviewer again for pointing out this aspect that needed clarification. We hope our explanations of the graph prompt can help you to fully understand this field. If you are satisfied with our rebuttal, could we kindly request you raise the confidence score?
>
> [1] Graph prompt learning: A comprehensive survey and beyond. arXiv 2024
>
> [2] Universal prompt tuning for graph neural networks. NeurIPS 2023
>
> [3] All in One: Multi-task Prompting for Graph Neural Networks. KDD 2023

---

> ### Author Response · Authors · 2024-08-29
> **Appreciation for review**
>
> Dear Reviewer v69Z,
>
> Thank you sincerely for your thorough review of our paper. We greatly appreciate the time and effort you have devoted to providing such valuable feedback.
>
> As the rebuttal phase nears its conclusion, with two days remaining.  We are grateful for your consideration, and please feel free to reach out if you have any further questions or need additional clarifications. We remain fully committed to addressing any concerns.
>
> Best regards,
>
> Authors

---

### Author Response · Authors · 2024-08-21
**Response Summary to All Reviewers**

Dear Reviewers,

We sincerely thank all the reviewers (v69Z, bbb5, fofj, AAD3) for their valuable feedback. We are glad that the reviewers appreciated the significance of our benchmark (v69Z, bbb5, fofj), the interest and novelty of our proposed framework (v69Z, bbb5, fofj), the soundness of our technical quality (v69Z, bbb5, fofj, AAD3), the comprehensiveness of our experiments (v69Z, bbb5, fofj), and the overall quality of our paper's writing (v69Z, bbb5, fofj, AAD3).

We have made every effort to faithfully address your comments in the responses, as suggested by the reviewers:

* Improved clarity of graph prompt and its applicability ([AAD3](https://openreview.net/forum?id=wqo6xEMyk9&noteId=PJ7BPNY3Rc)).
* Improved clarity of the three basic components of the graph prompts and the necessity to conduct "flexibility analysis" ([v69Z](https://openreview.net/forum?id=wqo6xEMyk9&noteId=vzRZV2004E)).
* More transparent experimental details ([bbb5](https://openreview.net/forum?id=wqo6xEMyk9&noteId=O9fjMbeE85)).
* Additional discussion about the increase in standard deviation of graph prompts ([bbb5](https://openreview.net/forum?id=wqo6xEMyk9&noteId=O9fjMbeE85)).
* Additional analysis of the training time between fine-tuning and graph prompt methods. ([bbb5](https://openreview.net/forum?id=wqo6xEMyk9&noteId=xRWmWxKunF)).
* Additional revision of some typos errors and the phrase "Artificial general intelligence" ([bbb5](https://openreview.net/forum?id=wqo6xEMyk9&noteId=xRWmWxKunF)).
* Additional experimental results to evaluate the performance of supervised, fine-tuning, and graph prompt methods on 1/3/5-shot link prediction tasks (One knowledge graph dataset and One citation dataset) ([bbb5](https://openreview.net/forum?id=wqo6xEMyk9&noteId=tdr37MxwiN)).
* Additional discussion about the negatives of graph prompt methods ([bbb5](https://openreview.net/forum?id=wqo6xEMyk9&noteId=PdLQrZmGAz)).
* Additional clarity of some information, such as the meaning of the symbol "↓ (%)" in the paper ([fofj](https://openreview.net/forum?id=wqo6xEMyk9&noteId=hZjwGhrlxw)).
* Additional cited papers.


We will accordingly include the content mentioned above in our revised version. **As the deadline for discussion (closes on Aug 31)** is fast approaching, we would be grateful if you could allocate some time to review our responses.

Thanks for all the reviewers' time again.

Best regards,

Authors

---

### Decision · Program_Chairs · 2024-09-26

**Decision:**

Accept (Poster)

**Comment:**

The paper introduces ProG, the first comprehensive benchmark for graph prompt learning. ProG includes various combinations of graph pre-training and graph prompt learning methods and evaluates them across diverse datasets to assess performance, flexibility, and efficiency. Additionally, ProG has been published as an easy-to-use open-source library that streamlines the execution of various graph prompt methods, facilitating objective evaluations.
After extensive discussion, most reviewers agree to accept it. So I also suggest to accept it.